# VODCA v2: Multi-sensor, multi-frequency vegetation optical depth data for long-term canopy dynamics and biomass monitoring

Ruxandra-Maria Zotta[1], Leander Moesinger[1], Robin van der Schalie[2], Mariette Vreugdenhil[1], Wolfgang Preimesberger[1], Thomas Frederikse[2], Richard de Jeu[2], and Wouter Dorigo[1]

[1]Department of Geodesy and Geoinformation, TU Wien, Wiedner Hauptstraße 8, 1040 Wien
[2]Planet Labs, Wilhelminastraat 43A, 2011 VK Haarlem, the Netherlands

**Correspondence:** Ruxandra-Maria Zotta (ruxandra-maria.zotta@geo.tuwien.ac.at)

**Abstract.** Vegetation optical depth (VOD) is a model-based indicator of the total water content stored in the vegetation canopy derived from microwave Earth observations. As such, it is related to vegetation density, abundance, and above-ground biomass (AGB). Moesinger et al. (2020) introduced the global microwave VOD Climate Archive (VODCA v1), which harmonises VOD retrievals from several individual sensors into three long-term, multi-sensor VOD products in the C-, X- and Ku fre-
quency bands, respectively. VODCA v1 was the first VOD dataset spanning over 30 years of observations, thus allowing the monitoring of long-term changes in vegetation. Several studies have used VODCA in applications such as phenology analysis, drought monitoring, gross primary productivity monitoring, and the modelling of land evapotranspiration, live fuel moisture, and ecosystem resilience.

This paper presents VODCA v2, which incorporates several methodological improvements compared to the first version
and adds two new VOD datasets to the VODCA product suite. The VODCA v2 products are computed with a novel weighted merging scheme based on first-order autocorrelation of the input datasets. The first new dataset merges observations from multiple sensors in the C-, X- and Ku frequencies into a multi-frequency VODCA CXKu product indicative of upper canopy dynamics. VODCA CXKu provides daily observations in a 0.25° resolution for the period 1987 - 2021. The second addition is an L-band product (VODCA L), based on the SMOS and SMAP missions, which in theory is more sensitive to the entire
canopy, including branches and trunks. VODCA L covers the period 2010 - 2021, has a temporal resolution of 10 days and a spatial resolution of 0.25°. The sensitivity of VODCA CXKu to the upper vegetation layer and that of VODCA L to above-ground biomass (AGB) are analysed using independent vegetation datasets.

VODCA CXKu exhibits lower random error levels and improved temporal sampling compared to VODCA v1 single-frequency products. It provides complementary spatio-temporal information to optical vegetation indicators containing addi-
tional information on the state of the canopy. As such, VODCA CXKu shows moderate positive agreement in short vegetation (Spearman's R: 0.57) and broadleaf forests (Spearman's R: 0.49) with Fraction of Absorbed Photosynthetically Active Radiation from MODIS. VODCA CXKu also shows moderate agreement with the slope of the backscatter incidence angle relation of Metop ASCAT in grassland (Spearman's R: 0.48) and cropland (Spearman's R: 0.46). Additionally, VODCA CXKu shows temporal patterns similar to the Normalised Microwave Reflection Index (NMRI) from in situ L-band GNSS measurements
of the Plate Boundary Observatory (PBO) and sapflow measurements from SAPFLUXNET. VODCA L shows strong spatial

agreement (Spearman's R: 0.86) and plausible temporal patterns with respect to yearly AGB maps from the Xu et al. (2021) dataset. VODCA v2 enables monitoring of plant water dynamics, stress and biomass change and can provide insights even in areas that are scarcely covered by optical data (i.e., due to cloud cover).

VODCA v2 is open access and available at: https://doi.org/10.48436/t74ty-tcx62 (Zotta et al., 2024).

## 1 Introduction

Vegetation attenuates microwave radiation emitted or reflected by the Earth's surface. This attenuation can be quantified through a metric known as Vegetation Optical Depth (VOD), which can be calculated both from passive and active microwave satellite observations (Vreugdenhil et al., 2016). Field studies have found that VOD is directly connected to vegetation water content (VWC) (Jackson and Schmugge, 1991; Sawada et al., 2015). VOD is influenced by various factors, including the density and relative moisture content of the vegetation, as well as the wavelength domain of the observations (Mo et al., 1982; Jackson and Schmugge, 1991; Kerr and Njoku, 1990; Owe et al., 2008). The sensitivity of VOD to the uppermost layer of vegetation increases with shorter measurement wavelengths (Tian et al., 2018; Konings et al., 2019). Consequently, spatial and temporal patterns observed at higher frequencies, such as those in the C-, X- and Ku-bands, tend to agree more closely with dynamics in the upper canopy (Teubner et al., 2018; Schmidt et al., 2023). Similarly, dynamics observed at lower frequencies, like P- and L-band, correspond more closely to those of overall above-ground biomass (AGB), including branches and trunks (Chaparro et al., 2019; Olivares-Cabello et al., 2021; Schmidt et al., 2023).

Compared to vegetation indicators in the optical domain, VOD offers distinct advantages, as it is unaffected by atmospheric conditions and the influence of solar illumination (Li et al., 2021). Due to its versatility, VOD has found utility in a wide range of applications, including monitoring drought and vegetation conditions (Van Dijk et al., 2013; Crocetti et al., 2020; Kumar et al., 2021; Moesinger et al., 2022; Vreugdenhil et al., 2022; Dorigo et al., 2021, 2022; Zotta et al., 2023), phenology analysis (Jones et al. (2011), Jones et al. (2014), Dannenberg et al. (2020)), and biomass monitoring (Liu et al. (2015), Rodríguez-Fernández et al. (2018), Brandt et al. (2018), Fan et al. (2019), Mialon et al. (2020), Wigneron et al. (2021); Qin et al. (2021), Bousquet et al. (2021), Olivares-Cabello et al. (2021), Yang et al. (2022), Yang et al. (2023)). It is also instrumental in estimating the likelihood of fire occurrence and monitoring fuel moisture (Forkel et al., 2017, 2019, 2023; Schmidt et al., 2023; Mukunga et al., 2023). VOD's applicability extends to crop yield assessment (Chaparro et al., 2018; Mateo-Sanchis et al., 2019) and prediction (Büechi et al., 2022). It has also been used to estimate gross primary production (Teubner et al., 2018, 2019, 2021; Wild et al., 2022) and to model land evapotranspiration (Martens et al., 2017). Furthermore, VOD contributes to the understanding of ecosystem resilience (Boulton et al., 2022; Smith et al., 2022, 2023)) and aids in assessing vegetation responses to precipitation (Harris et al., 2022).

VOD has been derived from various satellite radiometers (Owe et al., 2008; Meesters et al., 2005; Konings et al., 2016; Wigneron et al., 2007) and scatterometers (Vreugdenhil et al., 2016; Liu et al., 2023). However, these sensors come with varying lifespans and exhibit distinctive characteristics depending on their observation frequencies, incidence angles, orbital

characteristics, radiometric quality, and spatial coverage. To conduct long-term studies, merging data from multiple satellite sensors and addressing the systematic biases among them becomes necessary.

Moesinger et al. (2020) introduced the global long-term microwave Vegetation Optical Depth Climate Archive (VODCA v1) by combining VOD retrievals derived using the Land Parameter Retrieval Model (LPRM; Owe et al. (2008); Van der Schalie et al. (2017)) from multiple passive sensors: the Special Sensor Microwave/Imager (SSM/I), the Microwave Imager on board the Tropical Rainfall Measuring Mission (TMI), the Advanced Microwave Scanning Radiometer – Earth Observing System (AMSR-E), Windsat and the Advanced Microwave Scanning Radiometer 2 (AMSR2). VODCA v1 provides separate

products for microwave observations in three different spectral frequency bands: C-band (period 2002 – 2018), X-band (1997 – 2018), and Ku-band (1987 – 2017). This allows for preserving the unique sensitivity of the individual frequencies to the structural elements of the canopy. VODCA v1 harmonizes the VOD observations by first scaling SSM/I, TMI and Windsat to the climatology of AMSR-E VOD. AMSR2 VOD is scaled to TMI VOD (which in the first step was rescaled to AMSR-E) within the orbital coverage, i.e. within latitudes 38° N and S. Outside of this latitudinal range, AMSR2 is scaled to the last

three years of AMSR-E VOD instead, even though the sensors have no temporal overlap. After scaling the sensor data, the temporally overlapping observations are fused by taking their average.

Although VODCA v1 is a state-of-the-art dataset for long-term analysis, it also faces several limitations (Tagesson et al., 2021): A notable constraint lies in the approach used for merging AMSR2 VOD data. The scaling approach described above has resulted in a spatial break in trends, specifically in North America at 35° N (Moesinger et al., 2020). Additionally, averag-

ing temporally overlapping observations without considering their individual quality characteristics (i.e., through "unweighted averaging") equally weighs high-quality and noisy observations, e.g., those affected by residual radio frequency interference (RFI). Furthermore, even though the single-frequency products provided by VODCA v1 retain the unique response to vegetation characteristics of each band, they have different lifespans, with only VODCA v1 Ku-band covering 30 years of data, thus being the only product to fulfil the World Meteorological Organization standard period for calculating climate normals

(Organization, 2017). Moreover, the single-frequency products have occasional observation gaps related to the observation wavelength. Given that all three VODCA frequencies, despite their small differences in sensitivity, represent the upper vegetation layer (Moesinger et al., 2020, 2022; Wild et al., 2022), combining them would lead to an improved temporal sampling. Thus, the resulting product could be a robust alternative for studying long-term canopy dynamics. Moreover, VODCA v1 does not include a product in the L-band frequency, which has shown to be useful for many purposes, most importantly, for

monitoring AGB (Rodríguez-Fernández et al., 2018).

Therefore, to complement VODCA v1, we introduce two new datasets, hereafter referred to as VODCA v2. First, we present the methodological improvements compared to the previous version, including a new merging approach. Second, we introduce a multi-frequency merged product of unprecedented coverage (34 years), named VODCA CXKu, with lower random error levels and better temporal and spatial coverage than the single-frequency products. Third, we introduce an L-band product

(VODCA L) obtained by merging LPRM-derived VOD observations from the Soil Moisture and Ocean Salinity (SMOS) and the Soil Moisture Active Passive (SMAP) missions covering the period 2010 - 2021. VODCA v2 does not encompass an update of the single-frequency C-, X- and Ku products, as the novel merging methodology presented in this manuscript particularly

affects the merging of multiple frequencies, which have their specific sensitivities to noise. To evaluate the new data records and assess which ecosystem canopy dynamics are represented by VODCA CXKu, we compare it to other satellite and in situ variables. We use the fraction of Absorbed Photosynthetically Active Radiation (fAPAR) derived from optical remote sensing because, theoretically, there is a strong link between plant water status and the capacity of vegetation canopies to intercept solar radiation (Cammalleri et al., 2022). We also use the slope of the backscatter incidence angle relation of Metop ASCAT (Vreugdenhil et al., 2016). This radar observable is sensitive to vegetation water content and fresh biomass (Steele-Dunne et al., 2019; Petchiappan et al., 2022). Additionally, we use two ground-based vegetation datasets: Normalized Microwave Reflection Index (NMRI) measurements obtained from GPS reflectometry and sapflow observations from the SAPFLUXNET network. We evaluate the temporal and spatial sensitivity of VODCA L to biomass using yearly AGB maps. We conclude the paper by discussing the strengths and limitations of the products as well as potential future improvements.

## 2 Data

### 2.1 Vegetation optical depth datasets

#### 2.1.1 The Land Parameter Retrieval Model (LPRM)

VODCA v2 uses VOD datasets produced by Planet Labs in the framework of the European Space Agency Climate Change Initiative (CCI) (https://climate.esa.int/en/projects/soil-moisture/) and the Copernicus Climate Change Service (C3S) (https://climate.copernicus.eu/). For C-, X- and Ku-band frequencies, these datasets are derived from passive microwave radiometer data through LPRM v7, in the case of L-band through LPRM v6.2 (Dorigo et al., 2023). These are the algorithm versions used in the ESA CCI Soil Moisture v08.1 (Dorigo et al., 2023). LPRM is based on the radiative transfer model (RTM) proposed by Mo et al. (1982) and simultaneously retrieves soil moisture and VOD (Meesters et al., 2005) from V- and H-polarized microwave observations. LPRM distinguishes itself from other retrieval algorithms (e.g., Jackson (1993); Mladenova et al. (2014); Konings et al. (2016)) by its applicability to a wide range of frequencies (i.e., 1-20 GHz) and by using an analytical solution proposed by Meesters et al. (2005) for the derivation of VOD. Additionally, LPRM uses a frequency-dependent parametrization independent of land cover and is thus unconstrained by any ancillary vegetation data (Van der Schalie et al., 2017). LPRM uses land surface temperature (LST) data derived from Ka-band observations (Holmes et al., 2009) for the C-, X- and Ku-band sensors. For the L-band sensors, LST is derived from an intercalibrated dataset based on six passive microwave sensors as described in van der Schalie et al. (2021).

#### 2.1.2 Passive microwave sensor data

The passive microwave sensors used in VODCA v2 are presented in Fig. 1 and Table 1. The sensors newly introduced with this version are the Special Sensor Microwave Imager (SSM/I) F17, the Global Precipitation Measurement (GPM) Microwave Imager (GMI), the Soil Moisture and Ocean Salinity (SMOS) Microwave Imaging Radiometer using Aperture Synthesis (MI-

RAS) and Soil Moisture Active Passive (SMAP) radiometer (Van der Schalie et al., 2017). Only nighttime retrievals are used since they are proven to have a lower temperature-related error than daytime retrievals (Owe et al., 2008).

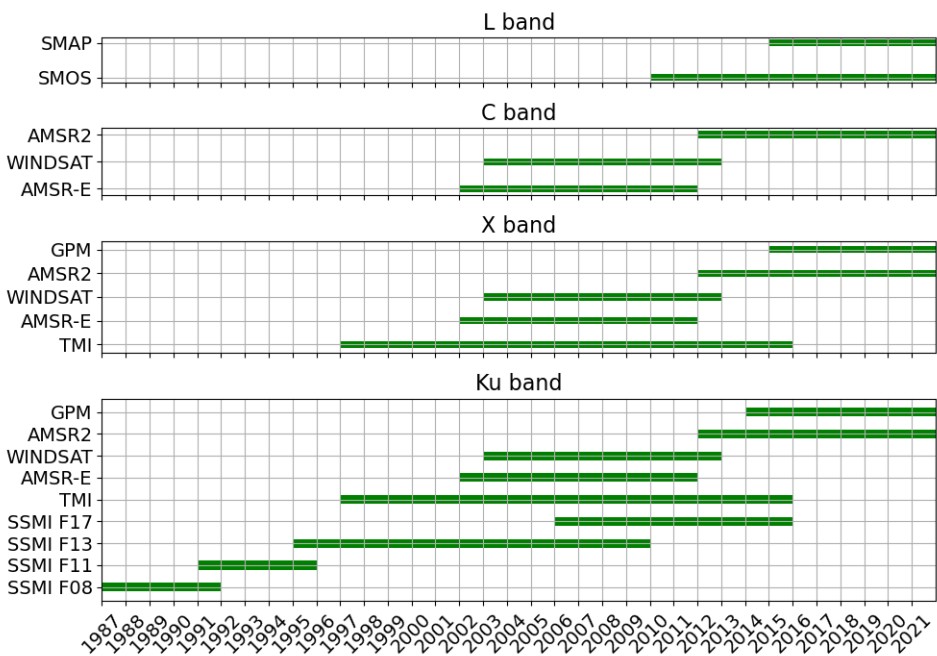

**Figure 1.** Timeline of input sensors used in VODCA v2.

The Special Sensor Microwave / Imager (SSM/I) is carried onboard a series of DMSP (Defence Meteorological Satellite Program) satellites. Out of the seven frequencies used by SSM/I to take global measurements (Wentz, 1997), we use daily Ku-band retrievals from F08, F11, F13 and F17 spanning from June 1987 to April 2015. To ensure consistency between the SSM/I sensors, VOD observations are retrieved from brightness temperature (Tb) data from the intercalibrated Fundamental Climate Data Record (FCDR) of Tb (Berg et al., 2016, 2018; Berg, 2021).

The Tropical Rainfall Measuring Mission (TRMM) Microwave Imager (TMI) has a non-polar orbit between 35° N and S, leading to increased coverage and varying overpass times in that region. TMI provides measurements in nine frequencies, of which we use X- (10.7 GHz) and Ku-band (19.4 GHz) (Kummerow et al., 1998).

    The Advanced Microwave Scanning Radiometer – Earth Observing System (AMSR-E) onboard the EOS Aqua (NASA) satellite provided global measurements between 19 June 2002 and 27 September 2011. Measurements are available in six

frequencies, of which we use C- (6.9 GHz), X- (10.7 GHz) and Ku-band (18.7 GHz) (Kawanishi et al. (2003); Knowles et al. (2006)).

**Table 1.** Specifications of the VODCA v2 input sensors. AECT shows the local ascending equatorial crossing times, while DECT shows the local descending equatorial crossing times. * The second set of footprints shown for TMI results from the altitude boosting, which happened in 2001, from 350 km to 400 km.

| Sensor | Timespan used | Frequency[GHz] | Footprint size | Nighttime overpass | Remarks |
|---|---|---|---|---|---|
| SMOS | Jan. 2010 – Dec. 2021 | 1.4 (L-band) | 43 km x 43 km | 06:00, AECT | Incidence angles: 37.5, 42.5, 47.5, 52.5, 57.5 |
| SMAP's radiometer | Mar. 2015 – Dec. 2021 | 1.41 (L-band) | 39 km x 47 km | 06:00, DECT | |
| AMSR-E | Jun. 2002 – Sep. 2011 | 6.9 (C-band) 10.7 (X-band) 18.7 (Ku-band) | 75 km x 43 km 51 km x 29 km 27 km x 16 km | 01:30, DECT | |
| AMSR2 | Jun. 2012 – Dec. 2021 | 6.9 (C-band) 7.3 (C2-band) 10.7 (X-band) 18.7 (Ku-band) | 62km x 35 km 58 km x 34 km 58 km x 34 km 22 km x 14 km | 01:30, DECT | |
| Windsat | Feb. 2003 – Jul. 2012 | 6.8 (C-band) 10.7 (X-band) 18.7 (Ku-band) | 39 km x 71 km 25 km x 38 km 16 km x 27 km | 06:00, DECT | |
| TMI | Dec. 1997 – Apr. 2015 | 10.7 (X-band)  19.4 (Ku-band) | 63 km × 37 km / 72km × 43 km*  30 km x 18 km / 35 km x 21 km* | Asynchronous | Coverage 35°N - 35°S |
| GPM | Mar. 2014 - Dec. 2021 | 10.7 (X-band)  18.7 (Ku-band) | 19 km x 32 km  18 km x 11 km | Asynchronous | Coverage below 70°N |
| SSM/I F8 | Jul. 1987 - Dec. 1991 | 19.4 (Ku-band) | 69 km x 43 km | 06:10, DECT | |
| SSM/I F11 | Dec. 1991 – May 1995 | 19.4 (Ku-band) | 69 km x 43 km | 05:00, DECT | |
| SSM/I F13 | May 1995 – Apr. 2009 | 19.4 (Ku-band) | 69 km x 43 km | 05:51, DECT | |
| SSM/I F17 | Dec. 2006 – Apr. 2015 | 19.4 (Ku-band) | 69 km x 43 km | 06:20, DECT | |

WINDSAT onboard Coriolis is a multi-frequency polarimetric microwave radiometer developed by the Naval Research Laboratory. It is in an 840 km sun-synchronous orbit since January 2003 and provides global measurements in five frequencies, of which we use C- (6.8 GHz), X- (10.7 GHz), and Ku-band (18.7 GHz) (Gaiser et al., 2004). Even though WINDSAT is still operational, we did not use data past July 2012 due to restricted access.

The Advanced Microwave Radiometer 2 (AMSR2) onboard GCOM-W (Global Change Observation Mission – Water) initiated by JAXA (Japan Aerospace Exploration Agency) is the follow-on instrument of AMSR-E installed on Aqua. AMSR2 provides daily measurements in six frequencies, similar to AMSR-E, between 7 GHz to 89 GHz, of which we use C-(6.9 GHz), X- (10.7 GHz) and Ku-band (18.7 GHz). Compared to AMSR-E, it incorporates a second C-band channel (C2) at 7.3 GHz aimed at mitigating the radio frequency interference (RFI) of 6.9 GHz (Meier et al., 2018).

GPM is a follow-up mission of TRMM initiated by NASA and JAXA and includes a consortium of international partners whose microwave sensors constitute the GPM constellation. The GPM core observatory launched on February 27, 2014, carries the GPM microwave imager (GMI) which provides daily measurements in 10–183 GHz of which we use the X- (10.65 GHz) and Ku-band (18.7 GHz). The GPM has a non-polar orbit and provides global coverage, except for latitudes higher than 70° (Draper et al., 2015).

The SMOS mission is ESA's second Earth Explorer mission and was launched in November 2009. SMOS carries the Microwave Imaging Radiometer with Aperture Synthesis (MIRAS), which is an interferometric L-band (1.4 GHz) 2-D radiometer that takes measurements for multiple incidence angles between 0 - 65°. The observations have a temporal resolution of 2 to 3 days (Kerr et al., 2010). Although SMOS operates in a protected band (1400–1427-MHz), RFI affects observations in many areas of the world (Oliva et al., 2012).

The SMAP mission, designed to map soil moisture and determine freeze/thaw state was launched in January 2015. It carries an L-band radiometer (1.41 GHz) that takes observations at an incidence angle of 40°. SMAP's radiometer has a temporal resolution of 2 to 3 days (Entekhabi et al., 2010).

## 2.2 Ancillary data

ERA5-Land is a global land-surface reanalysis dataset, which provides hourly output starting in 1981, with a 9 km spatial resolution (Muñoz-Sabater et al., 2021). We use ERA5-Land upper soil temperature (stl1; 0-7 cm; Muñoz-Sabater et al. (2021)) to mask VOD observations recorded whenever the soil temperature was below 275.15 K (2°C). Observations under frozen conditions are not used because the dielectric properties of water change drastically. ERA5-Land *stl1* is used in addition to the internal flagging coming with LPRM, which makes use of the K-, Ku, and Ka-bands to detect frozen conditions (van der Vliet et al., 2020).

## 2.3 Evaluation data

### 2.3.1 MODIS fAPAR

To evaluate the plausibility of temporal patterns of VODCA CXKu, we used the fraction of absorbed photosynthetic radiation (fAPAR) derived from the Moderate-resolution Imaging Spectroradiometer (MODIS) onboard the Terra and Aqua satellites (DOI:10.5067/MODIS/MCD15A2H.061; Myneni and Park (2021)). fAPAR is a biophysical variable that represents the fraction of radiation in the range of 400-700 nm which is absorbed by the green elements of the vegetation canopy for photosynthesis (Myneni et al., 2002) and is expressed as a non-dimensional value. fAPAR is a fundamental quantity related to the photosynthetic processes of plants, making it a pivotal indicator of the intensity of the terrestrial carbon cycle (Mason et al., 2010). Changes in VOD and fAPAR are expected to correlate because high VWC means more leaf tissue, which leads to more photosynthesis and, therefore, to higher fAPAR. Moreover, we expect agreement between the vegetation indicators because plant water status significantly influences a canopy's ability to intercept solar radiation (Osakabe et al., 2014). Both VOD and fAPAR have been widely used for vegetation condition monitoring. To match the VODCA dataset, we aggregated the native 500 m fAPAR data to the VODCA grid by taking the average. We use fAPAR data from February 2000 to August 2020. For the analyses using fAPAR, VODCA CXKu has been averaged to 8-daily observations to match its temporal resolution.

### 2.3.2 ASCAT Slope

We compare the temporal patterns of VODCA CXKu with active microwave remote sensing data from the ASCAT scatterometer onboard the Metop satellites. The instruments measure vertically polarized backscatter at incidence angles between 25° and 65° recorded at a 5.25 GHz frequency (C-band). ASCAT slope, derived using the TU Wien Soil Moisture Retrieval algorithm (Hahn et al., 2017), is a parameter of the second-order Taylor polynomial used to describe the incidence angle dependence of backscatter. The slope is sensitive to scattering mechanisms, where surface scattering leads to a steep slope, and volume scattering causes scattering in all directions, thus leading to a flatter slope. With increasing vegetation density, the volume scattering increases, and the slope flattens (Vreugdenhil et al., 2020). It has been shown that the slope is correlated with vegetation density (Hahn et al. (2017), Vreugdenhil et al. (2017)), above-ground fresh biomass (Steele-Dunne et al., 2019) and vegetation phenology and water status (Pfeil et al. (2020), Petchiappan et al. (2022)), similar as VOD. These studies also outline the importance of further research to overcome the limited understanding of the spatio-temporal dynamics of the slope parameter. As such, ASCAT slope is a relatively young parameter that has not yet been fully understood but can potentially offer valuable insight into vegetation water dynamics across a diverse range of biomes. Therefore, comparing VODCA with ASCAT slope serves more as a mutual evaluation of patterns, driven by similar vegetation properties, and less as a validation of the VODCA dataset. We argue that it is beneficial to provide such a comparison because the ASCAT slope is also derived from microwave remote sensing and uses radar observations. Therefore, it is an entirely independent dataset.

The ASCAT slope dataset used in this study is calculated from the EUMETSAT Metop-A ASCAT SZR Level 1b Fundamental Climate Data Record, which was pre-processed as described in (Hahn et al., 2017). A dynamic slope is calculated from ASCAT backscatter and incidence angle using the method developed by (Melzer, 2013) and demonstrated by (Hahn et al.,

2017). This method yields slope values for each day based on an Epanechnikov kernel with a half-width window of 21 days. We use data only from Metop-A, for 2007 to 2021, on descending overpass with 9:30 am (local) overpass time. This time is considered advantageous from a plant physiology point of view because the impact of dew should be less than at pre-dawn values (Steele-Dunne et al., 2019). We resampled the ASCAT slope dataset to match the VODCA grid by averaging all points within a VODCA grid cell. For the analysis involving ASCAT slope, ASCAT data and VODCA CXKu observations have been aggregated to 8-daily values to allow comparison with the results for fAPAR.

### 2.3.3 PBO Network NMRI

We use the Plate Boundary Observatory (PBO) Normalized Microwave Reflection Index (NMRI) dataset (Larson and Small, 2014) for the period 2008 to 2016 to assess temporal VOD dynamics. NMRI is a metric related to VWC, calculated using the interference between direct and reflected GPS signals, transmitted at a frequency of 1.5 GHz (L-band). Daily NMRI is available for over 300 sites in the Western United States and Alaska. The footprint of these measurements covers an area of at least 1000 $m^2$. The PBO sites are mostly installed on grassland and shrubland, while having limited representation in regions with higher biomass. NMRI was already used in vegetation monitoring studies complementary to optical-based products, such as NDVI (Evans et al. (2014), Small et al. (2018)). In addition, phenological parameters derived from PBO NMRI were compared with those obtained from AMSR-E Ku-band VOD, which indicated a broad regional agreement despite the large differences in relative footprint sizes and microwave frequency (Jones et al., 2014). Due to data gaps in the short (nine years) time series caused by freezing conditions and RFI, we aggregate the daily NMRI and VOD to monthly values by taking the median before calculating anomalies. We use LOESS decomposition (Cleveland et al., 1990) to obtain anomalies. We did not use PBO stations within VODCA grid cells with an open water fraction of over 50 %. This resulted in a selection of 296 stations.

### 2.3.4 SAPFLUXNET sapflow

We use in situ sapflow data from SAPFLUXNET (SFN 0.1.5, 10.5281/zenodo.3971689, Poyatos et al. (2020)), the first global database of plant-level sapflow measurements, to assess the temporal patterns of VODCA CXKu in respect to plant transpiration. Sapflow sensors measure the transpiration flow in stems, brunches and trunks as the ascent of sap within xylem tissues (Vandegehuchte and Steppe, 2013). VOD is directly related to leaf water potential (Konings and Gentine, 2017) and also to transpiration as it represents the non-linear response of vegetation to soil drying (Martens et al., 2017). Therefore, we expect a clear connection between VOD and sapflow.

For each site, SAPFLUXNET contains half-hourly tree-level sapflow for different trees, accompanied by tree metadata, site information and hydro-meteorological data. We use the method proposed by Bittencourt et al. (2023) for preprocessing sapflow data from tree level to satellite footprint level. This includes (1) removing the stations that have less than six months of data, (2) filtering out nighttime data (sun altitude < 0°), (3) averaging the data from hourly to daily sapflow for each tree, (4) standardising the daily average sapflow per tree by calculating Z-scores, and (5) averaging trees per site and thus scaling the temporal variability from tree level to site level. Bittencourt et al. (2023) argue that Z-scores remove the differences in absolute values across sites while preserving information on temporal variability. After preprocessing, 98 SAPFLUXNET

stations remain. To analyse if VODCA CXKu manages to capture events of low, median and high transpiration, we extract the VOD data at the SAPFLUXNET sites and take the monthly 5th, 50th and 95th percentile from both VOD and sapflow. Following this step, we standardise the VOD and sapflow percentiles with Z-scores so that the variability in sapflow and transpiration is now in the same range.

### 2.3.5 AGB

We use the dynamic above-ground biomass dataset by Xu et al. (2021) to assess the VODCA L spatial and temporal patterns. Xu AGB provides yearly global maps for the period 2000 - 2019, with a spatial resolution of 10 x 10 km$^2$, derived by applying the method developed by Saatchi et al. (2011) using a consistent set of satellite images. For this purpose, time-series of microwave (QuickSat) and optical (MODIS) satellite imagery are used in a machine learning framework trained against ground inventory plots, airborne lidar, and spaceborne lidar data from the Geoscience Laser Altimeter System onboard the Ice, Cloud, and land Elevation Satellite (ICESat). The maps give AGB estimates in Mg ha$^{-1}$.

Various studies highlighted the sensitivity of VOD to AGB (e.g., Liu et al. (2015), Rodríguez-Fernández et al. (2018), Mialon et al. (2020), Frappart et al. (2020); Schmidt et al. (2023)), emphasizing a stronger agreement of L-band VOD to AGB compared to Ku-, X- and C-bands, especially in densely vegetated areas, such as forests. Recent studies (Qin et al. (2021), Dou et al. (2023)) have shown that the annual 95th percentile of the daily observations is more sensitive to inter-annual biomass change than other aggregating metrics, likely because it manages to minimize the annual changes in the dielectric properties of vegetation caused by water stress. Moreover, Dou et al. (2023) has shown that nighttime observations are more suitable than daytime observations for estimating decadal (10-yearly) biomass carbon dynamics.

To analyse the sensitivity of VODCA L to AGB, we aggregate VODCA L to yearly 95th percentiles, calculated from 10-daily median VOD, to match the AGB temporal resolution and also to reduce the water-related seasonal phenology (Dou et al. (2023)). The Xu AGB maps are resampled to match the VODCA grid by taking the average observations within a 0.25° cell.

### 2.3.6 SMOS-IC VOD

To assess if the spatial change patterns of VODCA L are a result of natural variability we use SMOS-IC VOD v2 (Wigneron et al. (2021),Li et al. (2021), Li et al. (2022), Li et al. (2020)), produced by INRAE (Institut National de Recherche Agronomiques) Bordeaux Soil Moisture and VOD Products and made available at https://ib.remote-sensing.inrae.fr/. SMOS-IC uses the L-band Microwave Emission of the Biosphere (L-MEB) model to derive VOD and Soil Moisture from SMOS Tb (Wigneron et al., 2021). V2 employs an improved optimization process that considers a priori information on VOD over a period of 10 days for each retrieval (Li et al., 2020). SMOS-IC provides daily data in a global EASE Gird (Equivalent Area Scalable Earth) version 2 with a sampling resolution of 25 km.

To analyse the change in spatial patterns of SMOS-IC VOD compared to VODCA L, we resample it to match the VODCA grid using nearest neighbour interpolation and aggregate it to yearly maps.

### 2.3.7 ESA CCI Landcover

We use the ESA CCI Landcover v2 (available at maps.elie.ucl.ac.be/CCI/viewer/) for the epoch 2010 to analyse VOD charac-
265 teristics per land cover (LC) type. The LC map provided at 300 m resolution has been derived by combining MERIS surface
reflectance data acquired during 2008 - 2012 and ground observations (CCI, 2017). We resampled and projected the map onto a
common 0.25° grid using the majority class according to the CCI-LC User Tool (CCI, 2017). We aggregated the LC classes to
bare soil, sparse vegetation, grassland, cropland, shrubs, broadleaf deciduous forest (BDF), broadleaf evergreen forest (BEF),
needle-leaved evergreen forest (NEF), needle-leaved deciduous forest (NDF), and mixed forest (MF) to enable better visual-
270 ization (Figure A1).

## 3 Methods

### 3.1 General framework

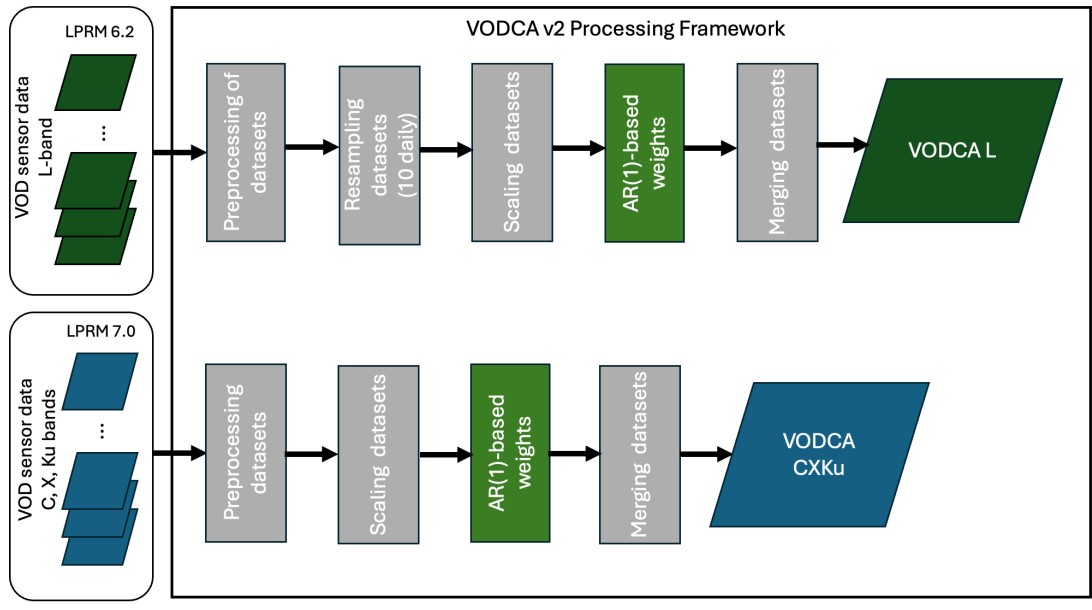

**Figure 2.** Schematic overview of the VODCA v2 processing framework.

The methodology for creating VODCA v2 (and v1) products is based on the methodology for creating harmonised long-term
multi-satellite-based climate data records within the ESA Soil Moisture CCI project (Dorigo et al. (2017); Gruber et al. (2019)).
In the following, we will focus on the VODCA v2 methodology (Fig. 2).

First, the LPRM-derived single-sensor VOD data is preprocessed (section 3.2) and scaled to a chosen reference sensor using cumulative distribution function (CDF) matching to remove the systematic biases between sensors (section 3.3). After scaling, we compute the per-pixel first-order temporal autocorrelation (AC(1)) for each sensor and overlapping period, which we use as an indicator of the random error (section 3.4). Next, we calculate weights based on the AC(1) for each location and overlapping period. These weights are used for fusing the scaled single-sensor observations (section 3.5).

We compute the new products VODCA L and VODCA CXKu independently. VODCA CXKu incorporates only high-frequency observations. We did not include the lower frequency L-band in the multi-frequency product as the latter is more sensitive to the VWC in the woody components of the vegetation layer compared to the higher frequencies (Schmidt et al., 2023). Several studies have already shown that VOD observations from the C-, X- and Ku-band frequencies are highly correlated with each other, except for biomes with little inter-and intra-annual variability (desert and humid tropics)(Moesinger et al. (2020); Wild et al. (2022); Moesinger et al. (2022)). Notably, the work of Moesinger et al. (2022) is very relevant in this respect because it merges C-, X- and Ku band observations to create a standardized vegetation optical depth index (SVODI) and carries out a temporal correlation analysis to show that there is a high agreement between bands (Moesinger et al. (2022), Fig. 1). We provide the results of a spatial (Fig. A3) and temporal correlation (Fig. A2) analysis based on data from the descending overpass of AMSR2 (Jun. 2012 – Dec. 2021), which was used as input in the multi-frequency VODCA CXKu product. Both the temporal and spatial analysis reveal strong and very strong, respectively, agreements between bands. Therefore, from a scientific point of view, it is legitimate to merge them into a single dataset with improved information content, temporal sampling, and reduced noise.

VODCA v2 does not entail single-frequency C-, X- and Ku products for two reasons: First, as mentioned earlier, the AC(1) method employed in the production of VODCA CXKu (section 3.4) is particularly relevant when merging VOD observations from multiple frequencies due to their distinctive sensitivity to noise. Second, in the new scaling framework (section 3.3), we use SSM/I F17 Ku-band as a reference for scaling AMSR2 C-, X- and Ku-band observations. This step, which is needed to bridge the gap between AMSR-E and AMSR2, implies the use of observations from multiple frequencies. Therefore, the single-frequency products would not be completely independent anymore but would rely on Ku-band observations for scaling. As we acknowledge the merit of the C-, X- and Ku-band single-frequency products, we plan to continue their temporal extension with the VODCA v1 framework.

## 3.2 Preprocessing

In VODCA v2, we expanded on the preprocessing methodology described in detail in Moesinger et al. (2020). Similarly to VODCA v1, we :

- Projected the data onto a common, regular $0.25° \times 0.25°$ grid using nearest neighbour resampling;

- Selected the closest nighttime value in a window of $\pm12$ hours for every 0:00 UTC;

- Masked for RFI using flags provided with the LPRM VOD data and based on de Nijs et al. (2015);

- Masked negative VOD retrievals;

- In the case of AMSR2 C-band, we used observations from the 6.9 GHz band if available; otherwise, observations from
the 7.3 GHz band are used instead (if unmasked) to fill gaps;

- In the case of SSM/I, we concatenate VOD data from the sensors F8, F11, and F13 to a single record since they are retrieved from intercalibrated Tb and do not overlap temporally.

Concerning the RFI flagging, we mention that for the C-, X- and Ku observations, the RFI detection uses the estimation of the standard error between two different frequencies de Nijs et al. (2015). For SMOS, since only one frequency is available, we
use the RFI probability information supplied with the SMOS Level 3 data. For SMAP, we use only the internal RFI mitigation supplied with the Level 3 data, because it already uses additional frequencies to filter out RFI contaminated observations (Dorigo et al., 2017).

We also utilized a new flag included in the L3 data from LPRM. This flag pertains to the analytical method for VOD retrieval detailed in Meesters et al. (2005), employing the Microwave Polarisation Difference Index (MPDI) to mitigate temperature-
320 related effects on Tb. This adjustment results in a parameter more closely associated with the dielectric properties of emitting surfaces, as outlined in Owe et al. (2001). Negative MPDI values reflect cases where the horizontal Tb is higher than the vertical Tb and the model does not converge. These valeus are flagged.

Although the flagging of frozen surfaces and snow cover has significantly improved with the new LPRM versions (van der Vliet et al., 2020), we also use ERA5-Land surface soil temperature *stl1*, because it allows for more conservative masking
when applying 275.15 K (2°C) as a threshold.

In order to reduce the random error levels before merging the sensors, we removed outliers using a standard median filter (known as Hampel filter; Pearson (2002)). For each window, the Hampel filter compares each observation with the median absolute deviation (MAD). The observations are considered outliers if they exceed the MAD by a certain number of times. We used the Hampel filter with a window size of 120 days and a threshold of 3 MADs. The window size of 120 days was chosen
to preserve the seasonality and ensure that outliers are identified without being misinterpreted as part of the seasonal trend. A threshold of 3 MADs has been selected to eliminate significant deviations that cannot be explained, given that we are looking at gradual changes in vegetation. We did not choose a lower MADs to prevent excluding valuable data.

CDF-matching reliability depends on the correct representation of the statistical moments of data distribution. Therefore, in the case of VODCA L, we temporally resampled the input sensors SMOS and SMAP to 10-day medians before CDF-
335 matching. This temporal downsampling was necessary for two reasons. First, the original temporal coverage of the SMOS and SMAP sensors is significantly imbalanced in some areas, with SMAP providing a much denser set of observations (Fig. A17) due to different masking strategies employed to both datasets. Although the flags of the SMOS and SMAP products show a similar general spatial pattern, SMOS is flagged more extensively and with more seasonal variation than SMAP, likely due to different thresholds for e.g. topography, vegetation and open water (van der Vliet et al., 2020). The flagging discrepancy
between products can lead to differences in the respective value distributions, making CDF-matching challenging. Second, by

aggregating the datasets into 10-day medians, apart from achieving a more equitable distribution, we obtain much smoother datasets with lower noise levels and outliers, leading to improved CDF-matching.

Following preprocessing, the VOD datasets comprise daily estimates for the C-, X-, and Ku-bands. The L-band includes three estimates each month, specifically on the 1st, 11th, and 21st. These observations represent the first ten days of the month, the subsequent ten days, and the remaining days for that month.

### 3.3 CDF-matching

We use the VODCA v1 CDF-matching method, which combines piecewise linear interpolation with linear least-squares regression (Moesinger et al., 2020). This method provides more robust scaling parameters by fitting a linear model using the sorted observations smaller than the second percentile with an intercept through the second percentile. This way, all the data between the lowest and second-lowest percentiles is used instead of just the lowest value (Moesinger et al., 2020).

Computing VODCA CXKu entails scaling SSM/I, TMI, and WINDSAT on a band-to-band basis to AMSR-E X-band observations. X-band has been chosen as the scaling reference because it exhibits the highest correlation with both C- and Ku-band, as shown in Moesinger et al. (2022) and Wild et al. (2022). The choice of AMSR-E as the reference sensor is motivated by its temporal overlap with these sensors and its superior temporal and spatial resolution, as outlined in the work of Liu et al. (2011). Scaling AMSR2 and GPM observations to AMSR-E is not optimal as there is no temporal overlap. More precisely, VODCA v1 scaled AMSR2 to TMI if enough overlap was available or directly to AMSR-E, without temporal overlap, above and below 35° latitude N and S, respectively (Moesinger et al., 2022). This led to spatial inconsistencies in VODCA v1 (Moesinger et al. (2020), Fig 13 b,c herein). Therefore, we changed the approach in VODCA v2 and used SSMI F17 Ku-band observations (scaled to AMSR-E X-band ) as reference to bridge the gap between AMSR-E and AMSR2. For VODCA L, we use SMAP as reference because it has a better spatio-temporal sampling than SMOS.

### 3.4 Temporal autocorrelation as a measure of random error

Various techniques have been proposed to estimate weights for an optimal merging of satellite data, e.g. for soil moisture, sea surface temperature, and precipitation (Beck et al., 2021). For soil moisture, Gruber et al. (2017) and Kim et al. (2020) proposed merging techniques that make use of the random errors estimated with the triple collocation approach. Kim et al. (2015) maximized the temporal correlation with a reference dataset to obtain weights. To compute weights, these studies use external datasets, e.g., model data (Gruber et al. (2017); Kim et al. (2020)) or reanalysis data (Kim et al., 2015). Unlike soil moisture, VOD is a radiative transfer model parameter rather than a well-defined biogeophysical variable (Li et al., 2021). As no model or independent reference VOD dataset is available, we use autocorrelation instead to quantify the uncertainty of VOD observations.

The use of autocorrelation with a lag of one period (AC(1)) as a measure of random error relies on the assertion that there should be a high degree of temporal AC between subsequent observations since VOD is related to gradual changes in plant water content and biomass (Momen et al. (2017); Konings et al. (2016); Moesinger et al. (2020)). Hence, a lower AC(1) represents a higher random error of a dataset. However, AC(1) is not only sensitive to random or measurement errors but also

to changes in the dynamical stability (resilience) of a system (Boulton et al. (2022); Smith et al. (2022), Smith et al. (2023)).

Therefore, AC(1) can be seen as an error indicator only when we compare measurements of the same area (pixel) over the same time period. Otherwise, when comparing AC(1) values of the same pixel but over different periods, the difference in AC(1) measurements obtained could reflect changes in the dynamic stability in one of the observed periods. Because of that, in the VODCA v2 framework, we compute weights based on AC(1) at each location for each overlapping period between sensors, and we use only collocated observations.

AC(1) means one day in the case of VODCA CXKu and 10 days for VODCA L.

### 3.5  Weighted merging

For a given sensor $s$, pixel $p$, and date $t$, we obtain the weight $w_{s,p,t}$ by scaling its AC(1) value $a_{s,p,t}$ between 0 and 1 using *MinMax* scaling, as shown in equation 1. The AC(1) can take values between -1 and 1. Then, we normalize the $w_{s,p,t}$ values for each pixel, observation date, and available sensor $s$ so that their sum equals 1. As shown in equation 2, we do this by dividing

the $w_{s,p,t}$ for each sensor by the sum of $w_{s,p,t}$ values of the $n$ available sensors at time $t$.

Following the calculation of weights, VODCA L and VODCA CXKu are obtained by multiplying the weights for each observation date, pixel and sensor with the corresponding CDF-matched sensor values $X_{s,p,t}$ as shown in equation 3.

$$w_{s,p,t} = \frac{a_{s,p,t} - \min(a)}{\max(a) - \min(a)}, \ \ \min(a) = -1, \max(a) = 1 \tag{1}$$

$$w_{\mathrm{norm},s=1,p,t} = \frac{w_{s=1,p,t}}{\sum_{s=1}^{n} w_{s,p,t}} \tag{2}$$

$$X_{\mathrm{merged}} = w_{\mathrm{norm},s=1,p,t} \cdot x_{s=1,p,t} + \ldots + w_{\mathrm{norm},s=n,p,t} \cdot x_{s=n,p,t} \tag{3}$$

# 4 Results and discussion

## 4.1 Global patterns

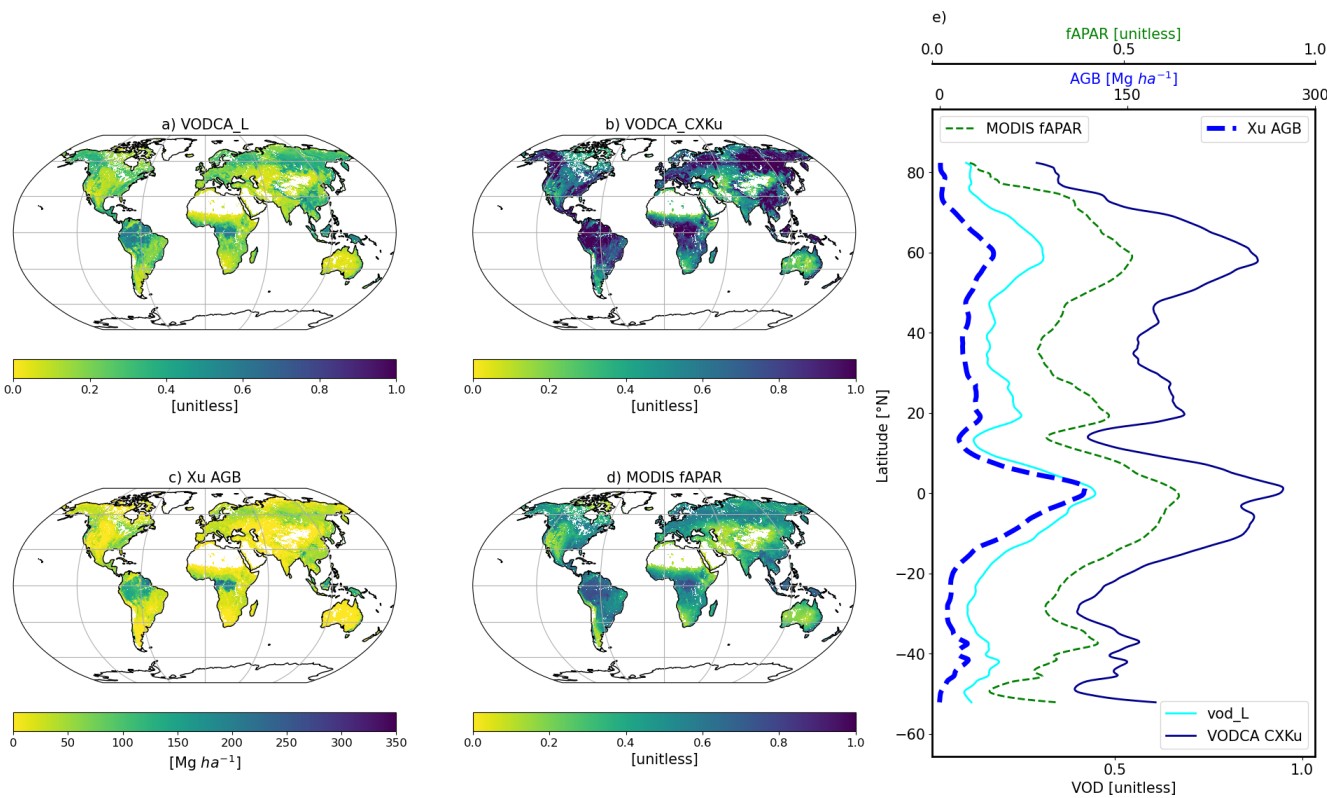

**Figure 3.** Temporally averaged VOD for VODCA L (a) and VODCA CXKu (b). Average Xu AGB (c) and MODIS fAPAR (d). Per latitude average of all four products (e). Temporal averages are computed over the overlapping period (2010 - 2019). We show only the spatial overlap between products.

To evaluate the characteristics of VODCA CXKu and VODCA L, we show in Fig. 3 (a-d) global maps of temporally averaged VOD together with Xu AGB and MODIS fAPAR for the common period (2010 - 2019). In addition, Fig. 3 (e) shows the average VODCA CXKu, VODCA L, AGB and fAPAR per latitude. At the global scale, the VODCA v2 products show similar patterns, with high VOD values in tropical (e.g., Amazon Basin, Congo Basin) and boreal (e.g., Northern Russia, Canada) forests, and low VOD values in arid and sparsely vegetated areas (e.g., Sahara). However, in VODCA L, the relative difference between tropical and boreal forests is much larger than for VODCA CXKu. VODCA L is, on average 37% higher in the tropical forest compared to boreal forests, while VODCA CXKu is only 4% higher (latitudes -5° to 5°: mean VODCA L = 0.40, mean VODCA CXKu = 0.87; latitudes 50° to 60°: mean VODCA L = 0.29, mean VODCA CXKu = 0.84). The figure shows that the spatial distribution of VODCA L is more similar to that of the AGB. These results are also supported by the spatial correlation

analysis (Table A1), which shows that AGB agrees better with VODCA L (Spearman's R: 0.874) than with VODCA CXKu (Spearman's R: 0.800). This confirms the theoretical assumption that L-band VOD is sensitive to the whole vegetation layer including stems, while high-frequency VOD is more sensitive to the upper canopy (Schmidt et al., 2023). Regarding absolute

VOD values, VODCA CXKu is generally higher than VODCA L since the attenuation of microwave radiation increases with frequency due to increasing canopy interference (Moesinger et al., 2020). However, the difference in absolute values between VODCA CXKu and VODCA L is also due to the different parametrisations employed in the retrieval algorithms (LPRM 7.0 vs. LPRM 6.2), especially concerning the single scattering albedo and roughness parameter (Van der Schalie et al. (2017), Dorigo et al. (2017)). The surface roughness parameter is much lower in the LPRM v7 retrievals used by VODCA CXKu,

which automatically leads to higher VOD. In the LPRM v6.2 retrievals used for VODCA L, the larger roughness reduces the VOD values (Dorigo et al., 2023). However, in the following analyses, we focus on the temporal dynamics and the relative spatial patterns of the products. As shown in Fig. A15 (c), VODCA CXKu exhibits the same value range for each landcover class as X-band VOD (Fig. A15 (d)) because the latter is used as reference for the CDF-matching. VODCA L (Fig. A15 (a)) has the same absolute value range as SMAP (Fig. A15 (b)).

We assess the patterns of vegetation variability in VODCA v2 by looking at the per-pixel coefficient of variation (CV) (Fig. 4). The CV gives a measure of the seasonality dynamics, and we computed it by dividing the standard deviation of monthly VOD by the mean VOD. To avoid bias, the pixels with a fractional cover (Fig. A4) of less than 10% of observations in the overlapping period between products and bare soils (Fig. A1) are masked. For comparison, we also provide the monthly MODIS fAPAR CV (Fig. 4 (c)). Additionally, Fig. A16 shows the CV distribution per LC class for all three products. We

observe low values around the equator across all three products, indicating low intra-annual variability caused by stable, all-year-round high vegetation density. In contrast, we observe relatively high CV values in grassland and cropland, which exhibit very strong intra-annual variability. In VODCA CXKu, BDF displays higher CV values than VODCA L, likely due to the upper canopy's stronger seasonal dynamics compared to the vegetation layer's woody components (Li et al., 2021).

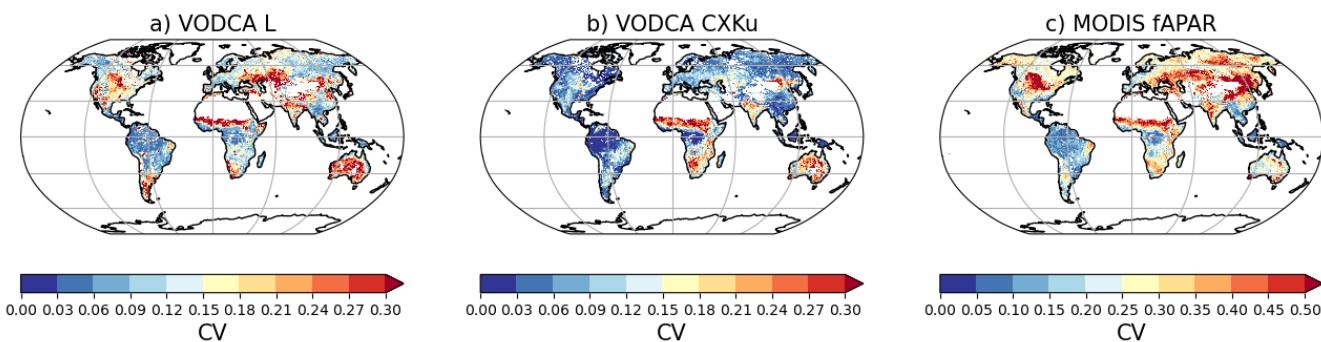

**Figure 4.** Spatial pattern of the CV for VODCA L (a), VODCA CXKu (b) and MODIS fAPAR (c) for the overlapping period (2010 – 2021). The range of the colorbar for (c) is wider because the CV in MODIS fAPAR is higher. We show only the spatial overlap between products.

VODCA CXKu and VODCA L have similar coverage (Figure A4 (a), (b), (c)), with VODCA L providing no data around most of the Sahara and Arabian Peninsula due to CDF-matching failing because of the low number of observations in SMOS VOD (Fig. A17). In Northern Latitudes, the fewer observations are due to the masking of frozen surfaces and snow cover in winter.

## 4.2 Spatio-temporal consistency

To ensure that the merging of multiple sensors and frequencies has not affected the continuity of VOD through time and space, we look at yearly global and hemisphere time-series and at several time-latitude plots at monthly and yearly scales.

The global and hemisphere time-series for VODCA CXKu (Fig. 5) show a clear positive trend, consistent with reports on global greening based on optical satellite sources (e.g., Piao et al. (2020), Chen et al. (2024), Zhang et al. (2017)). The patterns of decrease in 2003 and increase in 2012, although coincident with the introduction of AMSR-E and AMSR2, respectively, can also be observed in MODIS fAPAR (Fig. A6), so we attribute them to natural variability. Although the best intercalibrated SSM/I Tb record available (Berg et al., 2016, 2018) was used to retrieve VOD, we cannot exclude residual bias between F8 and F11 as a cause for the increase in VOD past 1992 due to a lack of credible validation data. To our knowledge, no independent VOD datasets provide data before 1992, while all optical-based vegetation datasets are multi-sensor products with known calibration issues (Brown et al., 2006; Tian et al., 2015).

VOD anomalies from high-frequency observations have been observed to coincide with El Niño-Southern Oscillation (ENSO) variations (Dorigo et al. (2021, 2022), Zotta et al. (2023)), especially in the Southern Hemisphere, where there is a clear connection between ENSO and vegetation activity (Martens et al., 2017). Negative VODCA CXKu anomalies can be observed in El Nino events (e.g., 1998 - 1999, 1991 - 1992), while positive anomalies can be observed in La Nina events (e.g., 2010 - 2011, 2011 - 2012). In VODCA L (Fig. 6), the fluctuations in the time series are minor. While the magnitude of the anomalies is considerably smaller, similar peaks as in VODCA CXKu emerge (e.g., 2012, 2014, 2020).

The seasonal dynamics of monthly VODCA CXKu over time and space (Fig. 7 upper) show consistent patterns with higher VOD in the summer months due to the increase in temperature (in the northern-southern region) or in precipitation (in the subtropics). In VODCA L (Fig. 7 lower), the seasonal patterns are less prevalent, which is to be expected because it also contains information on the woody components of the vegetation layer, which is more constant throughout the year. The seasonality and magnitude of VOD are consistent over time and space in both datasets. Most anomalies in VODCA CXKu and VODCA L (Fig. A5) appear limited in time, and their start and end do not coincide with sensor changes, thus indicating natural variability. Most patterns of negative and positive anomalies in VODCA CXKu are consistent with those of MODIS fAPAR (Fig. A14) and leaf area index (LAI) (Moesinger et al. (2020), Fig. 6). As already mentioned, we cannot exclude residual bias between SSM/I F08 and F11 as a possible cause for the low VOD anomalies before 1992.

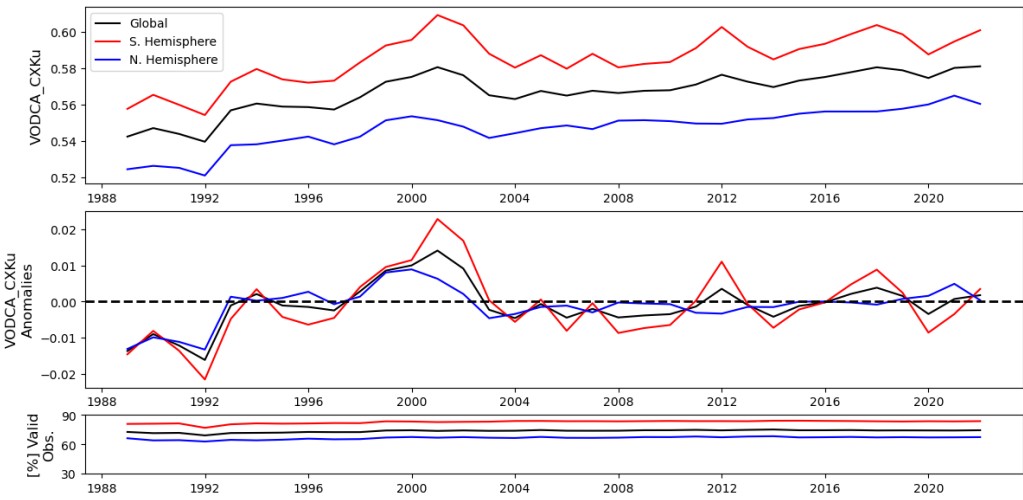

**Figure 5.** Global and hemisphere time-series of yearly VODCA CXKu showing the bulk signal (upper part), detrended anomalies (middle part) and percentage of valid observations (lower part).

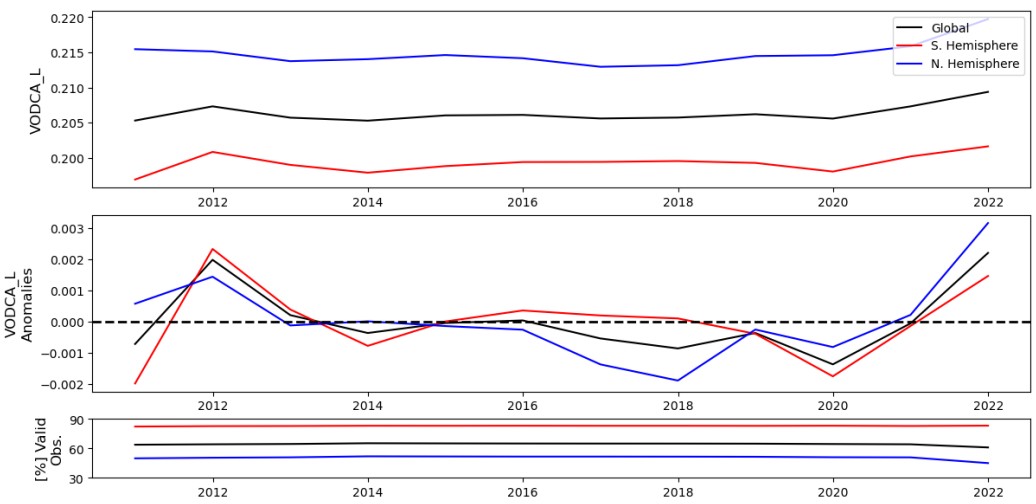

**Figure 6.** Global and hemisphere time-series of yearly VODCA L showing the bulk signal (upper part), detrended anomalies (middle part) and percentage of valid observations (lower part).

The yearly AC(1) appears consistent through time in VODCA CXKu (Fig. 8 upper), with some latitudes experiencing a
slight increase coincident with the introduction of AMSR-E (Jun. 2002) and TMI (Dec. 1997). At the same time, no consistent decreases in AC(1) can be observed, suggesting that no sensor has led to an increase in random error compared to the previous state of the product. In VODCA L (Fig. 8), we see an increase in AC(1) in almost all latitudes coincident with the introduction of SMAP (Mar. 2015). These results suggest that fusing observations in the overlapping period has led to a more robust

product in terms of random error than using only SMOS observations. As a result of this analysis, we reiterate that we expected
to see to some degree a change in AC(1) with the merging of sensors, as VODCA CXKu and VODCA L are harmonized
(through the removal of bias between sensors and fusion of overlapping observations) but not homogenized (forcing same data
characteristics throughout the entire period covered by the merged product). Therefore, it is crucial to consider the influence
of heterogeneous sensor constellation through time for research that delves into higher-order statistics such as variance and
autocorrelation temporally (Smith et al., 2023).

To be noted that in VODCA L, above 60 °N starting 2021, very strong anomalies can be observed that likely cannot be
explained by natural variability. Given that these patterns also coincide with a decrease in AC(1), they are likely either a result
of faulty retrieval or residual RFI.

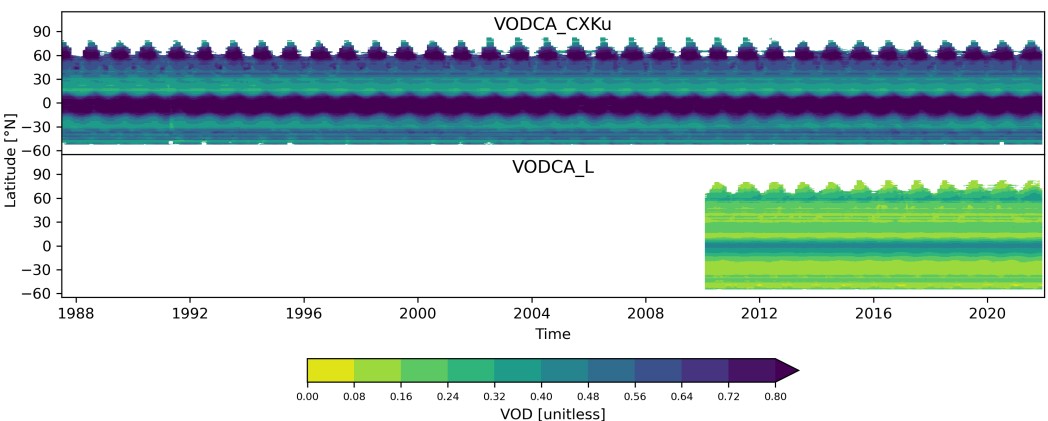

**Figure 7.** Hovmöller diagrams showing the monthly mean VOD per latitude for VODCA CXKu and VODCA L.

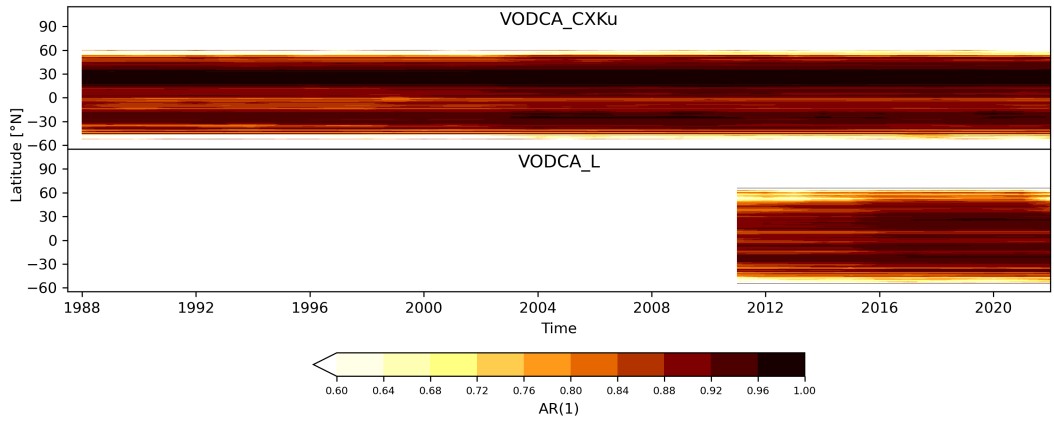

**Figure 8.** Hovmöller diagrams showing the yearly AC(1) per latitude for VODCA CXKu and VODCA L.

### 4.3 Changes in AC(1)

To evaluate the change in random error levels in VODCA CXKu as a result of using data from multiple frequencies, we computed single-sensor frequencies with the same VODCA v2 merging framework. We looked at the change in AC(1) between single- and multi-frequency products using only overlapping observations between products, given that the AC(1) coefficient strongly depends on the temporal resolution (Moesinger et al. (2020)). Almost everywhere, VODCA CXKu exhibits higher AC(1) values than the single-frequency products. A notable exception is the slightly higher AC(1) in VODCA Ku around desert regions such as the Sahara, although it is of a questionable nature since LPRM struggles to retrieve VOD around that area (Moesinger et al., 2020), and around parts of Mexico. The magnitude of AC(1) change seems to decrease with increasing frequency, possibly due to the larger number of sensors used with increasing frequency in the single-frequency products.

Compared to the LPRM SMOS product, VODCA L exhibits much higher AC(1) almost globally (Fig. 9 (d)). In contrast, VODCA L shows both areas with increased and decreased AC(1) compared to LPRM SMAP (Fig. 9 (e)). The areas with decreased AC(1) are primarily arid regions and deserts. A possible reason is the high difference in performance between SMOS and SMAP in these areas, as assessed with AC(1). When the single-sensor datasets are combined, the random noise of SMOS introduces additional random error into the merged dataset, reducing its overall AC(1). The mentioned decrease in AC(1) compared to SMAP was also observed by Moesinger et al. (2020) and is not inherent to the weighted merging procedure. These results indicate that the merging procedure employed for VODCA L is not flawless in all circumstances, and further research needs to be conducted to assess if a global optimum could be achieved. Particularly, testing whether to use only SMAP data in the overlapping period could be an interesting topic for future studies. Nevertheless, VODCA L represents a viable alternative to existing long-term L-band VOD products, providing a longer observation period (2010 - 2021) than SMAP (2015 - 2021) and globally decreased random error levels compared to SMOS.

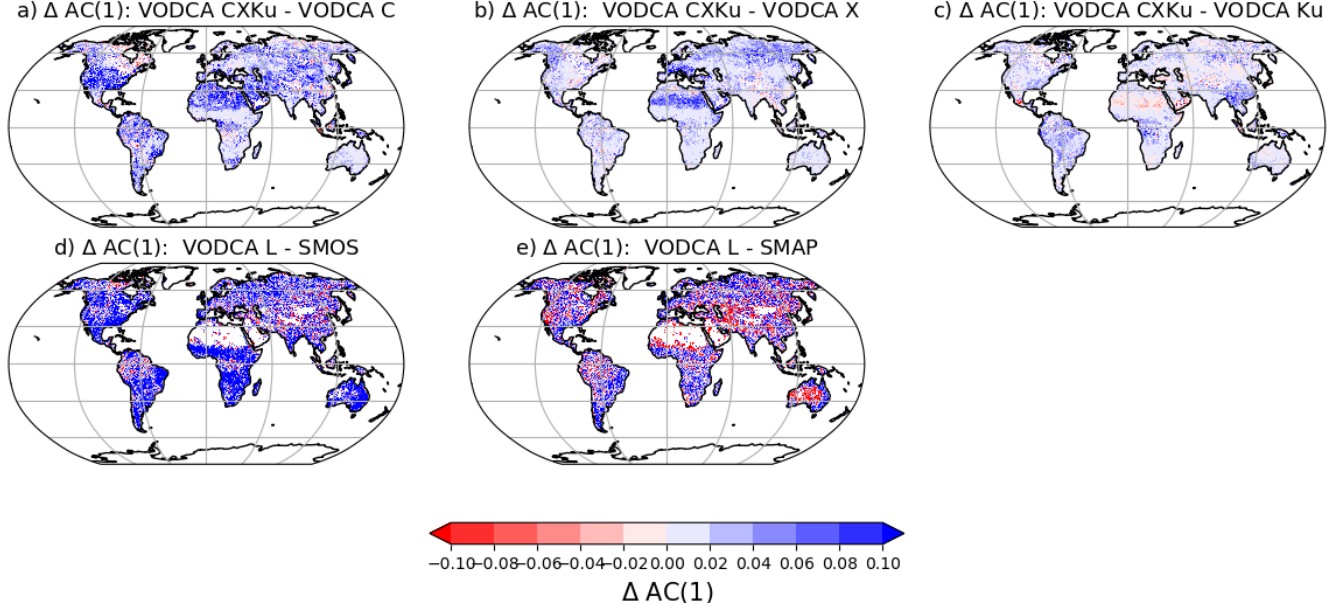

**Figure 9.** Difference in AC(1) between VODCA CXKU and VODCA C (a), VODCA X (b), and VODCA Ku (c). Difference in AC(1) between VODCA L and SMOS (d) and SMAP (e). Only collocated observations in time and space have been used. Blue regions indicate an increase in AC(1) in the VODCA v2 products, while red regions indicate a decrease.

## 4.4 Trends

We conducted a trend analysis to understand how VOD has changed over recent decades and assess the plausibility of change patterns in the products. We employed the Theil-Sen regressor on annual medians and masked slopes with upper and lower confidence intervals having opposite signs. We looked at trends for the complete product period (Fig. 10 (a), (b)), the overlapping period between VODCA L, VODCA CXKU and fAPAR (Fig. 10 (a), (c), (d)) and the overlapping period between VODCA L and Xu AGB (Fig. 10 (e), (f)). As the Xu AGB maps show forest AGB, the analyses in Fig. 10 (e), (f) are limited to locations corresponding to BEF BDF, NEF, NDF and MF land cover classes.

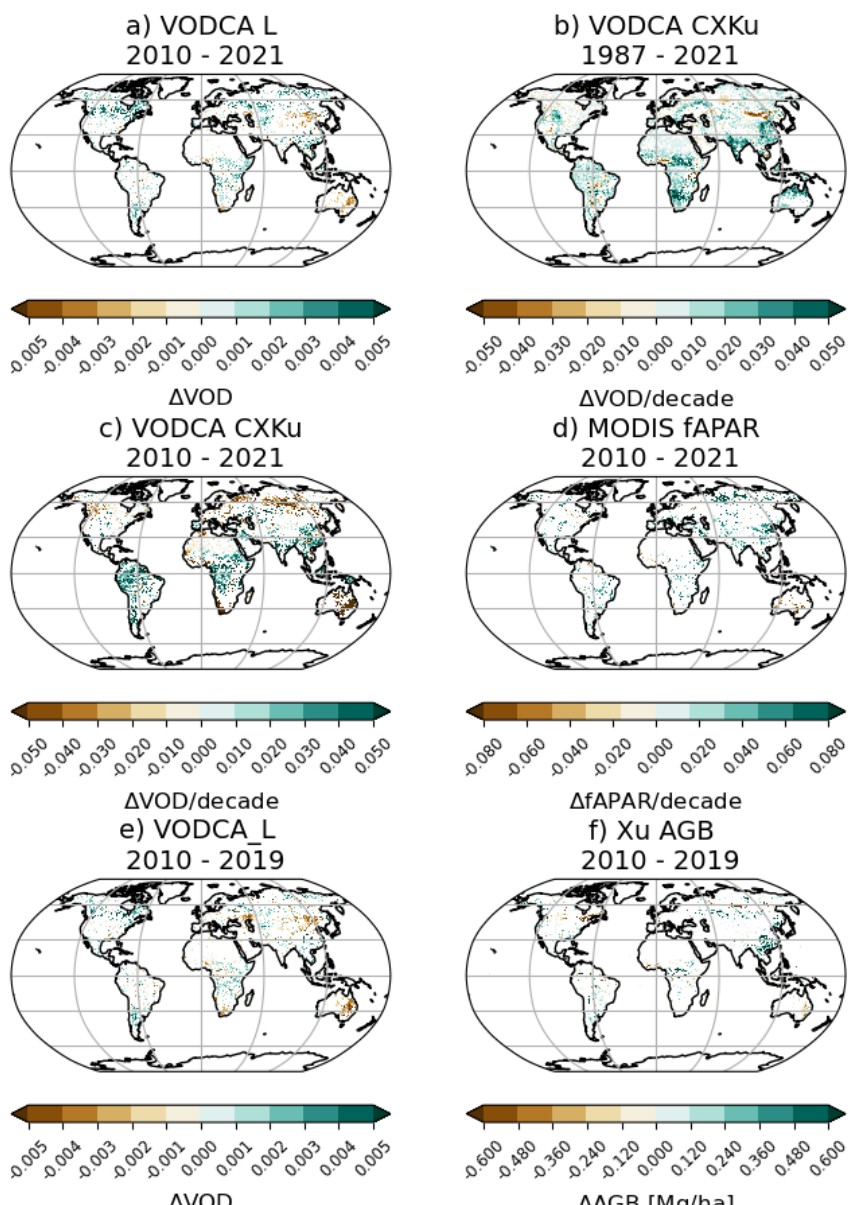

**Figure 10.** Trends over the entire product period calculated with the Theil-Sen regressor for VODCA L (a) and VODCA CXKu (b). Trends for the overlapping period 2010 - 2021 for VODCA CXKu (c) and MODIS fAPAR (d). Trends for the overlapping period 2010 - 2019 for VODCA L (e) and Xu AGB (f). Trends are considered not significant and masked when the upper and lower confidence intervals have conflicting signs.

Due to the short time series of the overlapping period between VODCA L, VODCA CXKu and fAPAR, and between VODCA L and Xu AGB, few pixels show significant trends. However, VODCA CXKu shows a more substantial agreement in

trends with fAPAR (Pearson correlation coefficient of 0.57) than VODCA L (correlation coefficient of 0.33). This is expected since VODCA L is less sensitive to variations in the upper canopy. Similar trends can be observed in VODCA CXKu and fAPAR in regions such as Africa, Australia, China, and India while differing patterns can be seen in boreal forests. We observe consistent VODCA L and AGB trends in regions along the Eastern Coast of Australia, Western Africa, China, Alaska, and Siberia. Some differences emerge in areas like Canada and the Congo Basin.

Some of the trend discrepancies in boreal forests between the VODCA products, fAPAR and AGB, are due to the fact that retrieving VOD in this area is challenging. Various complicating ecosystem properties, such as open water bodies, snow cover, and frozen soil conditions (as mentioned in Vreugdenhil et al. (2016), Vreugdenhil et al. (2020), Li et al. (2021)) make it challenging to retrieve VOD accurately. Additionally, boreal soils have a high organic content, leading to distinct dielectric behaviours (Wigneron et al. (2017), Bousquet et al. (2021)), which are not adequately accounted for in most retrieval algorithms, including LPRM 7.0 and LPRM 6.2, used for VODCA CXKu and VODCA L, respectively.

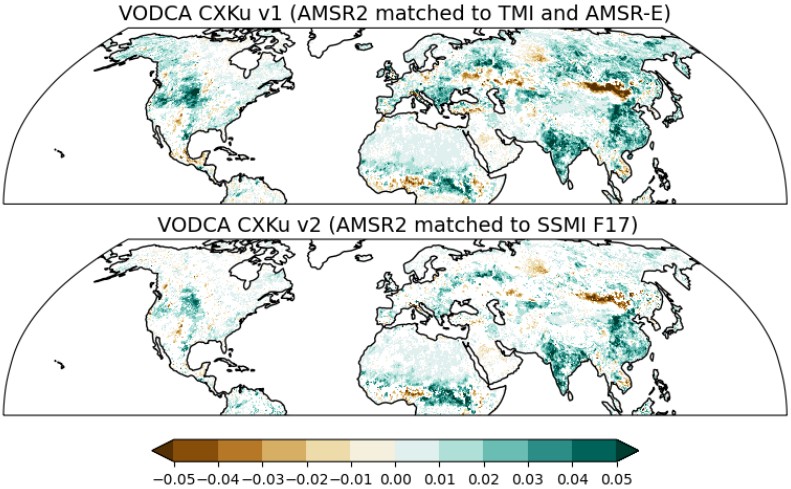

**Figure 11.** Trends (1987 - 2021) for VODCA CXKu v1 computed with the Moesinger et al. (2020) satellite constellation and CDF-matching framework, by merging AMSR2 to TMI below 35°N latitude and AMSR-E further north (top); and for VODCA CXKu v2 computed with the method from this paper, by matching AMSR2 to SSMI F17 (bottom). The same colorbar as in Fig. 10 (b) and (c) was used.

From a technical perspective, the improved merging methodology employed in VODCA CXKu has addressed a spatial break in trends observed in VODCA v1 X and Ku around 35° N latitude in North America (as discussed in Moesinger et al. (2020)). To illustrate this improvement, we computed a version of VODCA CXKu, which uses the same sensor constellation (SSMI without F17, TMI, Windsat, AMSR-E and AMSR2) and CDF-matching of AMSR2 as used in Moesinger et al. (2020) and is hereby refered to as VODCA CXKu v1. As can be observed in Fig. A19, the trends of the two VODCA CXKu products are very similar (Pearson's R 0.84), with differences primarily in magnitude. Matching AMSR2 to TMI below 35°N latitude and to AMSR-E without temporal overlap further north leads to a clear spatial break in North America (Fig. 11 top) and to a consequent positive bias above the 35°N latitude. In VODCA v2, the trends have improved spatially due to the global use of

SMI F17 as reference in the CDF-matching of AMSR2 and GPM (Fig. 11 bottom). To further illustrate the benefit of using SSMI F17 to scale AMSR2 observations, we show in Fig. A13 the time series of both VODCA CXKu v1 and v2 and AMSR2 matched to AMSR-E and SSMI F17, respectively, at a location outside the coverage of TMI. We can clearly observe that scaling AMSR2 to AMSR-E did not produce the desired outcome, which led to bias in the later period of VODCA CXKu v1.

## 4.5 Temporal dynamics of the upper canopy

### 4.5.1 MODIS fAPAR

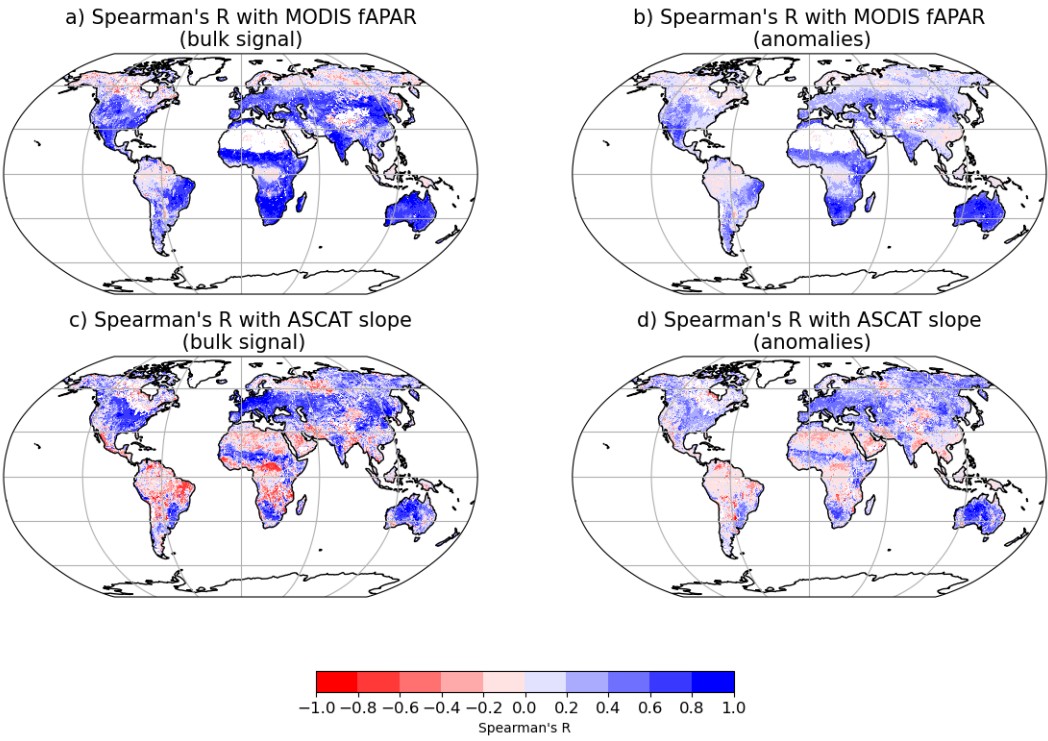

**Figure 12.** Spearman's R between 8-daily VODCA CXKu and MODIS fAPAR over the period 2000 - 2021 on the bulk signal (a) and anomalies (b). Spearman's R between 8-daily VODCA CXKu and ASCAT slope over the period 2007 - 2021 on the bulk signal (c) and anomalies (d).

Generally, considering the bulk signal (Fig. 12 (a)), VODCA CXKu agrees very well with fAPAR over vast areas, e.g. Africa, Australia, Europe, the Indian subcontinent and parts of North America, indicating that the products have similar seasonal dynamics. The best agreement is found in short vegetation types, i.e., grassland (median R = 0.57), cropland (0.56), shrubs 525 (0.59), and in BDF (0.49) (Fig. 13, Table A2). The weakest correlations are observed in NEF, NDF, BEF, and MF. Similar results have been observed when comparing high-frequency VOD products with various other optical vegetation indicators such as LAI (Moesinger et al. (2020), Vreugdenhil et al. (2017)), NDVI (Li et al. (2021), Liu et al. (2011)) and EVI (Li et al.

(2021)). The tropical regions are dominated by weak positive and weak negative correlations, possibly due to low intra-annual variability in VOD (Liu et al. (2011)). Another reason for the reduced correlations in this area could be drought periods during which VOD reduces as a result of lower VWC while fAPAR possibly increases because more radiation reaches the canopy, thus increasing photosynthetic activity (Myneni et al. (2007)). As mentioned earlier, the reduced correlations over the Northern Latitudes could have been caused by artefacts originating from the retrieval algorithm, possibly impacting the VOD seasonality or from differences in phenology between VOD and fAPAR. Regarding phenology, studies found a lag in boreal forests between other optical-based vegetation indicators, such as NDVI and EVI, with VOD-based start-of-season (SOS) preceding that based on optical data (Dannenberg et al. (2020), Jones et al. (2012)). This could be explained by snowmelt, which increases the water supply, and hence sap flow and the water amount in the canopy (Dannenberg et al. (2020)). VODCA CXKu and fAPAR anomalies (Fig. 12 (b)) are positively correlated over wide areas, respecting the same spatial distribution in land cover classes (Fig. A20, Table A2) and regions of strong and weak agreement.

Even though VODCA L is largely sensitive to deeper vegetation layers, we also analyse its agreement with fAPAR (Fig.A21; A23). As expected, the agreement between VODCA L and fAPAR is lower than between VODCA CXKu and fAPAR across all biomes and regions. The best agreement can be found in shrubs, needle-leaf and mixed forests.

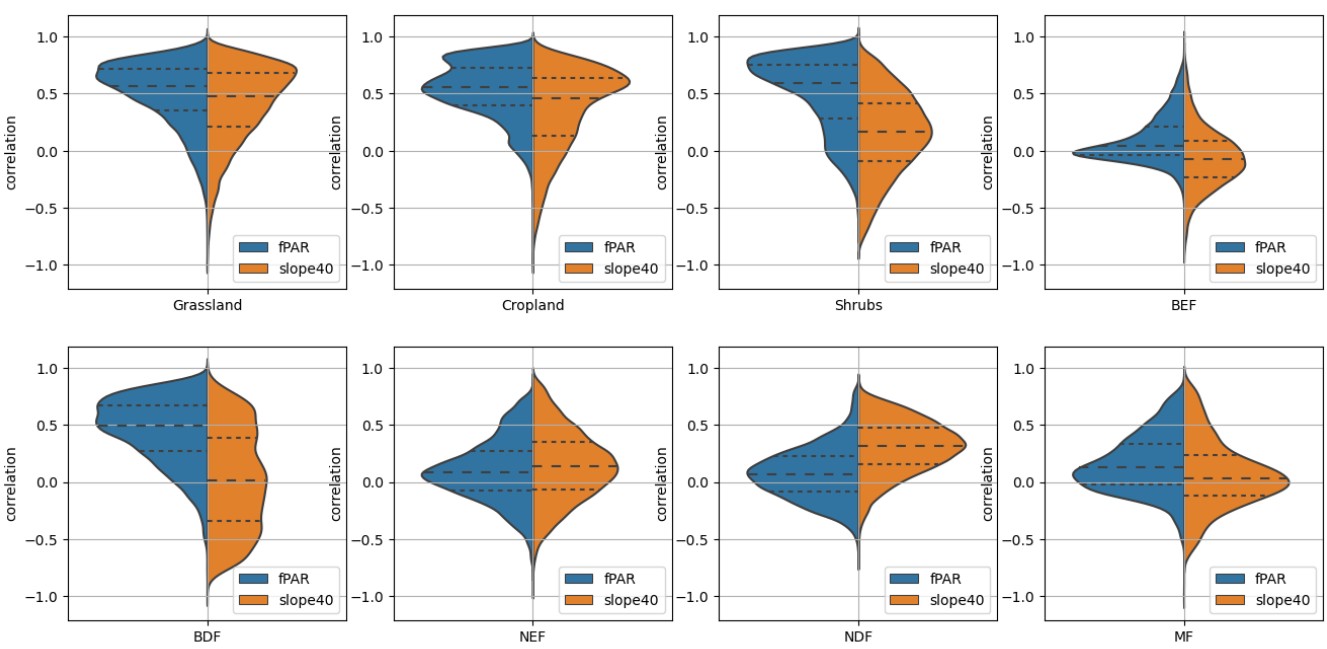

**Figure 13.** Violin plots showing the distribution of Spearman's R between 8-daily VODCA CXKu and MODIS fAPAR (blue) and VODCA CXKu and ASCAT slope (orange) on the bulk signal, per landcover (a-j).

### 4.5.2 ASCAT Slope

Concerning the bulk signal, vast areas with high positive correlations can be observed, e.g., in Europe, Australia, parts of North America, the Sahel region and parts of southern Africa (Fig. 8 (c), (d)). Weak positive-to-negative correlations can be observed, e.g., around the tropical region, Sahara, Arabian Peninsula, parts of Canada, Siberia and Northern Europe. Regarding the tropical regions, as observed in previous studies (Vreugdenhil et al. (2016)), the cause could be the low intra-annual variability for both active and passive data. In the Northern latitudes, the low correlations are due to challenging environmental conditions for the retrieval algorithms (Vreugdenhil et al. (2016), Moesinger et al. (2020)). We can observe positive correlations, especially in grassland (median R = 0.48 bulk signal, R = 0.32 anomalies) and cropland (0.48, 0.32) as shown in Fig. 13, A20 and Table A2. Conversely, low positive to negative correlations occur in BEF, BDF, NEF, MF, bare soil, and shrubs. These results were expected, as several studies reported similar agreement and disagreement patterns between ASCAT-derived vegetation indicators and passive VOD (Vreugdenhil et al., 2016, 2017, 2022). In BDF, the low correlations were attributed to the active signal being dominated by changes in vegetation structure due to an increase in tree foliage during the growing season, with leaves absorbing or forward scattering the signal (Vreugdenhil et al. (2017), Dostálová et al. (2018)). Similarly, the disagreement in the subtropical regions of South America is likely caused by the sensitivity of the ASCAT slope to vegetation structure changes, in pixels with heterogeneous landscape. Over deserts and other arid regions with sparse vegetation (e.g. shrubs), the low agreement with ASCAT data could be explained by the presence of sub-surface soil scattering in the active signal (Hahn et al. (2017); Vreugdenhil et al. (2020); Wagner et al. (2022)). Subsurface scattering occurs during dry conditions when the signal penetrates the sub-surface. This leads to volume scattering and, thus, to a flatter slope. Interestingly, there is good agreement between VODCA CXKu and ASCAT slope in NDF in the Russian Far East over both the bulk signal and the anomalies. As mentioned earlier, surface conditions in these regions are difficult for the retrieval algorithms but may similarly affect active and passive microwave observations. The good agreement in this area with ASCAT slope indicates a decoupling of phenology between optical and microwave-derived vegetation indicators (Fan et al., 2023), meaning that the latter could provide valuable, complementary information to optical information.

As expected, the agreement between VODCA L and ASCAT slope is lower than between VODCA CXKu and fAPAR (Figs A21, A22, A23).

### 4.5.3 PBO NMRI

Generally, there is good agreement between monthly VODCA CXKu and NMRI over the bulk signal (243 stations with significant correlations; Fig. 14 (a)) and anomalies (173 stations with significant correlations; Fig. 14 (b)), with a median Pearson's R of 0.55 and 0.44, respectively. Few stations exhibit weak positive to weak negative correlations, likely due to residual RFI, which is known to affect the C-band and, to a lesser extent, the X-band (de Nijs et al. (2015), Fig. A25) in the USA. We also show VODCA CXKu and NMRI correlation maps for daily observations (Fig. A26 (a)) and time series for the BRUCESPRIN PBO station (Fig. A26 (b)). The agreement is also good at a daily time step, with a median R of 0.40, indicating that VODCA CXKu can capture the daily variations in NMRI. These results confirm the findings of Jones et al.

(2014), showing that passive VOD (VODCA CXKu) follows the variations in NMRI, despite the huge difference in footprint size.

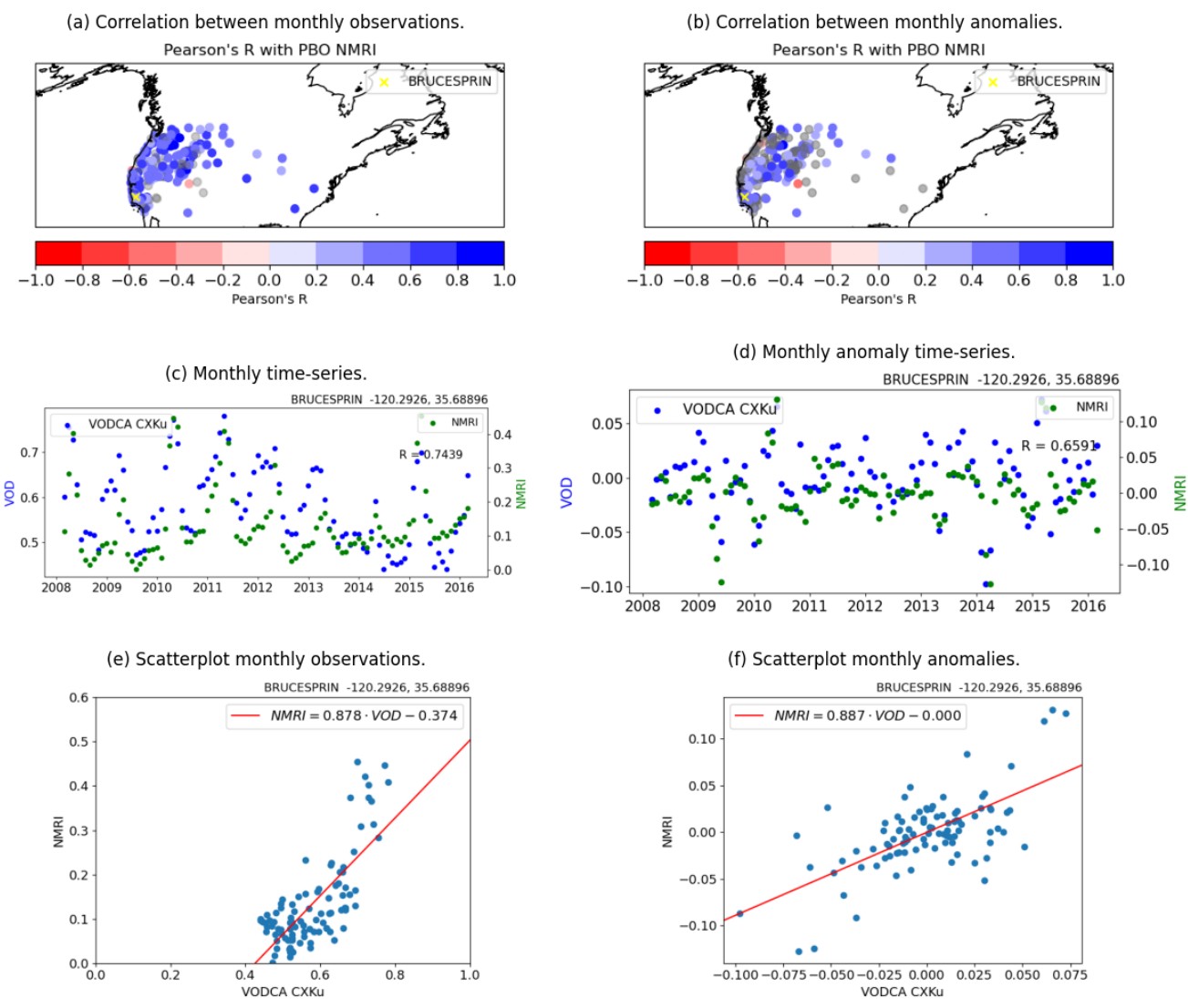

**Figure 14.** Pearson's R between monthly VODCA CXKu and NMRI for the PBO stations over the bulk signal (a) and anomalies (b). The stations with non-significant correlations are shown in grey. time series over the bulk signal (c) and anomalies (d) of VODCA CXKu (blue) and PBO NMRI (green), as well as scatterplot over the bulk signal (e) and over anomalies (f) for a station with good agreement between datasets (BRUCESPRIN, lon -120.29, lat 35.68, indicated with a cross in the correlation maps).

The agreement between NMRI and VODCA L is weaker both in terms of bulk signal (121 stations with significant correlations; Fig. 15 (a)) and anomalies (40 stations with significant correlations; Fig. 15 (b)), with a median Pearson's R of 0.36

and 0.28, respectively. This outcome is somewhat surprising, considering that NMRI measurements are derived from L-band GPS signals. One possible explanation for the much better agreement of NMRI with VODCA CXKu could be the sensitivity to green vegetation, which is known to increase with higher frequencies (Li et al. (2021)), especially given the PBO sites' locations in areas with shorter vegetation, such as grasslands and shrublands.

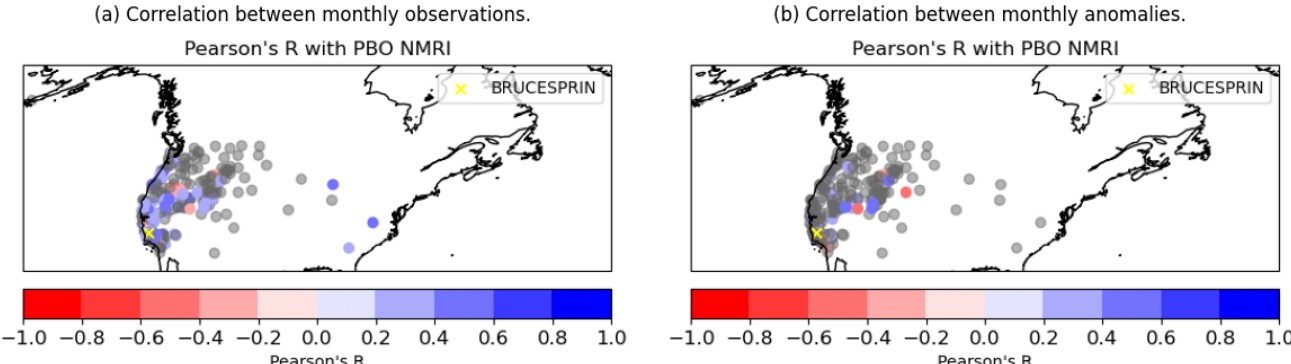

**Figure 15.** Pearson's R between monthly VODCA L and NMRI for the PBO stations over the bulk signal (a) and anomalies (b). The stations with non-significant correlation are shown in grey.

### 4.5.4 SAPFLUXNET

We examined how sensitive VODCA CXKu is to sapflow on days with low, normal and high transpiration rates in the scatterplots from Fig. 16 (a) and by using the Pearson and Spearman regression coefficients. We only considered stations with a significant correlation between VODCA CXKu and sapflow (Fig. 16 (b)). We observed positive correlations between VODCA CXKu and sapflow during normal (Spearman's R: 0.52, 51 stations with significant correlations; Pearson's R 0.48, 50 stations with significant correlations) and high (Spearman's R: 0.48, 53 stations with significant correlations; Pearson's R 0.43, 54 stations with significant correlations) transpiration days. The agreement between VODCA CXKu and sapflow during days with low transpiration (5th percentile) is weaker (Spearman's R: 0.39, 37 stations with significant correlations; Pearson's R: 0.35, 27 stations with significant correlations). The lower correlations and the lower number of stations with significant correlations indicate a weaker linear relationship between VOD and sapflow, compared to days with normal and high transpiration rates. Generally, the high amount of stations where the correlation between VODCA CXKu and sapflow is non-significant is likely caused by the short temporal overlap between datasets, with an average of 21 observations, compared to 31 observations on average for the stations where a significant relationship has been found.

Fig. 16 (b) and Fig. A24 show the spatial distribution of the SAPFLUXNET stations used in this analysis and the agreement between VODCA CXKu and sapflow for each station. The highest correlations can be observed for days with normal transpiration, with a median Spearman's R of 0.67 followed by high (median Spearman's R: 0.64) and low (median Spearman's R: 0.57) transpiration days (Fig. A24). In contrast, the agreement is weaker on days with low transpiration rates. Compared to the

global analysis (as performed in Fig. 16 (a)), the higher correlations on a station basis indicate that the complex, non-linear relationship between VOD and sapflow is better described when modelled at each location.

(a)

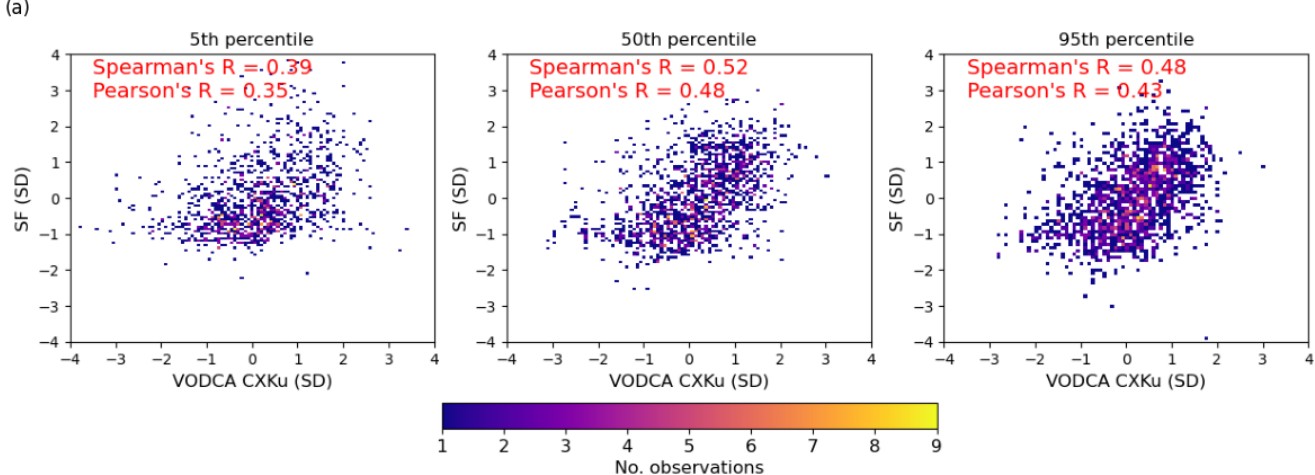

(b)

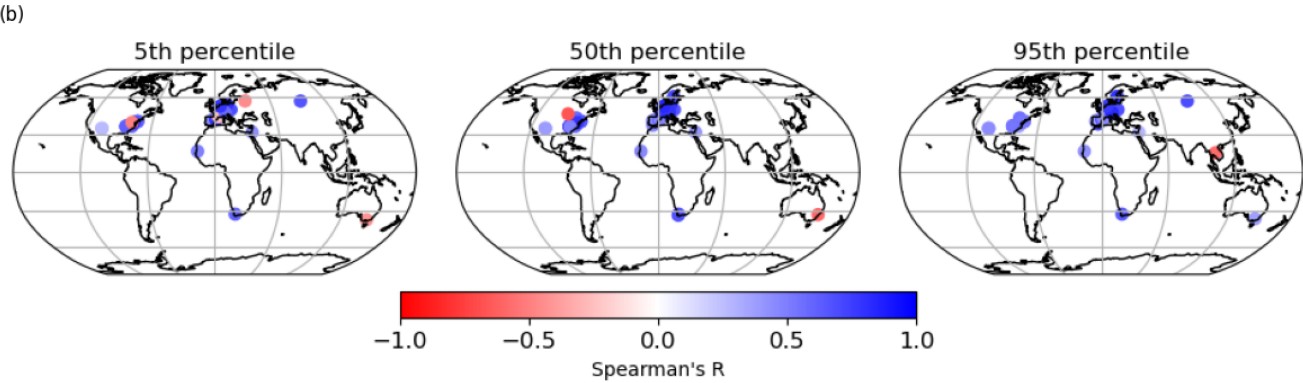

**Figure 16.** (a) Density scatterplots outlining the relationship between VODCA CXKu and sapflow (SF) standard deviations (SD) for the 5th, 50th and 95th percentiles of monthly data, for the stations with significant correlations. The colorbar shows the number of observations in each bin. (b) Maps showing the agreement between SF (SD) and VODCA CXKu (SD) for each SAPFLUXNET station, for 5th, 50th and 95th percentiles of monthly data. Stations with non-significant correlations are masked ($p > 0.05$).

## 4.6  AGB

Generally, there is a very good yearly spatial agreement between VODCA L and Xu AGB, with Spearman's R values around 0.86 for each year. The relationship between VODCA L and Xu AGB follows a logistic function (Fig. A27) as found in earlier

studies investigating the connection between L-band VOD and AGB maps (Rodríguez-Fernández et al. (2018), Mialon et al. (2020)).

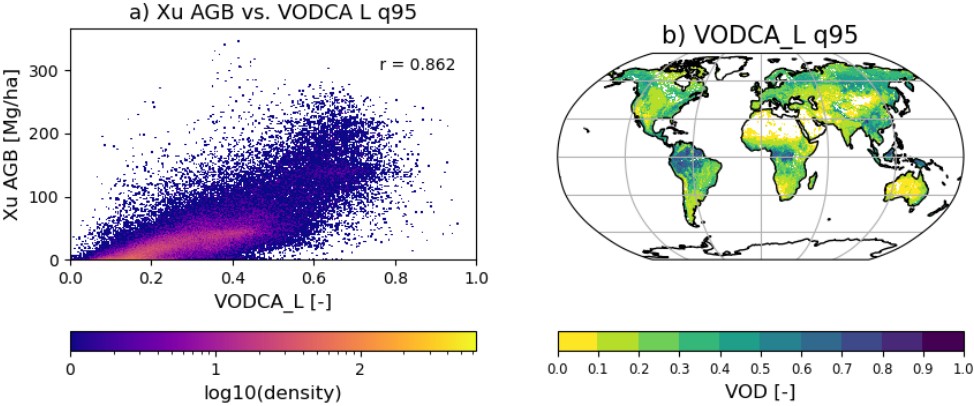

**Figure 17.** a) Temporal-spatial relationship between VODCA L q95 and Xu AGB over all overlapping years. The scatterplot is colored by the density of observations. Spearman's r is shown in the upper right side. (b) Median of VODCA L q95 of each year in 2010 - 2019.

Regarding the temporal dynamics, our analysis (Fig. 18) is limited to the availability of only annual Xu AGB observations. We calculate the difference in VODCA L, SMOS-IC VOD and AGB between 2011 and 2019, similar to Araza et al. (2023). Araza et al. (2023) investigates the change in AGB for forested areas based on four multi-date AGB maps: Xu AGB, SMOS-

derived AGB (Wigneron et al. (2021)), ESA CCI AGB (Santoro and Cartus (2021)) and a carbon flux model Harris et al. (2021). In our analysis, we are more interested in the agreement between positive and negative change patterns rather than the absolute change. Patterns of decreasing VODCA L agree with decreases in Xu AGB and the maps presented in Araza et al. (2023), e.g. for the Siberian boreal forest, west and central Africa, southwestern Amazon and the East coast of Australia. These are either well-known deforestation hotspots (Song et al. (2018), Feng et al. (2022)) or areas that were affected by

severe wildfires in the past decade according to the Global Fire Atlas (Andela et al. (2019)). Patterns of increasing VODCA L coincide with increases in AGB around China, western Canada, and scattered locations across Europe and Asia. Some of these locations have been subject to reforestation efforts in the last decade (Song et al. (2018)). The comparison with Xu AGB also reveals areas with mismatching change patterns, such as the North American boreal forests and the Amazon basin. However, the distribution of positive and negative change patterns in VODCA L is similar to that of SMOS-IC VOD, including in the

areas with no agreement with Xu AGB. Therefore, the dissimilar patterns are not caused by artefacts originating from the VODCA L merging framework. Differences between VODCA L and SMOS-IC VOD can be observed mainly in magnitude, which makes sense since VODCA L uses a different retrieval algorithm and includes observations from SMAP.

To enable comparison, we provide the same type of analysis in Fig. A28 for VODCA CXKu. As expected, since VODCA CXKu entails information largely on the upper canopy dynamics, trends and change patterns are more dissimilar to those

observed for Xu AGB, especially in boreal and tropical forests.

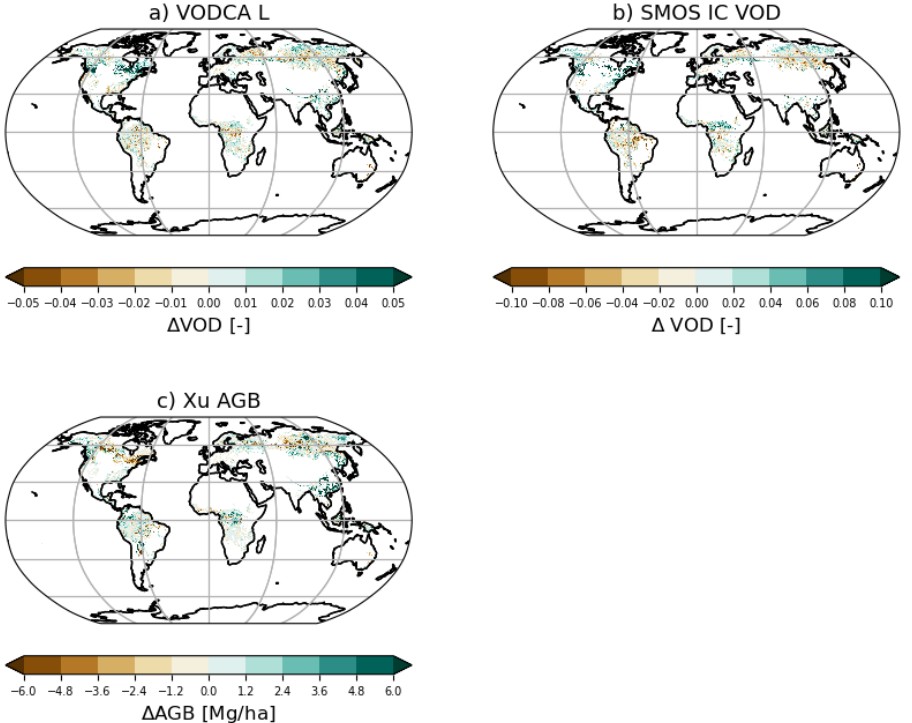

**Figure 18.** Difference between the years 2019 and 2011 for a) VODCA L and b) SMOS-IC VOD and c) Xu AGB (method of Araza et al. (2023) for reducing the inter-annual variability of the original time-series). The analysis is limited to forested areas.

## 5 Conclusion and outlook

In VODCA v2, two new VOD datasets were introduced: VODCA CXKu, a daily multi-frequency product spanning 1987 to 2021, and VODCA L, a 10-daily L-band product covering the period 2010-2021. The datasets were computed by scaling data from each satellite mission and frequency band to a reference mission and band (SMAP for VODCA L and AMSR-E X-band for VODCA CXKu) using CDF matching. We made several changes compared to VODCA v1. First, instead of three single high-frequency products, we provide VODCA CXKu, which uses observations from the C-, X-, and Ku-band frequencies and are indicative of VWC dynamics in the upper canopy. Combining the high-frequency bands maximises the information contained while increasing the number of daily observations and reducing noise. Second, we merged VOD estimates using a novel weighted merging method that relies on temporal first-order autocorrelation (AC(1)) to compute weights for locations and timesteps where multiple observations are available. The assumption is that VOD, linked to gradual changes in vegetation water content, should exhibit a high degree of temporal AC(1) between subsequent observations, while sudden changes indicate noise. Third, VODCA v2 CXKu includes two additional high-frequency sensors, SSM/I F17 and GPM GMI. Specifically, SSM/I F17 addresses the gap between AMSR-E and AMSR2, which in VODCA v1 led to trend breaks above 35°N and below 35°S. We demonstrated that due to these methodological improvements, VODCA CXKu exhibits lower random error levels (higher

AC(1)) than the single-frequency products. VODCA L shows higher AC(1) compared to LPRM-derived SMOS globally and areas with increased and decreased AC(1) compared to LPRM-derived SMAP. Therefore, VODCA L has the advantage of a longer time series compared to SMAP and lower random error levels compared to SMOS. The areas with decreased AC(1) are primarily arid regions and deserts. We showed that the spatial trend patterns have been significantly improved in VODCA CXKu due to using SSM/I F17.

In summary, our validation results show the following:

– VODCA CXKu and MODIS fAPAR demonstrate similar temporal patterns in various regions, especially for short vegetation types and broadleaf forests, as observed in literature with other optical vegetation indicators. Artefacts from the retrieval algorithm are likely causing dissimilarities in boreal forests, while in tropical regions, the lack of agreement is due to the minimal intra-annual variability in VOD.

– VODCA CXKu agrees with ASCAT slope, particularly in cropland, grassland and needle-leaf deciduous forest. However, weak correlations were found in needle-leaf evergreen and broadleaf forests, consistent with prior studies.

– VODCA CXKu aligns well with in situ NMRI data at daily and monthly aggregations. Even though NMRI uses L-band GPS observations, the agreement with VODCA L is weak. This is likely because VODCA L is predominantly sensitive to woody biomass, while the NMRI stations are mainly located in grassland and cropland. Also, the difference in footprint
size may play a role.

– Preliminary findings suggest that VODCA CXKu captures transpiration patterns, especially on days with medium and high transpiration, but further research is needed to disentangle the VOD-sapflow connection.

– Yearly estimates of VODCA L correspond closely with yearly Xu AGB maps, with their relationship being described by a logistic function. Trends and changes in VODCA L exhibit similar patterns to Xu AGB and with previous studies in
deforestation, reforestation, and wildfire hotspots. Further research is required to analyse sub-yearly patterns.

– VODCA CXKu and VODCA L show mostly consistent patterns through time and space, unaffected by the fusion of multiple sensors and frequencies. However, in VODCA CXKu, we cannot exclude residual bias between SSM/I F8 and F11, even though the best available intercalibrated brightness temperature datasets have been used to retrieve VOD.

Based on our findings, we conclude that VODCA CXKu provides useful complementary information to optical vegetation
indicators to study the vegetation canopy response to climate variability and anthropogenic impacts. We suggest using it for long-term vegetation monitoring studies, focusing on short vegetation types and broadleaf forests. We recommend that users consider the possibility of residual bias between data before 1992 and after. Nevertheless, for research that delves into higher-order statistics such as variance and autocorrelation temporally, it is crucial to consider the influence of the heterogeneous sensor constellation through time, as these statistics may also be sensitive to the overall noise levels, which vary over time.
VODCA L provides valuable insight into biomass and biomass change but further research is needed to determine its suitability for intra-annual AGB monitoring. Given that our methodology for creating VODCA L is not flawless in all circumstances (e.g., arid regions), future studies should explore alternative methods for merging SMOS and SMAP.

*Data availability.* VODCA v2 is open access and available at: https://doi.org/10.48436/t74ty-tcx62

(Zotta et al., 2024).

**Appendix A: Appendix**

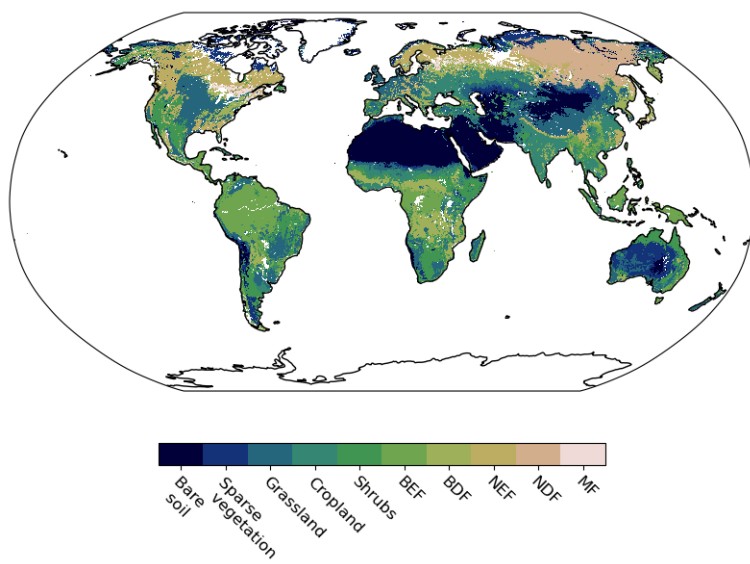

**Figure A1.** ESA CCI Landcover map v2 (maps.elie.ucl.ac.be/CCI/viewer/) for 2010, aggregated to major classes. Locations containing open water have been masked.

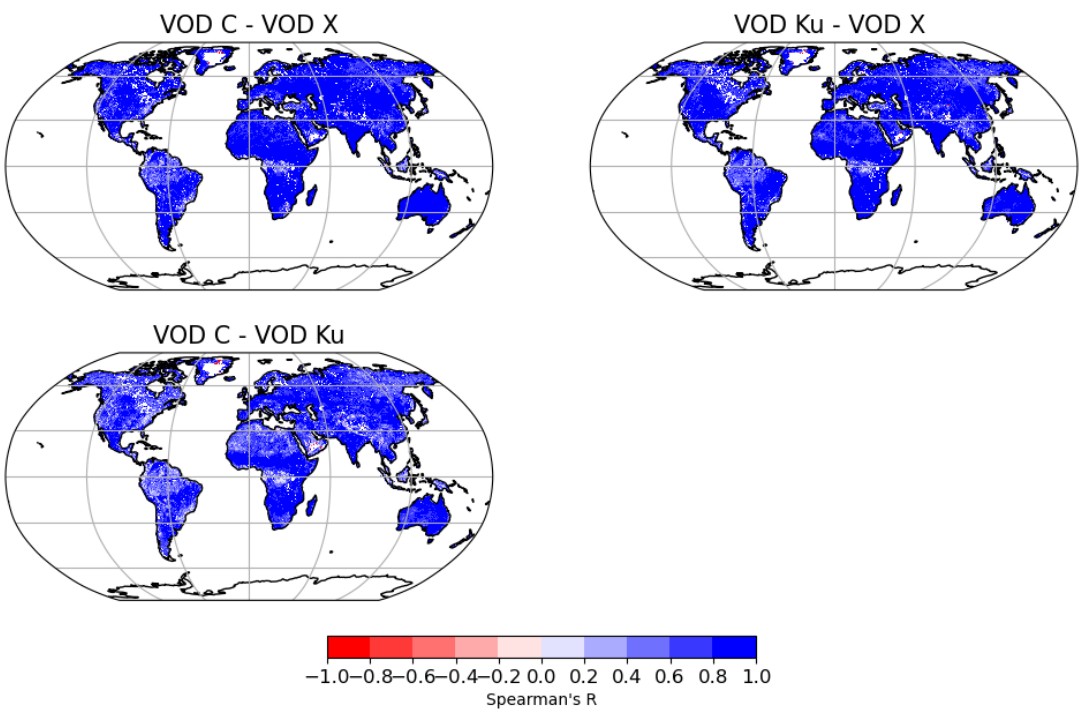

**Figure A2.** Spearman's R between AMSR2 (2012 - 2021) VOD C, VOD X and VOD Ku data. The correlations are based on daily data.

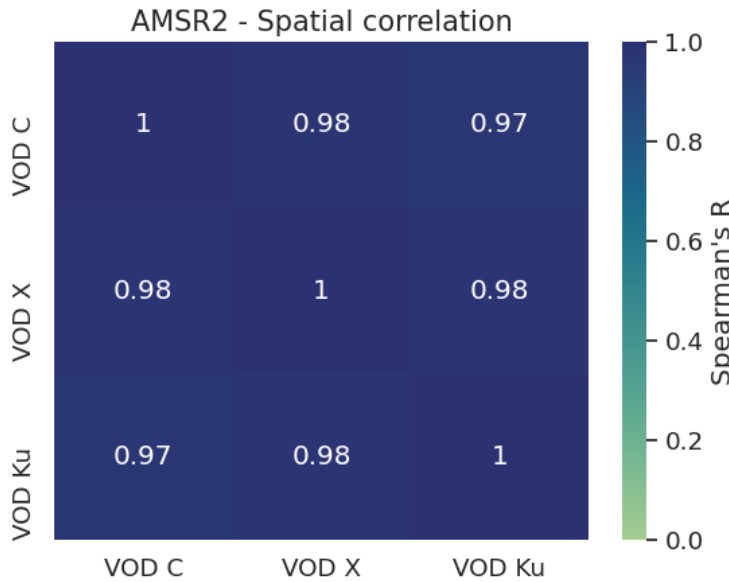

**Figure A3.** Spatial correlation (Spearman's R) between average AMSR2 (2012 - 2021) VOD C, VOD X and VOD Ku data.

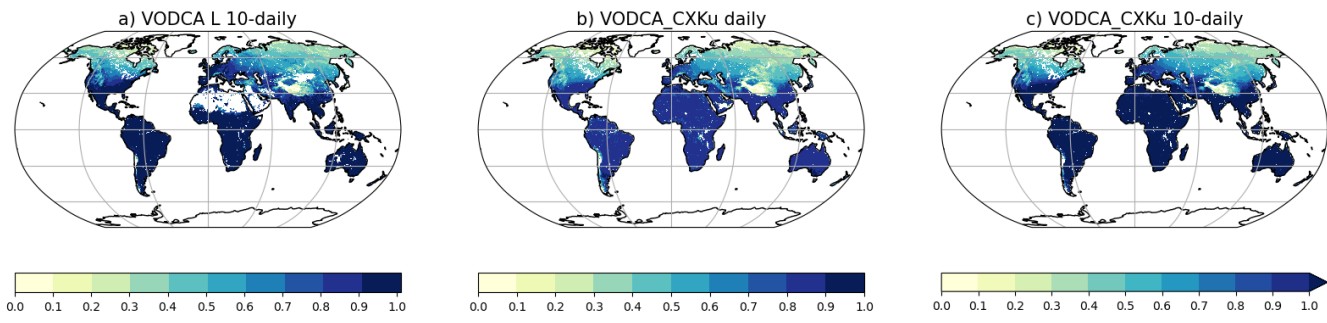

**Figure A4.** Fractional coverage of the VODCA v2 products, expressed as the total number of available observations divided by the total number of possible observations in the overlapping period January 2010 - December 2021. Pixels that have a fractional cover of exactly 0 are shown in white.

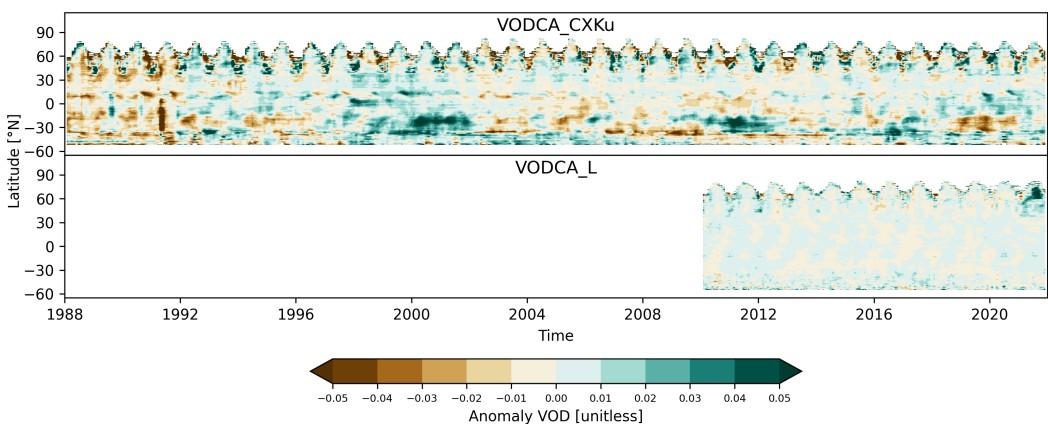

**Figure A5.** Hovmöller diagrams showing anomalies of the monthly means per latitude for VODCA CXKu and VODCA L. Anomalies have been computed as deviations from the climatology of the periods 1990 - 2020 (VODCA CXKu) and 2010 - 2021 (VODCA L).

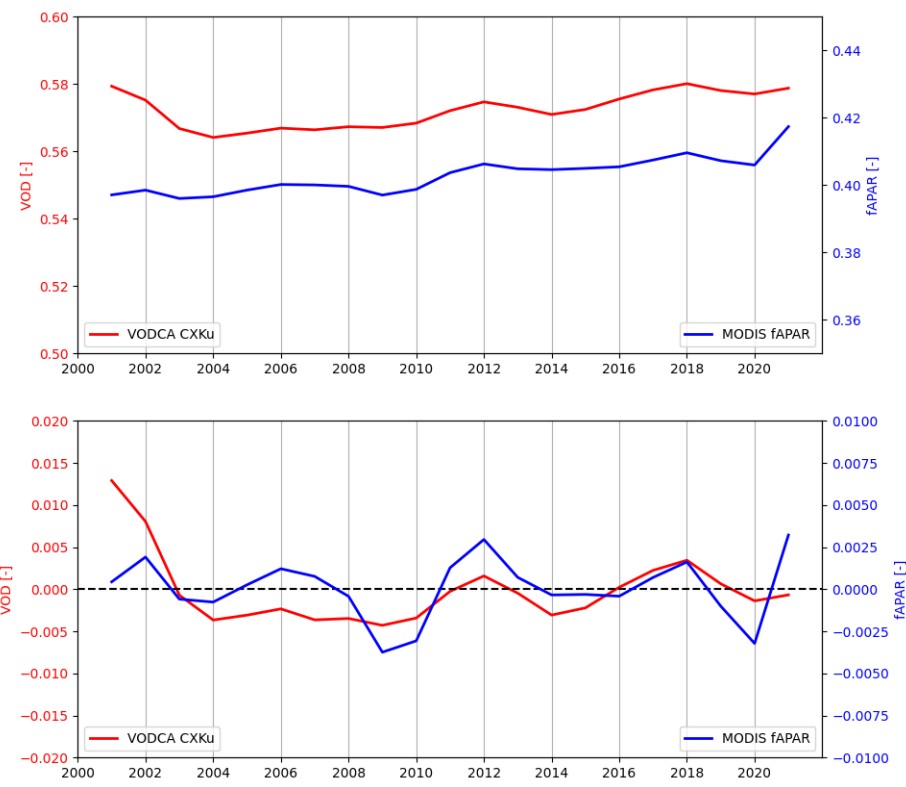

**Figure A6.** Yearly global time-series of VODCA CXKu and fAPAR for the bulk signal (upper graphic) and for anomalies (lower graphic).

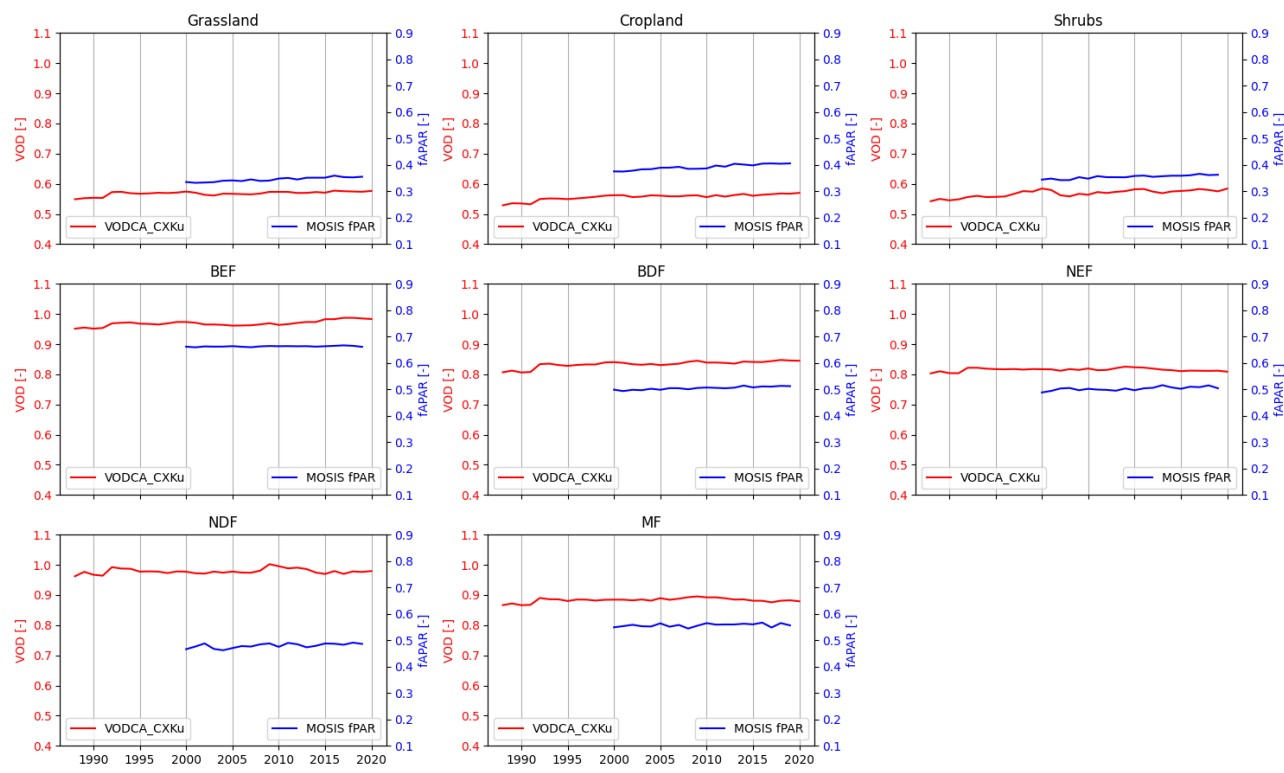

**Figure A7.** Yearly time-series per landcover class for VODCA CXKu and MODIS fAPAR. ESA CCI LC was used.

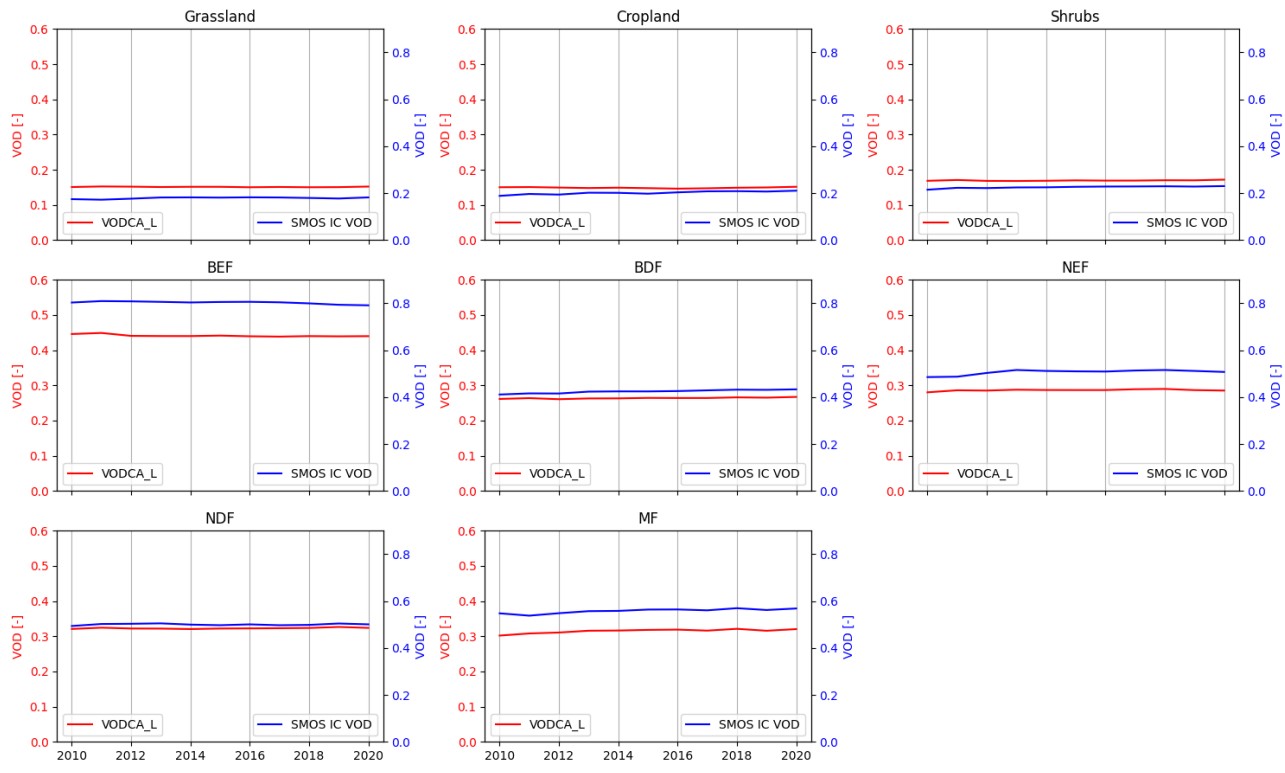

**Figure A8.** Yearly time-series per landcover class for VODCA L and SMOS IC VOD. ESA CCI LC was used.

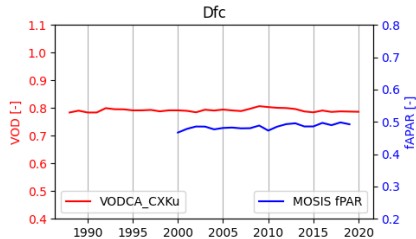

**Figure A9.** Yearly time-series for the Dfc climate, corresponding to boreal forest, for VODCA CXKu and MODIS fAPAR.

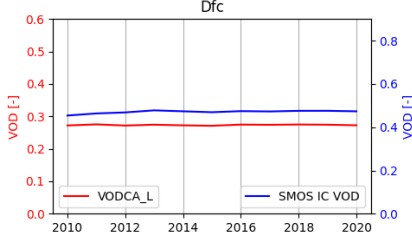

**Figure A10.** Yearly time-series for the Dfc climate, corresponding to boreal forest, for VODCA L and SMOS IC VOD.

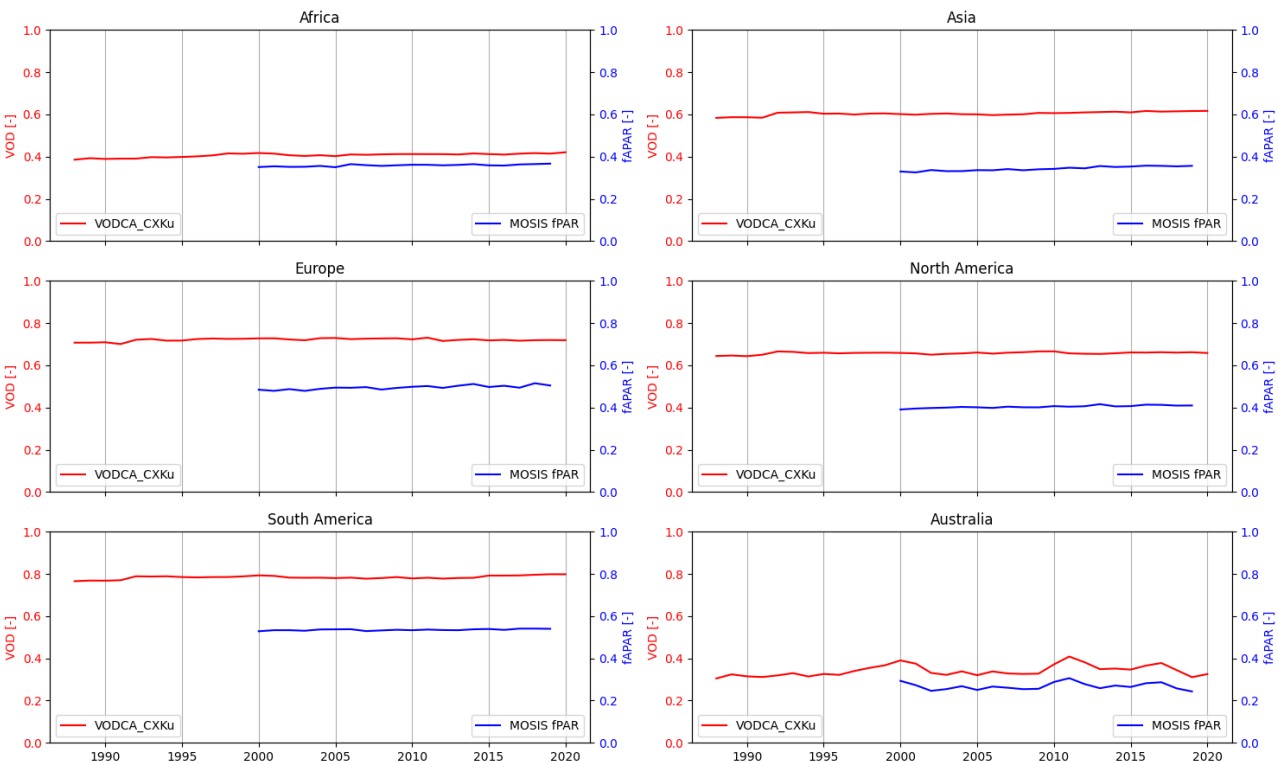

**Figure A11.** Yearly time-series per continent for VODCA CXKu and MODIS fAPAR.

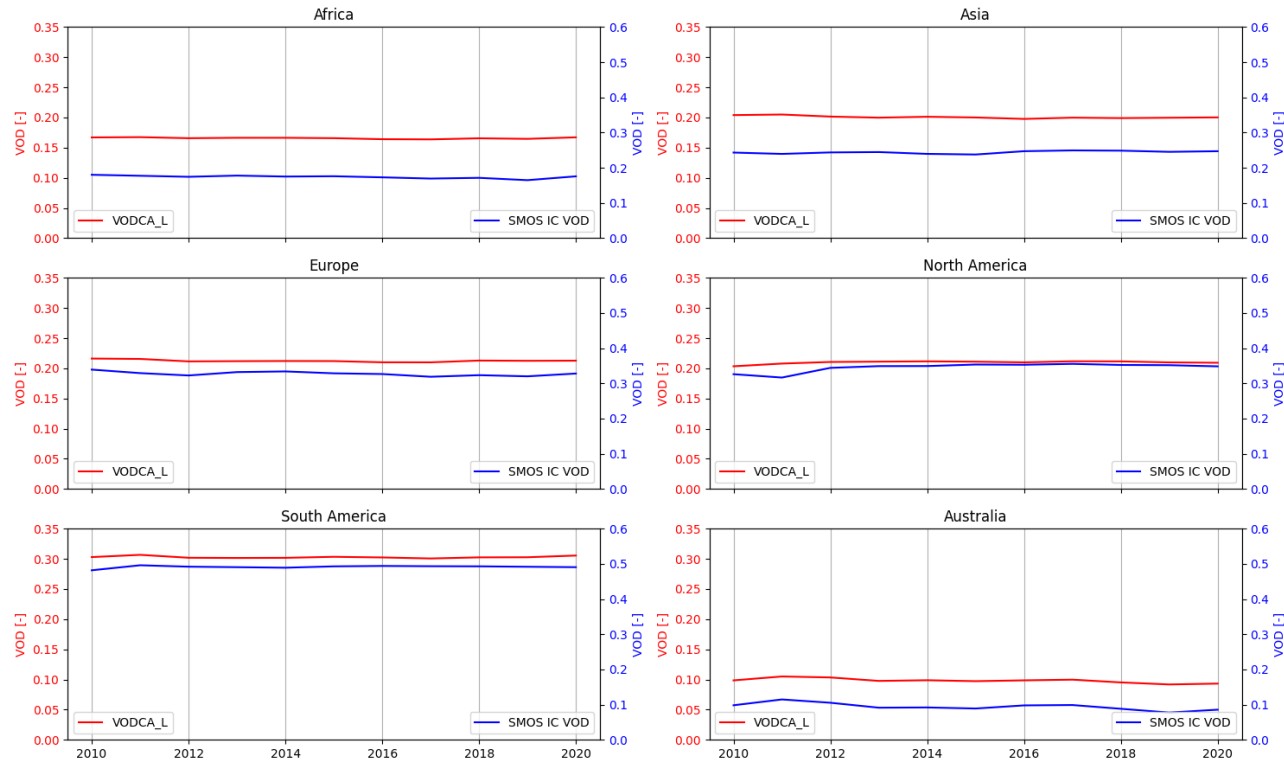

**Figure A12.** Yearly time-series per continent for VODCA L and SMOS IC VOD.

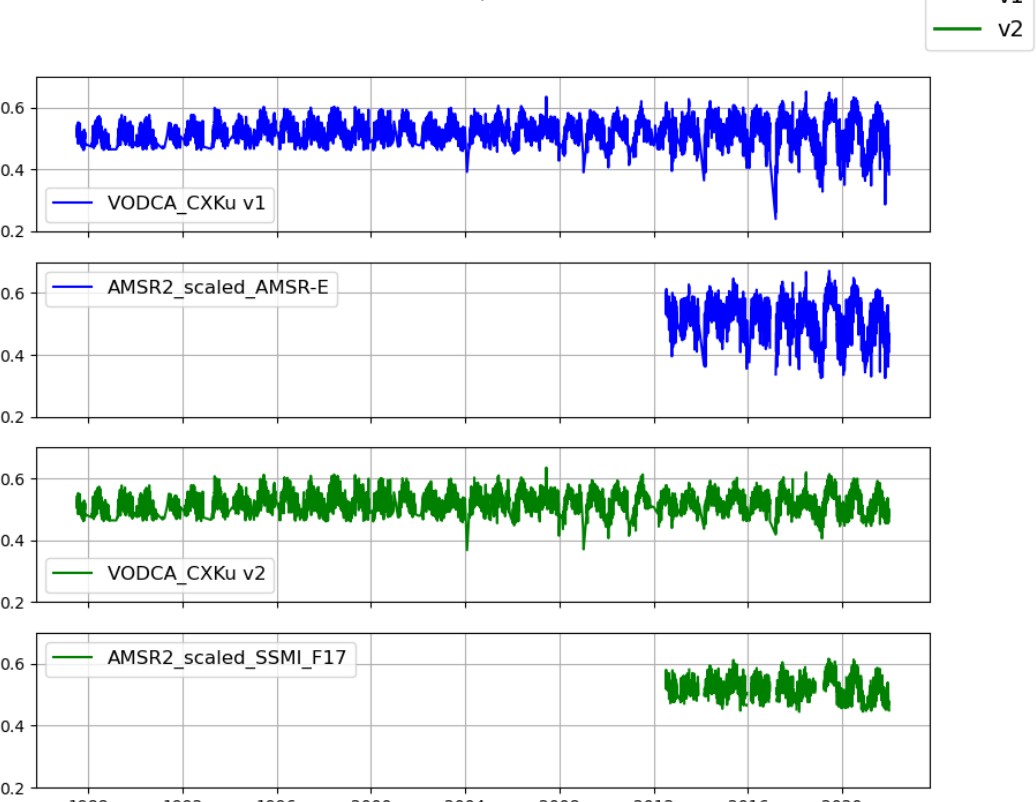

**Figure A13.** Time series of VODCA CXKu v1 (blue, computed with the old methodology), AMSR2 scaled to AMSR-E (blue), VODCA CXKu v2 (green) and AMSR2 scaled to SSMI F17 (green), at a location where TMI is not available.

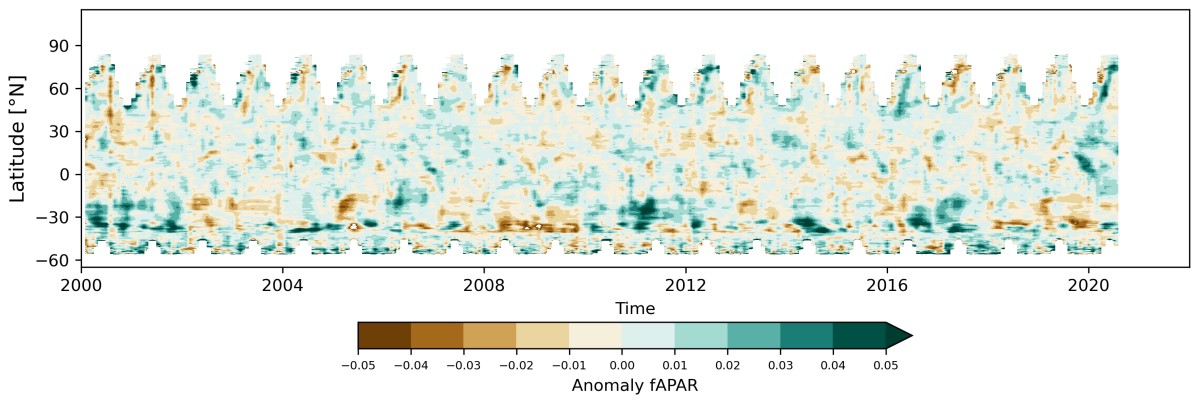

**Figure A14.** Hovmöller diagram showing anomalies of the monthly means per latitude for MODIS fAPAR. Anomalies have been computed as deviations from the long-term climatology (2000 - 2020).

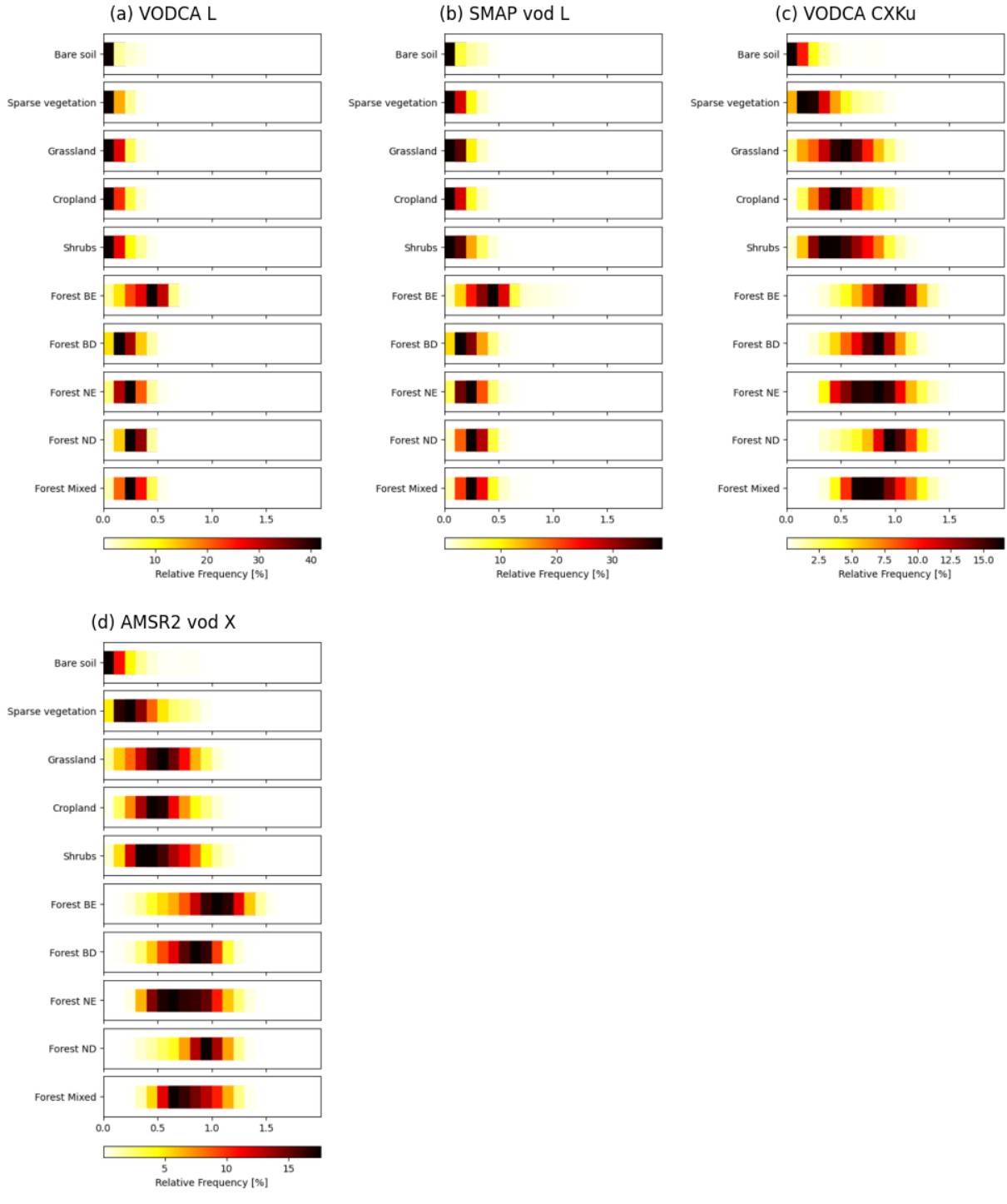

**Figure A15.** Values ranges per LC class for VODCA L (a), SMAP vod L (b), VODCA CXKu (c) and AMSR2 vod X (d). Bins are colored by the relative frequency in percent of the respective values. ESA CCI LC was used.

**Table A1.** Spatial correlation computed over the overlapping period 2010 - 2019, between the VOD products, fAPAR and AGB.

| | Spearman's R | |
|---|---|---|
| | VODCA L | VODCA CXKu |
| AGB | 0.874 | 0.800 |
| fAPAR | 0.78 | 0.714 |

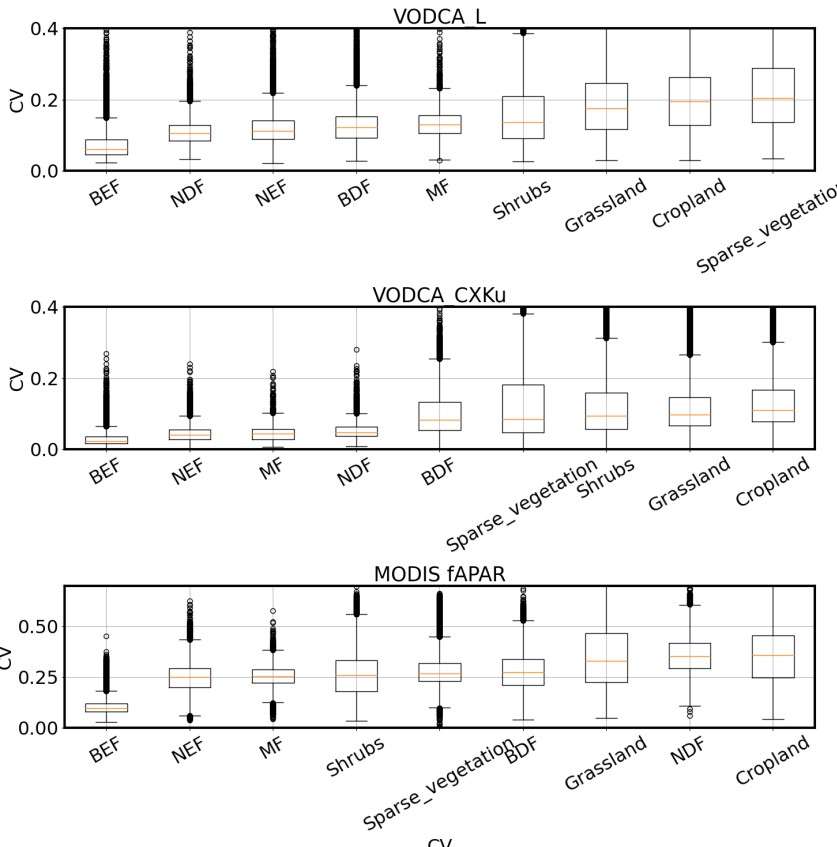

**Figure A16.** Coefficient of variation per LC for VODCA L, VODCA CXKu and fAPAR. ESA CCI LC was used.

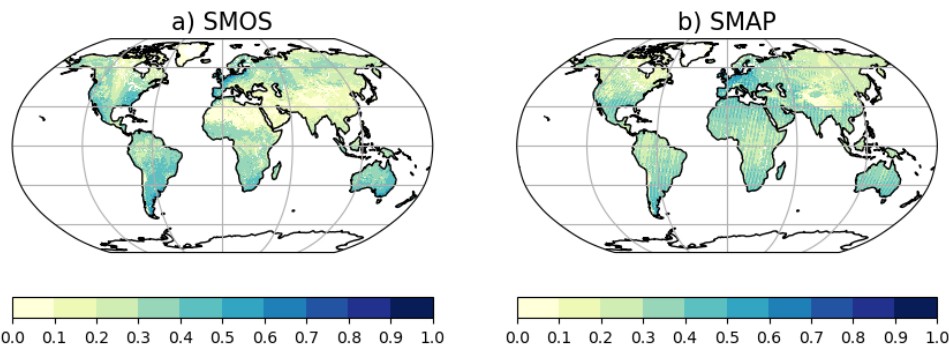

**Figure A17.** Fractional cover of a) SMOS (2010 - 2021) and b) SMAP VOD (2015 - 2021), after flagging. Pixels that have a fractional cover of exactly 0 are shown in white.

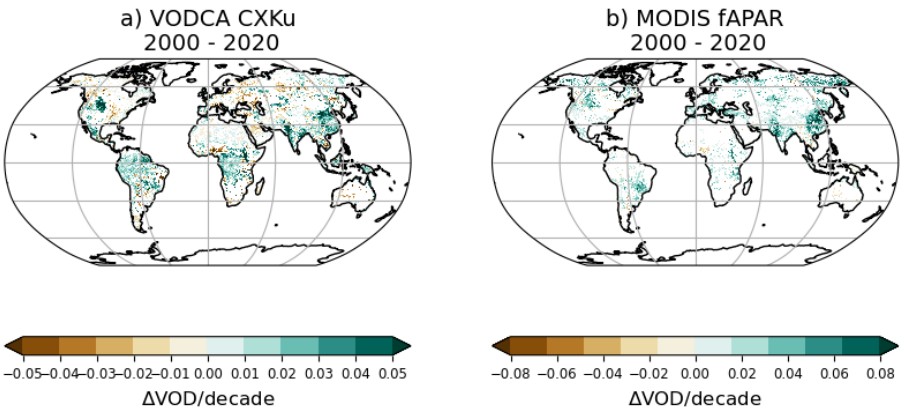

**Figure A18.** Trends for the overlapping period (February 2000 - August 2020) for a) VODCA CXKu and b) fAPAR.

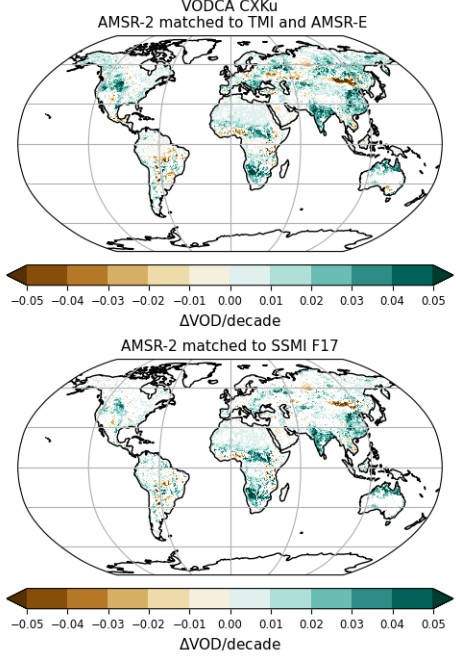

**Figure A19.** Trends (1987 - 2021) for VODCA CXKu computed with the Moesinger et al. (2020) satellite constellation and CDF-matching framework, by merging AMSR2 to TMI below 35°N latitude and AMSR-E above; and for VODCA CXKu computed with the method from this paper, by matching AMSR2 to SSMI F17

**Table A2.** Median Spearman's R between VODCA CXKu and MODIS fAPAR and ASCAT slope for the bulk signal and anomalies.

| Median Spearman's R | | | | |
|---|---|---|---|---|
| LC | fAPAR (bulk) | fAPAR (anomalies) | slope (bulk) | slope (anomalies) |
| All | 0.34 | 0.23 | 0.19 | 0.17 |
| Bare soil | 0.31 | 0.22 | 0.16 | 0.16 |
| Sparse veg. | 0.43 | 0.32 | 0.36 | 0.35 |
| Grassland | 0.57 | 0.30 | 0.48 | 0.32 |
| Cropland | 0.56 | 0.30 | 0.46 | 0.30 |
| Shrubs | 0.59 | 0.41 | 0.16 | 0.15 |
| Forest BE | 0.04 | 0.01 | -0.07 | -0.04 |
| Forest BD | 0.49 | 0.19 | 0.01 | 0.09 |
| Forest NE | 0.09 | 0.04 | 0.14 | 0.13 |
| Forest ND | 0.07 | 0.14 | 0.32 | 0.35 |
| Forest Mixed | 0.13 | 0.05 | 0.04 | 0.13 |

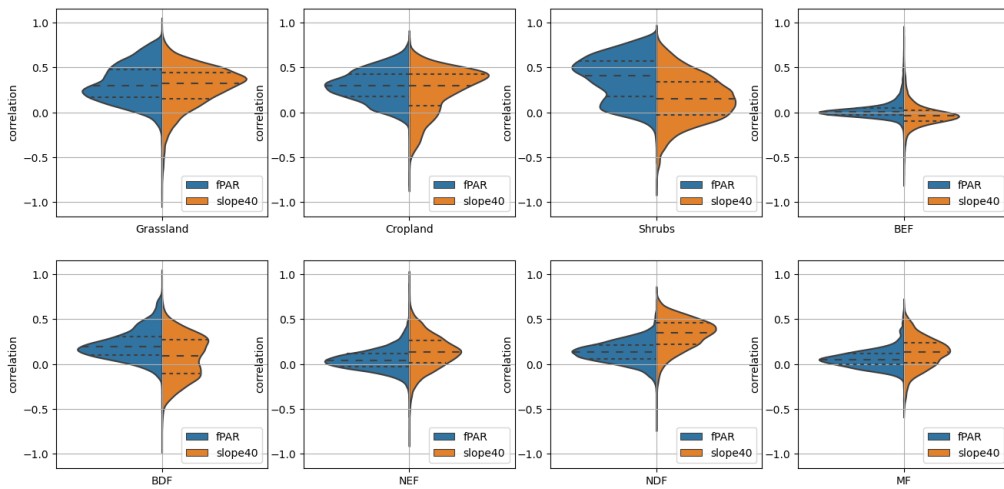

**Figure A20.** Violin plots showing the distribution of Spearman's R between 8-daily VODCA CXKu and MODIS fAPAR (blue) and VODCA CXKu and ASCAT slope (orange) calculated on anomalies, per landcover (a-j).

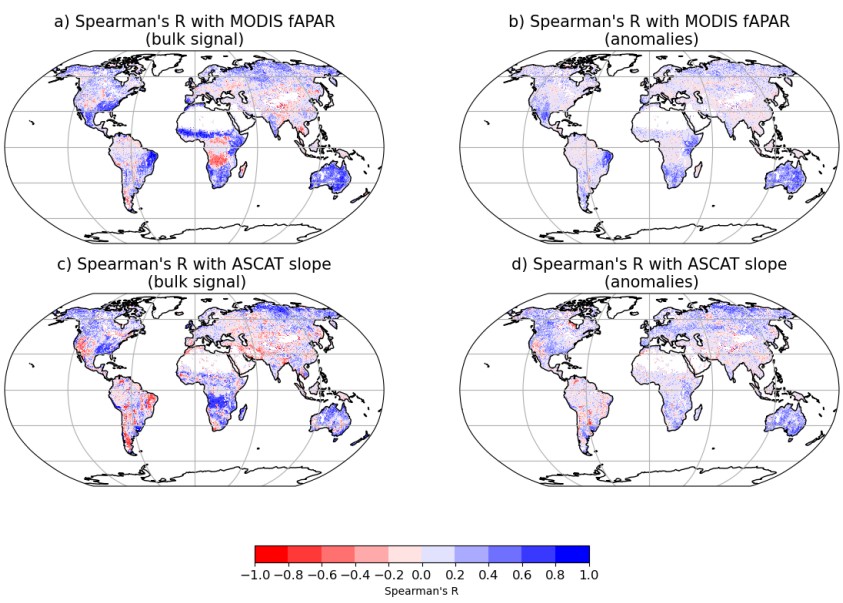

**Figure A21.** Spearman's R between monthly VODCA L and MODIS fAPAR over the period 2010 - 2021 on the bulk signal (a) and anomalies (b). Spearman's R between monthly VODCA L and ASCAT slope over the period 2010 - 2021 on the bulk signal (c) and anomalies (d).

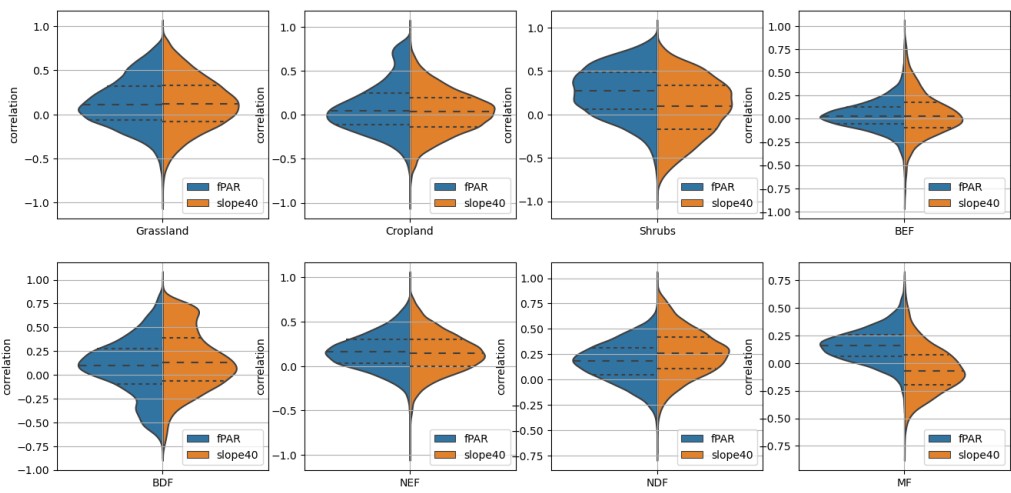

**Figure A22.** Violin plots showing the distribution of Spearman's R between monthly VODCA L and MODIS fAPAR (blue) and VODCA L and ASCAT slope (orange) on the bulk signal, per landcover.

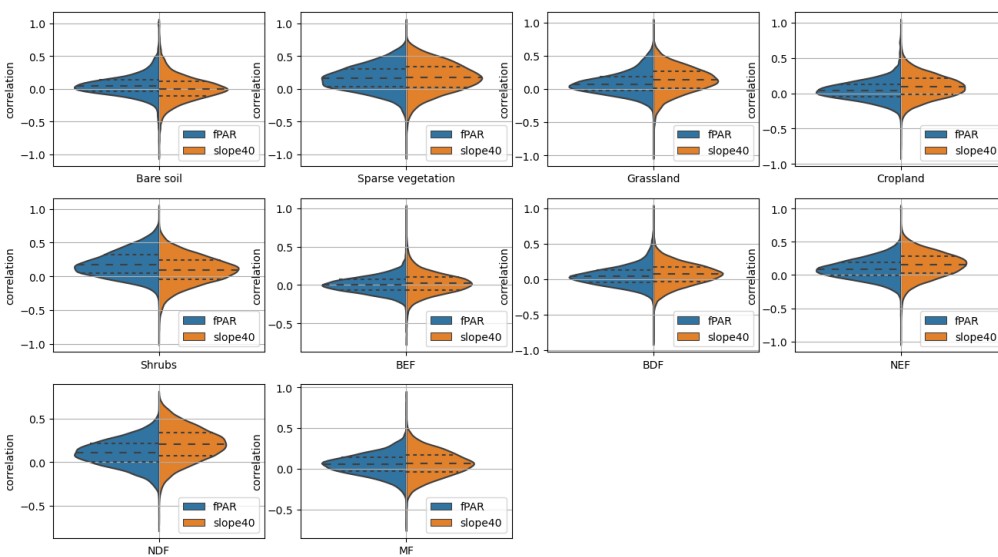

**Figure A23.** Violin plots showing the distribution of Spearman's R between monthly VODCA L and MODIS fAPAR (blue) and VODCA L and ASCAT slope (orange) over anomalies, per landcover.

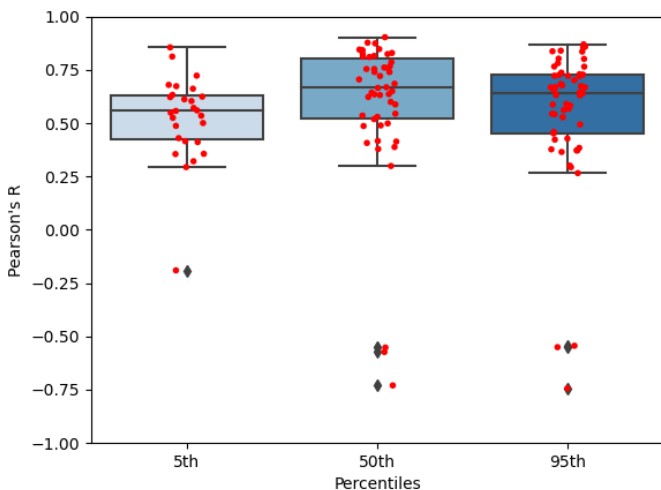

**Figure A24.** Agreement between VODCA CXKu (SD) and sapflow (SD) for the monthly 5th, 50th and 95th percentiles. Red points represent the correlation for each station.

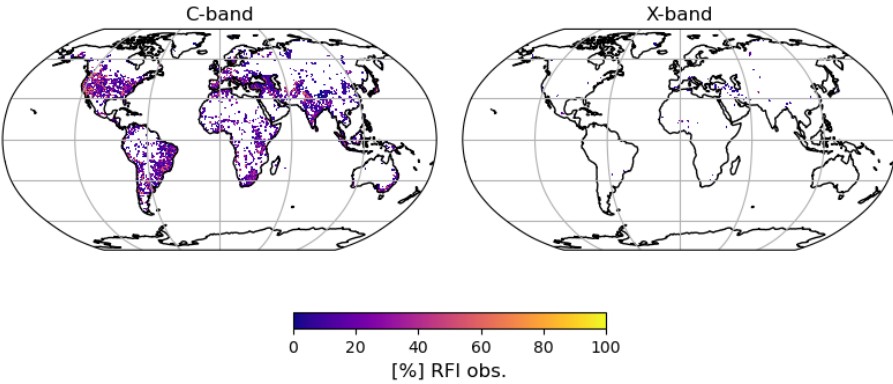

**Figure A25.** Percentage of observations affected by RFI in AMSR2 (2012 - 2021) C- and X-band. The internal flag provided with the LPRM data was used.

(a)

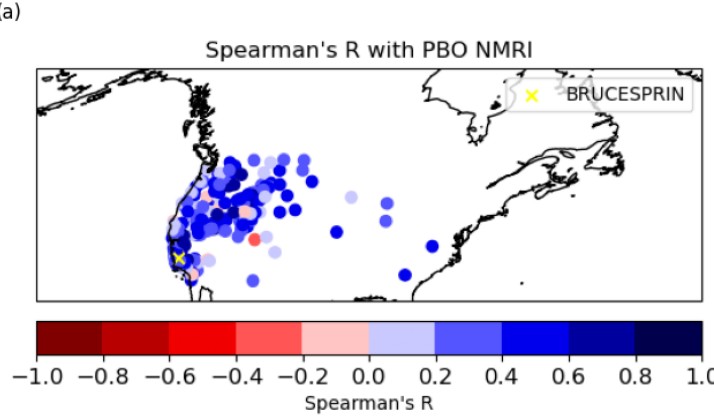

(b)

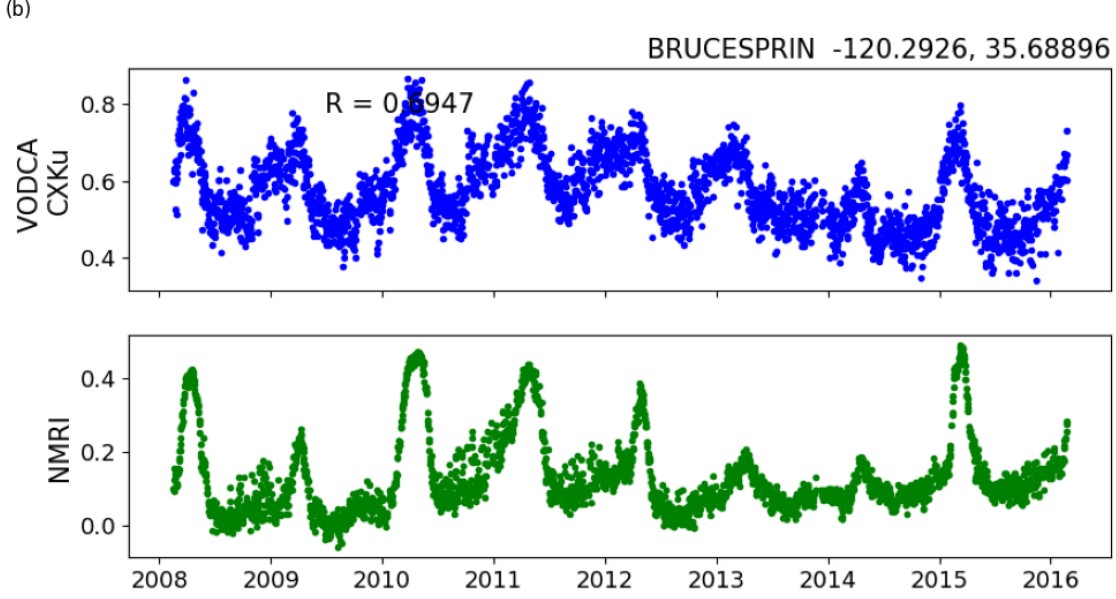

**Figure A26.** (a) Pearson's R between daily VODCA CXKu and NMRI for each PBO station. Only sites with significant correlation are shown. (b) time series of VODCA CXKu (blue) and PBO NMRI (green) for a station with above average agreement between datasets.

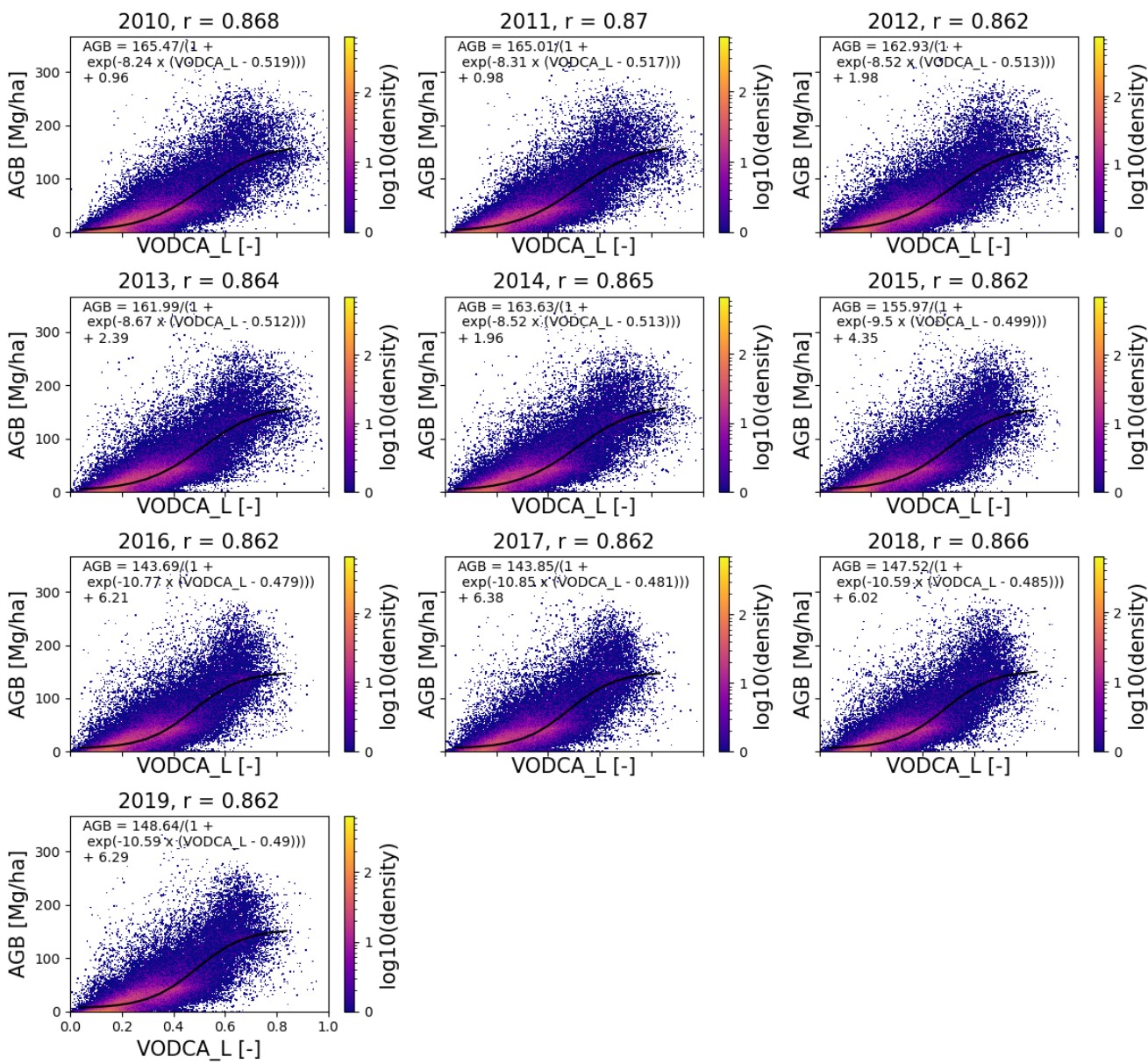

**Figure A27.** Agreement between VODCA CXKu q95 and Xu AGB for each year in the interval 2010 - 2019.

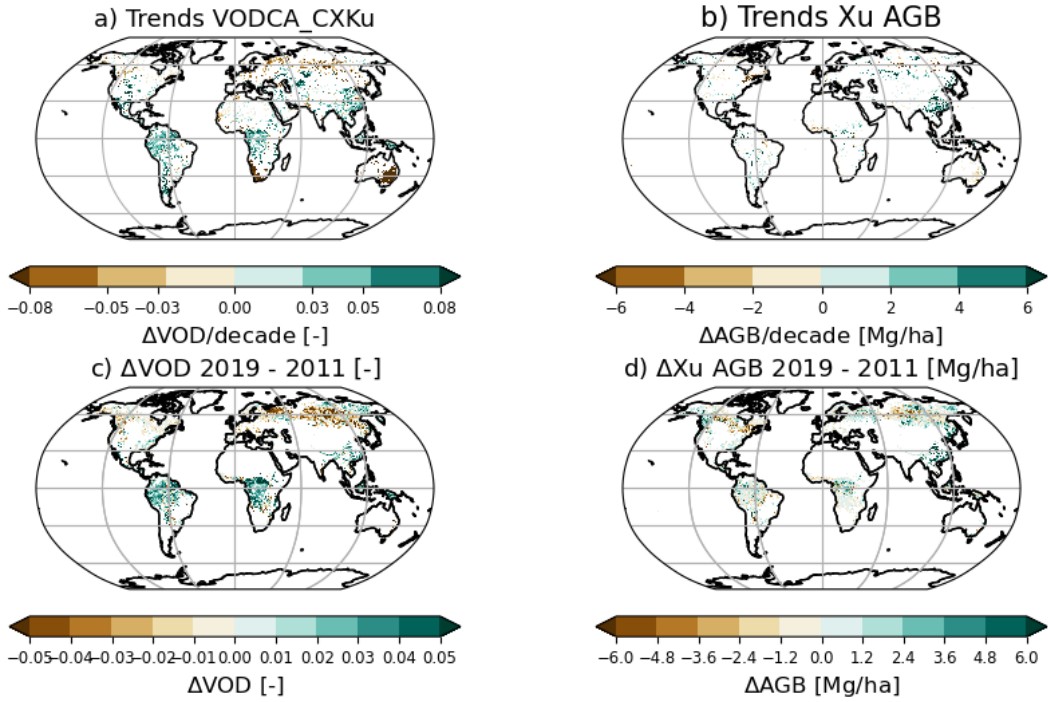

**Figure A28.** a) Theil-Sen trends for VODCA CXKu for 2011 - 2020. b) Theil-Sen trends for Xu AGB for the same period. Difference between the years 2019 - 2011 for c) VODCA CXKu and d) Xu AGB (method of Araza et al. (2023)).

*Author contributions.* WD and RMZ designed the study. RZ performed the analysis and wrote the paper. WP preprocessed auxiliary and evaluation data. All authors contributed to discussions about the methods and results and provided feedback on the paper and data

*Competing interests.* The authors declare that they have no conflict of interest.

*Acknowledgements.* The authors acknowledge the TU Wien University Library for financial support through its Open Access Funding
Program.

We thank Anonymous Reviewer 1 and Reviewer 2, Xiaojun Li, for reviewing this manuscript. Their feedback helped significantly improve our paper.

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
