# Peer review of "VODCA v2: Multi-sensor, multi-frequency vegetation optical depth data for long-term canopy dynamics and biomass monitoring"

_Earth System Science Data, 2024_

## Author Comment (AC1)

**Responses to Anonymous Referee #2**

We thank the Anonymous Referee #2 for their time and effort to review our manuscript, which helped to further increase the quality of the paper. All comments have been addressed carefully.

Below, reviewer comments are marked in red.
Responses to the comments are marked in blue.
Changes that have been made in the manuscript are marked in *italic*.

Major Comments

1. The rationale provided for combining high-frequency VODs (i.e., C, X, and K-band) is currently inadequate. The authors should include detailed scientific evidence, such as assessments of the spatial and temporal correlations among these three frequency VODs.

We agree that the rationale for combining the high-frequency (C-, X- and Ku band) observations was not provided in the first submission, and we thank the reviewer for pointing out this flaw. We relied on the findings of other studies, such as Moesinger et al. (2020, 2022) and Wild et al. (2022), which showed that the three bands correlate strongly. Notably, the work of Moesinger et al. (2022) is very relevant in this respect because it merges C-, X- and Ku band observations to create a standardized vegetation optical depth index (SVODI) and carries out a temporal correlation analysis to show that there is a high agreement between bands (Moesinger et al. (2022), Fig. 1). We provide the results of a spatial and temporal correlation analysis based on data from the descending overpass of AMSR2 (Jun. 2012 – Dec. 2021), which has been used as input in the multi-frequency VODCA CXKu product. The temporal correlation analysis (Fig. 1) reveals strong correlations. The degree of temporal agreement between the different frequencies is similar in magnitude and spatial patterns, with high Spearman's R except for biomes with little inter- and intra-annual variability (desert and humid tropics). These results align with the abovementioned studies (e.g., Fig. 1 from Moesinger et al. (2022)). The spatial correlation analysis also shows very strong agreement between frequencies (Fig. 2). The demonstrated high agreement between the three frequencies justifies fusing observations into one record. We note that we obtained similar results by conducting analyses based on other sensors that provide multiple frequencies (AMSR-E, WINDSAT, TMI, and GPM, results not shown).

[Figure]

Figure 1: Spearman's R between AMSR2 (2012 - 2021) VOD C, VOD X and VOD Ku data. The correlations are based on daily data.

[Figure]

Figure 2: Spatial correlation (Spearman's R) between average AMSR2 (2012 - 2021) VOD C, VOD X and VOD Ku data.

*Figs. 2 and 1 have been added to the Annex Figs. A3 and A2. We further argumented the merging of frequencies in Chapter 3.1, lines 279 - 288.*

2. Despite the work done by the authors in evaluating VOD products, however, I did not see interesting information considering the low correlation values between VOD and the vegetation proxies

selected, for example with the ASCAT slope. Could the authors clearly indicate what the ASCAT slopes represent? Is it vegetation water content or?

We thank the reviewer for the valuable comment and acknowledge that the ASCAT slope parameter was not sufficiently explained in the initial submission of the manuscript. Thus, the results, particularly the low correlation values over certain land cover types, need to be clarified for the readers. The ASCAT scatterometers onboard the Metop satellites measure vertically polarized backscatter at incidence angles between 25° and 65° recorded at a 5.25 GHz frequency (C-band). ASCAT slope, derived using the TU Wien Soil Moisture Retrieval algorithm (Hahn et al., 2017), is a parameter of the second-order Taylor polynomial used to describe the incidence angle dependence of backscatter. The slope is sensitive to scattering mechanisms, where surface scattering leads to a steep slope (i.e. a strong decrease of backscatter with increasing incidence angle), and volume scattering (e.g., caused by vegetation cover) causes scattering in all directions, and thus leads to a flatter slope. Denser vegetation increases volume scattering, and consequently, the slope flattens (Vreugdenhil et al., 2020). ASCAT slope has been shown to be correlated with vegetation density (Hahn et al. (2017), Vreugdenhil et al. (2017)), above-ground fresh biomass (Steele-Dunne et al., 2019) and vegetation phenology and water status (Pfeil et al. (2020), Petchiappan et al. (2022)), similar as VOD. These studies also outline the importance of further research to overcome the limited understanding of the spatio-temporal dynamics of the slope parameter. As such, ASCAT slope is a relatively young parameter that has not yet been fully understood but can potentially offer valuable insight into vegetation water dynamics across a diverse range of biomes. Therefore, comparing VODCA with ASCAT slope serves more as a mutual evaluation of patterns driven by similar vegetation properties and less as a validation of the VODCA dataset. We argue that it is beneficial to provide such a comparison because the ASCAT slope is derived from radar microwave remote sensing. Therefore, it is an entirely independent dataset. Our analysis has revealed widespread moderate correlations between datasets, especially in grassland (median R over all pixels = 0.48) and cropland (0.48) and low to negative correlations in BEF, BDF, NEF MF and shrubs. These results were expected, as several studies (Dostálová et al. (2018), Vreugdenhil et al. (2016, 2017, 2020)) reported similar agreement patterns between ASCAT-derived vegetation indicators and passive VOD. Notably, in BDF, the low correlations were attributed to the active signal being dominated by changes in vegetation structure caused by the increase in tree foliage during the growing season, with leaves absorbing or forward scattering the signal (Dostálová et al., 2018). Over deserts and other arid regions with sparse vegetation (e.g. shrubs), the low agreement with ASCAT data could be explained by the presence of sub-surface scattering in the active signal (Hahn et al. (2017), (Vreugdenhil et al., 2020)). Regarding BEF, the weak positive to weak negative agreement could be caused by the low intra-annual variability for both active and passive data (Vreugdenhil et al., 2016). In NEF, the low correlations could be attributed to challenging environmental conditions for the retrieval algorithms (Vreugdenhil et al. (2016), Moesinger et al. (2020)).

*A more detailed description of the ASCAT slope parameter has been added to Chapter 2.3.2, as well as the argument that the comparison with ASCAT serves as a mutual evaluation, not as validation. Further arguments have been given to Chapter 4.5.2 ASCAT Slope.*

3. As a long-term product fused from multiple satellite sources, it is essential for the authors to dedicate a section to evaluating the continuity of the developed VOD, particularly over time. While the temporal correlation between VOD and vegetation proxies was calculated, this alone does not adequately assess the continuity of VOD.

We thank the reviewer for this comment and agree that the consistency of the VODCA v2 products in time needs to be assessed more thoroughly in the manuscript. We fully agree that the evaluation of VODCA v2 consistency deserves a dedicated section.

To ensure that the merging of multiple sensors and frequencies has not affected the consistency of VOD through time and space, we have computed several additional analyses:

- Time-latitude plots showing the mean monthly VOD (Fig. 3).

- Time-latitude plots showing the monthly anomalies (Fig. 4) with respect to the reference period 1990 - 2020. The anomalies were calculated by the following steps: collecting all observations for a given latitude, determining the monthly mean, subtracting the multi-year monthly average, and eliminating any potential linear trends using ordinary least-squares regression. Therefore, anomalies should represent either natural variability or artefacts due to changes in the sensor constellation.

- Time-latitude plots showing the first-order autocorrelation (AC(1)) of VOD observations in each year (Fig. 6). These were calculated by collecting all observations for a given latitude, filling in missing observations using linear interpolation because AC(1) coefficients strongly depend on the temporal resolution (Moesinger et al., 2020), and calculating the yearly AC(1). We rely on the assertion that there should be a high degree of temporal AC(1) between subsequent observations since VOD is related to gradual changes in plant water content and biomass (Momen et al. (2017), Konings et al. (2016), Moesinger et al. (2020)). Therefore, in Fig. 6, a decrease in AC(1) coincident with the timing of introducing a new sensor would indicate that the merging has led to larger random errors in the product. Similarly, an increase would suggest that the merging has led to lower random error levels.

- Global and hemispherical plots showing the yearly average, yearly anomalies and yearly percentage of valid observations (Figs. 7, 8). These have been calculated from the yearly averages weighted by the cosine of the latitude.

The seasonal dynamics of VODCA CXKu over time and space (Fig. 3 upper) show consistent patterns, with higher VOD in the summer months due to the increase in temperature (in the northern-southern region) or in precipitation (in the subtropics). In VODCA L (Fig. 3 lower), the seasonal patterns are less prevalent, which is to be expected because it also contains information on the woody components of the vegetation layer, which is more constant throughout the year. The seasonality and magnitude of VOD are consistent over time and space in both datasets. Most anomalies in VODCA CXKu and VODCA L (4) appear limited in time, and their start and end do not coincide with sensor changes, thus indicating natural variability. Moreover, the anomaly patterns of VODCA CXKu are very similar to those of MODIS fAPAR (Fig. 5). The yearly AC(1) appears consistent through time in VODCA CXKu (Fig. 6 upper), with some latitudes experiencing a slight increase at approximately 30 - 60 N and 0 - -30 N coincident with the introduction of AMSR-E (Jun. 2002) and TMI (Dec. 1997). At the same time, no consistent decreases in AC(1) can be observed, suggesting that no sensor has led to an increase in random error compared to the previous state of the product. In VODCA L (Fig. 6), we see an increase in AC(1) in almost all latitudes coincident with the introduction of SMAP (Mar. 2015). These results suggest that fusing observations in the overlapping period has led to a more robust product in terms of random error than using only SMOS observations. As a result of this analysis, we reiterate that we expected to see to some degree a change in AC(1) with the merging of sensors, as VODCA CXKu and VODCA L are harmonized (through the removal of bias between sensors and fusion of overlapping observations) but not homogenized (forcing same data characteristics throughout the entire period covered by the merged product). Therefore, as already mentioned in the manuscript, it is crucial to consider the influence of heterogeneous sensor constellation through time for research that delves into higher-order statistics such as variance and autocorrelation temporally (Smith et al., 2022). The global and hemisphere mean VOD plots show no breaks when new sensors are introduced in either VODCA CXKu (Fig. 7) or VODCA L (Fig. 8). Therefore, we attribute the changes in VOD to natural variability. Concerning VODCA CXKu, anomalies have been observed to coincide with El Niño-Southern Oscillation (ENSO) variations (Dorigo et al. (2021), Dorigo et al. (2022), Zotta et al. (2023)) especially in the Southern Hemisphere, where there is a clear connection between ENSO and vegetation activity (Martens et al. (2017)). Regarding the bulk signal, we can observe a clear positive trend in VODCA CXKu, consistent

[Figure]

with reports on global greening based on different and independent satellite sources (e.g., Piao et al. (2020), Chen et al. (2024), Zhang et al. (2017)).

Figure 3: Hovmöller diagrams showing the monthly mean VOD per latitude for VODCA CXKu and VODCA L.

[Figure]

Figure 4: Hovmöller diagrams showing anomalies of the monthly means per latitude for VODCA CXKu and VODCA L.

[Figure]

Figure 5: Hovmöller diagrams showing anomalies of the monthly means per latitude for MODIS fAPAR.

[Figure]

Figure 6: Hovmöller diagrams showing the yearly AC(1) per latitude for VODCA CXKu and VODCA L.

[Figure]

Figure 7: Global and hemisphere time-series of VODCA CXKu.

[Figure]

Figure 8: Global and hemisphere time-series of VODCA L.

*We introduced a new section in the Chapter Results (Chapter 4.2), "Spatio-temporal consistency", that presents the abovementioned results. We include Figures 3, 4 and 6 in Chapter 4.2. We include the global and hemisphere time series (Fig. 7, VODCA CXKu) and (Fig. 8, VODCA L) in the Annex, as well as the time-latitude plot showing fAPAR (Fig. 5).*

4. Why were all sensors fused together instead of selecting the highest quality VODs during overlapping observation periods? For example, considering SMAP's superior quality over SMOS, which is more prone to radio frequency interference (RFI), was an evaluation done to determine if fusing SMOS and SMAP data during their overlapping periods might compromise data quality compared to using SMAP alone? Notably, L-VOD can provide global coverage with a 10-day temporal resolution, regardless of whether it uses only SMAP or both sensors.

We thank the reviewer for these critical questions and remarks. The methodology for creating VODCA v2 (and v1) products is based on the methodology for creating harmonised long-term multi-satellite-based climate data records within the ESA Soil Moisture CCI (CCI SM) project (Dorigo et al. (2017); Gruber et al. (2019)). In the early stages of the CCI SM project, namely in version CCI SM v2, a decision-tree-based approach, similar to that suggested by the reviewer, was used to select the best SM time series (Gruber et al. (2019)). While this method proved useful for merging soil moisture products from various satellites, it was not the optimal statistical choice, as according to statistical theory, an optimal estimate would be the weighted average of the individual measurements (Gelb et al. (1974)). Building on the lessons learned from CCI SM, we decided to use both SMOS and SMAP observations rather than choosing the best sensor. We evaluated the changes in data quality due to fusing SMOS and SMAP data and provided the results in Fig. 5 in the original manuscript. We looked at the difference in AC(1) between VODCA L and SMOS (Fig. 5 d; for 2010 - 2021) and between VODCA L and SMAP (Fig.5 e; for 2015 - 2021), using collocated observations in time and space. Compared to SMOS, VODCA L exhibits higher AC(1) almost globally. Compared to SMAP, VODCA L exhibits areas with both decreases and increases in AC(1). Areas with consistent decreases in AC(1) arise, especially in desert and arid regions, while in vast parts of the vegetated areas, we see an increase in AC(1) when using SMOS and SMAP jointly instead of SMAP alone. Therefore, even though our solution is not optimal globally, we decided to use both sensors instead of only SMAP. We acknowledge that the methodology used to produce VODCA L represents one solution for fusing SMOS and SMAP out of several and is not necessarily flawless in all circumstances, and we admit that further research might be needed to fully assess under which circumstances (land cover type, moisture conditions, RFI, etc.) it might be advantageous to use SMAP alone.

Moreover, by using both sensors, we create a product that is also robust in case of temporary outages,

such as recently encountered for SMAP in November 2023 (NSIDC, 2023) and for SMOS in February 2024 (ESA, 2024), and in case of permanent sensor failures.

We agree with the reviewer that L-VOD can provide global coverage with a 10-day temporal resolution regardless of whether it uses only SMAP or both sensors. However, having more sensors and thus more observations to compute the 10d median should be favoured as they better represent the actual dynamics within this time window instead of having a snapshot based on fewer (1 - 2) observations.

Even though a more optimal solution for fusing SMOS and SMAP could exist and should be the topic of future research, VODCA L represents a viable alternative to existing long-term L-band VOD products, providing a longer observation period (2010 - 2021) than SMAP (2015 - 2021) and globally lower random error levels compared to SMOS.

*We modified the caption of Fig. 5 (original manuscript, Fig. 8 rebuttal) to stipulate that the difference in AC(1) is computed over collocated observations, which means that only the overlapping observations between VODCA L and SMAP (2015 - 2021) and VODCA L and SMOS (2010 - 2021) have been used. We acknowledge the limitations of our framework for merging SMOS and SMAP in Chapter 4.3 Changes in AC(1) in lines 464 - 473 and outline the importance of further research.*

5. What advanced features do the fusion products offered by the authors have compared to existing L-band fused VOD products? Furthermore, comparisons with the Xu' AGB cannot highlight any advantages of VODCA L. For example, in Fig. 14, obvious inconsistencies can be found in the Amazonian and North American boreal forests. The authors might consider comparing this with existing L-band products to better illustrate its benefits.

We thank the reviewer for these valuable questions and suggestions. We acknowledge the existence of other fused products, namely SMOSSMAP-IB, described in Li et al. (2022). However, we did not compare VODCA L with SMOSSMAP-IB, as this product is unavailable to us and has no open access. We mention that the purpose of our product is not necessarily to outperform others but to present an alternative based on a different VOD retrieval algorithm than the available products, and with a different merging methodology. We kindly note that ESSD papers focus on accurately describing and evaluating new datasets, while comparisons to other methods are beyond regular articles's scope (ESS AIMS & Scope). We agree that the comparison with Xu AGB reveals dissimilar patterns in the Amazon and North American boreal forests. Because of that, in the manuscript, we referenced the research conducted by Araza et al. (2023), who obtained similar biomass change patterns from L-band VOD. However, we agree with the reviewer that including a comparison between VODCA L and an existing L-band product would complement this analysis. Therefore, we include SMOS-IC v2 (Wigneron et al. (2021), Li et al. (2021), Li et al. (2022), Li et al. (2020)) produced by INRAE (Institut National de Recherche Agronomiques) Bordeaux Soil Moisture and VOD Products and made available at `https://ib.remote-sensing.inrae.fr/`. We compute the difference between the average SMOS-IC VOD for 2019 and 2011 (Fig. 9 b)) in a similar fashion as conducted for VODCA L (Fig.9 a)). The distribution of positive and negative change patterns is similar to that observed for VODCA L, including in the Amazon basin and the North American boreal forests. Differences can be observed mainly in magnitude, which was expected since VODCA L uses a different retrieval algorithm and also includes observations from SMAP.

[Figure]

Figure 9: Difference between the years 2019 and 2011 for a) VODCA L, b) SMOS-IC VOD, c) Xu AGB (method of Araza et al. (2023) for reducing the inter-annual variability of the original time-series). The analysis is restricted to forested areas.

*We added a section describing SMOS-IC in Chapter 2.3.6. We included the abovementioned analysis, including SMOS-IC, in Chapter 4.6 AGB.*

6. In line 45 of the introduction, the authors overlooked the significant contribution of some French teams to the application of VOD in biomass studies. A more thorough review of existing research is needed.

We thank the reviewer and agree that we overlooked some valuable contributions regarding the application of VOD in biomass studies. We referenced the following studies (listed in chronological order):

- Brandt et al. (2018),

- Fan et al. (2019)

- Wigneron et al. (2021),

- Qin et al. (2021),

- Yang et al. (2022),

- Yang et al. (2023)

*We added the abovementioned studies in lines 46-48 in Chapter 1 Introduction.*

Specific Comments

Abstract: The Spearman correlations between the vegetation proxies used by the authors and VOD were all below 0.6. Given this, how can one assert that there is 'good agreement' or that they 'agree well'?

We thank the reviewer for this comment and reemphasize that this is a comparison to evaluate VODCA's information content, but not a validation with the goal of obtaining maximum correlation. We agree that correlations below 0.6 do not qualify as 'good agreement' and corrected to: "moderate agreement" in lines 20, 22. We also mention that VODCA provides complementary spatio-temporal information containing additional information on the state of the canopy in lines 19-20.

Lines 79-80: As mentioned above, the authors need to provide a substantial basis for their claims.

Comment has been addressed in the response to Major Comment 1

Lines 93-95: It is critical to assess the spatial and temporal continuity of VODCA V2

Comment has been addressed in the response to Major Comment 3

Table 1: It appears that not all sensors provide global coverage, which the authors should clarify. Additionally, which observation angle was used for SMOS in this study?

We added a column to the table called "Remarks", where we specify the coverage of sensors that do not provide global coverage (TMI, GPM) and the observation angles used for SMOS ([37.5, 42.5, 47.5, 52.5, 57.5]).

Line 46: Bin center 2.5-62.5°?

We assume the reviewer referred to line 146. Modified to "which is an interferometric L-band (1.4 GHz) 2-D radiometer that takes measurements for multiple incidence angles between 0 - 65°".

Line 47: Yes, but what strategies did the authors use to rigorously raise the issue of RFI in SMOS?

We assume the reviewer referred to line 146. For SMOS, the filtering of RFI is based on the probability information that is supplied by the SMOS Level 3 data. We added more information on the RFI mitigation in Chapter 3.2 Preprocessing, in lines 305 - 310.

Line 191: Duplicated.

Modified to: "We resampled the ASCAT slope dataset to match the VODCA grid by averaging all points within a cell."

Methods: Flowcharts need to be used to describe the fusion process.

Added a flowchart in the manuscript at Fig. 2 in Chapter 3.1 General framework.

[Figure]

Figure 10: Schematic overview of the VODCA v2 processing framework.

Line 295: Why?

The window size of 120 days was chosen to preserve the seasonality and ensure that outliers are identified without being misinterpreted as part of the seasonal trend. A threshold of 3 MADs has been selected to eliminate significant deviations that cannot be explained, given that we are looking at gradual changes in vegetation and to prevent excluding valuable data. Added in lines 323 - 326.

Line 321: Further detailed results, such as plotting the time series, are needed to demonstrate the improvements achieved by using SSMI F17 as a reference.

[Figure]

[Figure]

Figure 11: Timeseries of VODCA CXKu v1 (computed with the old methodology) and VODCA CXKu v2 at a location where TMI is not available.

To illustrate the inconsistencies mentioned in line 321, we show time series (Fig. 11) at a location outside the coverage of TMI (latitude above 38 °) for both scaling methodologies. V1 (blue) shows VODCA CXKu and AMSR2 time series computed with the method presented in Moesinger et al. (2020) where AMSR2 is scaled to AMSR-E without overlap. V2 (green) shows VODCA CXKu and AMSR2 time series computed using the method presented in the manuscript. We can observe that in v1, the CDF-matching did not produce the desired outcome, resulting in the inconsistencies mentioned in the manuscript. We show Fig. 11 in the revised manuscript in Fig. A7 and we discuss it in Chapter 4.4 Trends, in lines 503-506.

Lines 338-339: For pixels that do not meet the specified condition, how is the AC (1) value determined, and how is the weighting assigned?

At each location, we compute AC(1) for each overlapping period between sensors. Therefore, we have several sets of weights at each pixel based on the number of overlapping periods. By comparing AC(1) based on the same observational period, if a disturbance happens, it should be reflected in each sensor's AC(1). We make this more clear in lines 371 -373 in Chapter 3.4 Temporal autocorrelation as a measure of random error.

Lines 369-370: How did the authors conclude this? The effective scattering albedo seems to have a greater effect on VOD values than the surface roughness parameter.

The reviewer is correct and both single scattering albedo and roughness parameter have an effect on the retrieved VOD values. We clarified in lines 400 - 403.

Did Figure 4 convey any useful information?

We agree that Figure 4 is not vital and move it to the Annex.

Figure 5: Typos. Additionally, the discussion of AC(1) is overly technical for the results section and would be more appropriately detailed in the methodology.

We agree that the AC(1) discussion is somewhat technical; however, we believe it fits better in the results section as it shows an analysis that is based on the datasets resulting from the VODCA v2 framework. We moved the AC(1) discussion to its own dedicated subchapter: 4.3 Changes in AC(1).

Figure 6: The units don't seem to make sense.

For a) and e) We changed $\Delta$ VOD/decade to $\Delta$ VOD, as we have only a decade of observations for VODCA L. We also changed the $\Delta$ AGB/decade to $\Delta$ AGB so that it fits the unit of measurement. Figure 6 is Figure 9 in the updated manuscript.

Figure 7: There is no colorbar in this figure.

We added a color bar.

Why there is a negative correlation between VODCA CXKu and ASCAT slope in Fig. 8? The authors' explanation in line 457 lacks persuasiveness. Could the authors elaborate, particularly concerning the types of savannas in the Brazilian Amazon as depicted in Fig. 8c?

The subtropical regions in South-America are characterised by heterogeneous landscapes, with crop-, grass- and shrublands and forests. In addition, some areas are seasonally flooded. Petchiappan et al. (2022) performed a detailed analysis of ASCAT slope in relation to meteorological drivers precipitation, radiation and specific humidity. In pixels with 80% dominant land cover, slope followed the precipitation seasonality over land covers with shorter rooting depths, crop- and grasslands. Over herbaceous cover, peaks in slope were observed during the peak of precipitation and radiation. For BEF, the peak of the slope coincided with the peak in radiation. Only for shrublands the peak of slope did not correspond to the peak in radiation or precipitation. As also pointed out in Petchiappan et al. (2022) and analyzed by Chave et al. (2010), the highest seasonality in litterfall in South America is in low-stature forests as those found in the Cerrado. Thus, for pixels with heterogeneous land cover in the Cerrado, the negative correlations may be attributed to the sensitivity of the slope to vegetation structure changes as found in BDF Vreugdenhil et al. (2017).

Line 471: need a reference, e.g., 10.1038/s41561-022-01087-x

We added the suggested reference: Fan et al. (2023).

Line 478: It is necessary to show RFI map to make this clear.

We computed RFI maps based on AMSR2 C- and X-band using the RFI flag provided with the LPRM data. The maps show the percentage of observations flagged as RFI in the AMSR2 period. We added the maps to Fig. A20.

[revised manuscript text omitted]

---

## Author Comment (AC2)

**Responses to Anonymous Referee #1**

We thank the Anonymous Referee #1 for their time and effort to review our manuscript, which helped to further increase the quality of the paper. All comments have been addressed carefully.

Below, reviewer comments are marked in red.
Responses to the comments are marked in blue.
Changes that have been made in the manuscript are marked in *italic*.

Major Comments

1. Given the potential for extensive use in long-term ecological applications due to its 35-year timespan (1987-2021), ensuring consistency throughout the VODCA CXKu & L data is paramount. To this end, I strongly recommend including yearly time series of VODCA CXKu / L at global, continental, or landcover scales.

We thank the reviewer for this valuable comment and agree that the consistency of VODCA v2 products in space and time is critical, given the potential and likely user uptake for long-term ecological applications. Therefore, we added a dedicated section in the Chapter Results, "Spatio-temporal consistency".

To ensure that the merging of multiple sensors and frequencies has not affected the consistency of VOD through time and space, we have computed several additional analyses:

- Global and hemispherical plots showing the yearly average, yearly anomalies and yearly percentage of valid observations (Figs. 1, 2). These have been calculated from the yearly averages weighted by the cosine of the latitude.

- Time-latitude plots showing the mean monthly VOD (Fig. 3).

- Time-latitude plots showing the monthly anomalies (Fig. 4) with respect to the reference period 1990 - 2020. The anomalies were calculated by the following steps: collecting all observations for a given latitude, determining the monthly mean, subtracting the multi-year monthly average, and eliminating any potential linear trends using ordinary least-squares regression. Therefore, anomalies should represent either natural variability or artefacts due to changes in the sensor constellation.

- Time-latitude plots showing the first-order autocorrelation (AC(1)) of VOD observations in each year (Fig. 5). These were calculated by collecting all observations for a given latitude, filling in missing observations using linear interpolation because AC(1) coefficients strongly depend on the temporal resolution (Moesinger et al., 2020), and calculating the yearly AC(1). We rely on the assertion that there should be a high degree of temporal AC(1) between subsequent observations since VOD is related to gradual changes in plant water content and biomass (Momen et al. (2017), Konings et al. (2016), Moesinger et al. (2020)). Therefore, in Fig. 5, a decrease in AC(1) coincident with the timing of introducing a new sensor would indicate that the merging has led to larger random errors in the product. Similarly, an increase would suggest that the merging has led to lower random error levels.

The global and hemisphere mean VOD plots show no breaks when new sensors are introduced in either VODCA CXKu (Fig. 1) or VODCA L (Fig. 2). Therefore, we attribute the changes in VOD to natural variability. Concerning VODCA CXKu, anomalies have been observed to coincide with El Niño-Southern Oscillation (ENSO) variations (Dorigo et al. (2021), Dorigo et al. (2022), Zotta et al. (2023)) especially in the Southern Hemisphere, where there is a clear connection between ENSO and vegetation activity (Martens et al. (2017)). Regarding the bulk signal, we can observe a clear positive trend in VODCA CXKu, consistent with reports on global greening based on different and independent satellite sources (e.g., Piao et al. (2020), Chen et al. (2024), Zhang et al. (2017)).

The seasonal dynamics of VODCA CXKu over time and space (Fig. 3 upper) show consistent patterns, with higher VOD in the summer months due to the increase in temperature (in the northern-southern region) or in precipitation (in the subtropics). In VODCA L (Fig. 3 lower), the seasonal patterns are less prevalent, which is to be expected because it also contains information on the woody components of the vegetation layer, which is more constant throughout the year. The seasonality and magnitude of VOD are consistent over time and space in both datasets. Most anomalies in VODCA CXKu and VODCA L (4) appear limited in time, and their start and end do not coincide with sensor changes, thus indicating natural variability. Moreover, the anomaly patterns of VODCA CXKu are very similar to those of MODIS fAPAR (Fig. 6). The yearly AC(1) appears consistent through time in VODCA CXKu (Fig. 5 upper), with some latitudes experiencing a slight increase at approximately 30 - 60 N and 0 - -30 N coincident with the introduction of AMSR-E (Jun. 2002) and TMI (Dec. 1997). At the same time, no consistent decreases in AC(1) can be observed, suggesting that no sensor has led to an increase in random error compared to the previous state of the product. In VODCA L (Fig. 5), we see an increase in AC(1) in almost all latitudes coincident with the introduction of SMAP (Mar. 2015). These results suggest that fusing observations in the overlapping period has led to a more robust product in terms of random error than using only SMOS observations. As a result of this analysis, we reiterate that we expected to see to some degree a change in AC(1) with the merging of sensors, as VODCA CXKu and VODCA L are harmonized (through the removal of bias between sensors and fusion of overlapping observations) but not homogenized (forcing same data characteristics throughout the entire period covered by the merged product). Therefore, as already mentioned in the manuscript, it is crucial to consider the influence of heterogeneous sensor constellation through time for research that delves into higher-order statistics such as variance and autocorrelation temporally (Smith et al., 2022).

[Figure]

Figure 1: Global and hemisphere time-series of VODCA CXKu.

[Figure]

Figure 2: Global and hemisphere time-series of VODCA L.

[Figure]

Figure 3: Hovmöller diagrams showing the monthly mean VOD per latitude for VODCA CXKu and VODCA L.

[Figure]

Figure 4: Hovmöller diagrams showing anomalies of the monthly means per latitude for VODCA CXKu and VODCA L.

[Figure]

Figure 5: Hovmöller diagrams showing the yearly AC(1) per latitude for VODCA CXKu and VODCA L.

[Figure]

Figure 6: Hovmöller diagrams showing anomalies of the monthly means per latitude for MODIS fAPAR.

*We introduced a new section in the Chapter Results (Chapter 4.2), "Spatio-temporal consistency", that presents the abovementioned results. We include Figures 3, 4 and 5 in Chapter 4.2. We include the global and hemisphere time series (Fig. 1, VODCA CXKu) and (Fig. 2, VODCA L) in the Annex, as well as the time-latitude plot showing fAPAR (Fig. 6).*

Minor Comments

1. Line 15: does the canopy include trunks?

We found conflicting information based on the domain (biology vs. forest ecology) concerning the definition of the canopy. To avoid confusion, we defined it in line 15 of the original manuscript as including branches and trunks.

2. Line 125 "Ku-band (19.4 GHz)" and "Ku-band (18.7 GHz)", microwave at 19.4 GHz and 18.7 GHz belongs to K-band (18-27 Ghz), why "Ku-band" was used in this ms?

Thank you very much for your comment. Indeed, both frequencies used belong to the K-band.

We are using the 18.7 GHz band, which is at the edge of the K band. Some sensors, such as AMSR2, also provide the 23.6 GHz band from the middle of the K band. Therefore, we call the 18.7 GHz band Ku-band to differentiate between these K-band frequencies. This notation has been used within the framework of the European Space Agency Climate Change Initiative (CCI) (`https://climate.esa.int/en/projects/soil-moisture/`) and the Copernicus Climate Change Service (C3S) (`https://climate.copernicus.eu/`), and we adopted it since the single-sensor VOD data from VODCA is produced in these projects. The notation is used throughout the CCI Soil Moisture ATBD in Dorigo et al. (2017). Moreover, many independent studies which use data in the 18.7 GHz and 19.4 GHz frequencies use the terminology Ku-band (e.g., Fan et al. (2018), Santi et al. (2017)).

**References**

Chen, X., Chen, T., He, B., Liu, S., Zhou, S., and Shi, T.: The global greening continues despite increased drought stress since 2000, Global Ecology and Conservation, 49, e02 791, 2024.

Dorigo, W., Wagner, W., Albergel, C., Albrecht, F., Balsamo, G., Brocca, L., Chung, D., Ertl, M., Forkel, M., Gruber, A., et al.: ESA CCI Soil Moisture for improved Earth system understanding: State-of-the art and future directions, Remote Sensing of Environment, 203, 185–215, 2017.

Fan, L., Wigneron, J.-P., Xiao, Q., Al-Yaari, A., Wen, J., Martin-StPaul, N., Dupuy, J.-L., Pimont, F., Al Bitar, A., Fernandez-Moran, R., et al.: Evaluation of microwave remote sensing for monitoring live fuel moisture content in the Mediterranean region, Remote Sensing of Environment, 205, 210–223, 2018.

Konings, A. G., Piles, M., Rötzer, K., McColl, K. A., Chan, S. K., and Entekhabi, D.: Vegetation optical depth and scattering albedo retrieval using time series of dual-polarized L-band radiometer observations, Remote sensing of environment, 172, 178–189, 2016.

Martens, B., Miralles, D. G., Lievens, H., Van Der Schalie, R., De Jeu, R. A., Fernández-Prieto, D., Beck, H. E., Dorigo, W. A., and Verhoest, N. E.: GLEAM v3: Satellite-based land evaporation and root-zone soil moisture, Geoscientific Model Development, 10, 1903–1925, 2017.

Moesinger, L., Dorigo, W., de Jeu, R., van der Schalie, R., Scanlon, T., Teubner, I., and Forkel, M.: The global long-term microwave vegetation optical depth climate archive (VODCA), Earth System Science Data, 12, 177–196, 2020.

Momen, M., Wood, J. D., Novick, K. A., Pangle, R., Pockman, W. T., McDowell, N. G., and Konings, A. G.: Interacting effects of leaf water potential and biomass on vegetation optical depth, Journal of Geophysical Research: Biogeosciences, 122, 3031–3046, 2017.

Piao, S., Wang, X., Park, T., Chen, C., Lian, X., He, Y., Bjerke, J. W., Chen, A., Ciais, P., Tømmervik, H., et al.: Characteristics, drivers and feedbacks of global greening, Nature Reviews Earth & Environment, 1, 14–27, 2020.

Santi, E., Paloscia, S., Pampaloni, P., Pettinato, S., Nomaki, T., Seki, M., Sekiya, K., and Maeda, T.: Vegetation water content retrieval by means of multifrequency microwave acquisitions from AMSR2, IEEE Journal of Selected Topics in Applied Earth Observations and Remote Sensing, 10, 3861–3873, 2017.

Smith, T., Traxl, D., and Boers, N.: Empirical evidence for recent global shifts in vegetation resilience, Nature Climate Change, 12, 477–484, 2022.

Zhang, Y., Song, C., Band, L. E., Sun, G., and Li, J.: Reanalysis of global terrestrial vegetation trends from MODIS products: Browning or greening?, Remote Sensing of Environment, 191, 145–155, 2017.

---

## Author Response (AR2)

**Responses to Anonymous Referee #1**

We thank the Anonymous Referee #1 for their time and effort to review our revised manuscript, which helped to further increase the quality of the paper. All comments have been addressed carefully.

Below, reviewer comments are marked in red.
Responses to the comments are marked in blue.
Changes that have been made in the manuscript are marked in *italic*.

Comments

The authors partially addressed my comments. But, I still have some concerns which is mainly about the consistency evaluation and assessment.

1) The sharp increase of VOD CXKu in 1992 coincides with the introduction of a new sensor, SSMI F11. Is it caused by the unresolved bias among sensors? Similar trend can be observed with the 2003 decrease (coinciding with the introduction of AMSR-E), and the 2012 increase (coinciding with the introduction of AMSR2). Please explain it

We thank the reviewer for this important question concerning the continuity of VODCA CXKu.

- Regarding the SSM/I sensors, VOD (and soil moisture) observations were retrieved from the GPM SSMI Common Calibrated Brightness Temperatures L1C produced by NASA (Berg, 2021). This L1C dataset provides recalibrated brightness temperatures for the SSM/I sensors using a common basis (GPM GMI brightness temperature) to enable retrievals of consistent datasets for the entire SSM/I period (Berg et al., 2016, 2018; Berg, 2021). This dataset is part of the Fundamental Climate Data Record (FCDR) of brightness temperatures (Tb) and provides, according to NASA, the best intercalibration of SSM/I sensors available (Berg et al., 2018). In the VODCA framework, we, therefore, concatenate the VOD retrievals from SSM/I F8, F11, and F13 into a single record without further inter-calibration at the VOD level. As F8 and F11 do not overlap with each other or any other radiometer used in the VODCA framework, further intercalibration and validation steps are challenging and thus we rely on the FCDR as provided by NASA. The patterns in VODCA CXKu are fully compliant with patterns observed in the single-frequency VODCA v1 Ku-band, both in terms of global anomaly time-series (Dorigo et al. (2021), Fig. SB2.6) and latitude-longitude anomalies Moesinger et al. (2020), Fig. 6), indicating they are not a result of combining multiple frequencies. There are no independent VOD datasets covering the period before 1992 that could serve as validation data. Additionally, all optical-based vegetation datasets available before 1992 are multi-sensor products, such as those using the Advanced Very High Resolution Radiometer (AVHRR) sensors onboard NOAA, which are known to have their own (inter)calibration issues (Brown et al., 2006; Tian et al., 2015); therefore, their usage for this type of evaluation is questionable. In conclusion, we cannot exclude sensor intercalibration issues between SSM/I F8 and F11. We will modify the text to make the users aware of this possible issue. Potential intercalibration issues between SSM/I F8 and F11 originate from the brightness temperature calibration and not from the VODCA framework.

- To analyse the plausibility of the observed VODCA patterns, such as the decrease in 2003 and the increase in 2012, we computed global time series of yearly fAPAR (Fig. 1), as well as a

Hovmoeller diagram of monthly fAPAR anomalies (Fig. 2) using MODIS fAPAR data (Myneni and Park, 2021), which is a single-sensor dataset. We can observe a decrease in global yearly fAPAR in 2003 and an increase in 2012. Similar patterns can be observed in the Hovmoeller diagram, particularly in the Southern Hemisphere. Additionally, these patterns of decrease in 2003 are even more visible in the Hovmoller plot showing MODIS LAI anomalies, presented in (Moesinger et al., 2020) in Fig. 6. We therefore strongly believe that the patterns mentioned by the reviewer are plausible and we attribute them to natural variability.

*Fig. 1 will be added to the Annex. SSM/I information on brightness temperature calibration will be added to Chapter 2.1.2 "Passive microwave data". Information about the concatenation of SSM/I will be added to Chapter 3.2 "Preprocessing". The patterns mentioned by the reviewer (1992, 2003, 2012) are discussed in Chapter 4.2 "Spatio-temporal consistency". We mention that we cannot exclude an intercalibration issue between F08 and F11 in Chapter 4.2 "Spatio-temporal consistency" and in Chapter 5 "Conclusion".*

[Figure]

Figure 1: Yearly global time-series of VODCA CXKu and fAPAR for the bulk signal (upper graphic) and for anomalies (lower graphic).

[Figure]

Figure 2: Hovmöller diagrams showing anomalies of the monthly means per latitude for MODIS fAPAR.

2) Figure A5 & A6 should be placed in the main text than Figure 5 & 6. Because the monthly variations in Figure 5 & 6 make the system bias less noticeable. Readers need to use the yearly time series to assess the consistency.

We moved the Figures A5 and A6 to the main text body.

*Figures A5 and A6 will be moved to Figure 5 and 6 in Chapter 4.2 "Spatio-temporal consistency".*

3) In my previous review, I requested the inclusion of yearly time series for VOD products at continental or land cover scales. However, these results were not included. For this round, please add the yearly time series VOD specifically for tropical and boreal forests.

We apologize for the misunderstanding, we did not mean to omit providing continental or landcover time-series. From the formulation in the initial request ("at global, continental, or landcover scales.") we understood that the reviewer lets the authors decide which type of time-series to provide, so we provided global and hemisphere plots. We gladly provide the landcover and continental plots in this iteration. As in this iteration the reviewer asked specifically for tropical forests, we refer to the BEF class in the landcover time-series plot. To isolate boreal forest, we created time-series plots for the *Dfc* Koeppen-Geiger climate classification (Kottek et al., 2006). We also provide fAPAR time-series for comparison with VODCA CXKu and SMOS IC VOD time-series for comparison with VODCA L.

*We provide the time series per landcover (Figs. 3, 4), per continent (Figs. 7, 8) and for the Dfc climate class corresponding to boreal forest (Figs. 5, 6) in the Annex.*

[Figure]

Figure 3: Time-series per landcover class for VODCA CXKu and MODIS fAPAR. ESA CCI LC was used.

[Figure]

Figure 4: Time-series per landcover class for VODCA L and SMOS IC VOD. ESA CCI LC was used.

[Figure]

Figure 5: Time-series for the Dfc climate, corresponding to boreal forest, for VODCA CXKu and MODIS fAPAR.

[Figure]

Figure 6: Time-series for the Dfc climate, corresponding to boreal forest, for VODCA L and SMOS IC VOD.

[Figure]

Figure 7: Time-series per continent for VODCA CXKu and MODIS fAPAR.

[Figure]

Figure 8: Time-series per continent for VODCA L and SMOS IC VOD.

**References**

Berg, W.: GPM SSMI on F08 Common Calibrated Brightness Temperatures L1C 1.5 hours 13 km V07, Greenbelt, MD, USA, Goddard Earth Sciences Data and Information Services Center (GES DISC), 2021.

Berg, W., Bilanow, S., Chen, R., Datta, S., Draper, D., Ebrahimi, H., Farrar, S., Jones, W. L., Kroodsma, R., McKague, D., et al.: Intercalibration of the GPM microwave radiometer constellation, Journal of Atmospheric and Oceanic Technology, 33, 2639–2654, 2016.

Berg, W., Kroodsma, R., Kummerow, C. D., and McKague, D. S.: Fundamental climate data records of microwave brightness temperatures, Remote Sensing, 10, 1306, 2018.

Brown, M. E., Pinzón, J. E., Didan, K., Morisette, J. T., and Tucker, C. J.: Evaluation of the consistency of long-term NDVI time series derived from AVHRR, SPOT-vegetation, SeaWiFS, MODIS, and Landsat ETM+ sensors, IEEE Transactions on geoscience and remote sensing, 44, 1787–1793, 2006.

Dorigo, W., Mösinger, L., van der Schalie, R., Zotta, R.-M., Scanlon, T. M., and De Jeu, R.: Long-term monitoring of vegetation state through passive microwave satellites [in" State of the Climate in 2020"], Bulletin of the American Meteorological Society, 102, S110–S112, 2021.

Kottek, M., Grieser, J., Beck, C., Rudolf, B., and Rubel, F.: World map of the Köppen-Geiger climate classification updated, 2006.

Moesinger, L., Dorigo, W., de Jeu, R., van der Schalie, R., Scanlon, T., Teubner, I., and Forkel, M.: The global long-term microwave vegetation optical depth climate archive (VODCA), Earth System Science Data, 12, 177–196, 2020.

Myneni, R. and Park, T.: MODIS/Terra+ Aqua Leaf Area Index/FPAR 4-Day L4 Global 500 m SIN Grid V061, The Land Processes Distributed Active Archive Center (LP DAAC): Sioux Falls, SD, USA, 2021.

Tian, F., Fensholt, R., Verbesselt, J., Grogan, K., Horion, S., and Wang, Y.: Evaluating temporal consistency of long-term global NDVI datasets for trend analysis, Remote Sensing of Environment, 163, 326–340, 2015.

**Responses to Anonymous Referee #2**

We thank the Anonymous Referee #1 for their time and effort to review our revised manuscript, which helped to further increase the quality of the paper. All comments have been addressed carefully.

Below, reviewer comments are marked in red.
Responses to the comments are marked in blue.
Changes that have been made in the manuscript are marked in *italic*.

Comments

Thank you to the authors for their efforts; they have addressed most of my concerns. However, I still have a concern regarding the Hovmöller diagrams in Figure 6. Why is the VODCA CXK anomaly significantly lower and showing gaps during the period from 1988 to 1992? Could this be due to imperfections in the data fusion process? The decrease in 2003 (coinciding with the introduction of AMSR-E) and the increase in 2012 (coinciding with the introduction of AMSR2) also show similar trends. Please explain.

We thank the reviewer for these important questions. Regarding the 1988 to 1992 period, we mention that there are no data gaps; we mistakenly cut out values below -0.05 in the graphic. We corrected that as seen in Fig. 1. There are indeed lower anomalies in the period before 1992, and hence between SSM/I F8 and F11. We mention that the data producers intercalibrated the SSM/I sensors at the brightness temperature level. VOD (and soil moisture) observations were retrieved from the GPM SSMI Common Calibrated Brightness Temperatures L1C produced by NASA (Berg, 2021). The L1C dataset provides recalibrated brightness temperatures for the SSM/I sensors using a common basis (GPM GMI brightness temperature) to enable retrievals of consistent datasets for the entire SSM/I period (Berg et al., 2016, 2018; Berg, 2021). This dataset is part of the Fundamental Climate Data Record (FCDR) of brightness temperatures (Tb) and provides, according to NASA, the best intercalibration of SSM/I sensors available (Berg et al., 2018). In the VODCA framework, we concatenate the VOD retrievals from SSM/I F8, F11, and F13 into a single record without further inter-calibration at the VOD level. As F8 and F11 do not overlap with each other or any other radiometer used in the VODCA framework, further intercalibration and validation steps are challenging. We mention that the patterns in VODCA CXKu are fully compliant with patterns observed in the single-frequency VODCA v1 Ku-band (Dorigo et al. (2021), Fig. SB2.6; Moesinger et al. (2020), Fig. 6), indicating they are not a result of combining multiple frequencies. There are no independent VOD datasets covering the period before 1992 that could serve as validation data. Additionally, all optical-based vegetation datasets available before 1992 are multi-sensor products, such as those using the Advanced Very High Resolution Radiometer (AVHRR) sensors onboard NOAA. These products are known to have calibration issues (Brown et al., 2006; Tian et al., 2015); therefore, their usage for this type of validation is questionable. In conclusion, we cannot exclude sensor intercalibration issues between SSM/I F08 and SSM/I F11. We will modify the text to make the users aware of this possible issue. Potential intercalibration issues between SSM/I F8 and F11 originate from the brightness temperature calibration and not from the VODCA framework.

To analyse the plausibility of the observed VODCA patterns, such as the decrease in 2003 and the increase in 2012, we computed global time series of yearly fAPAR (Fig. 2), as well as a Hovmoeller

diagram of monthly fAPAR anomalies (Fig. 3) using MODIS fAPAR data (Myneni and Park, 2021), which is a single-sensor dataset. We can observe a decrease in global yearly fAPAR in 2003 and an increase in 2012. Similar patterns can be observed in the Hovmoeller diagram, particularly in the Southern Hemisphere. Additionally, these patterns of decrease in 2003 are even more visible in the Hovmoller plot showing MODIS LAI anomalies, presented in (Moesinger et al., 2020) in Fig. 6. We therefore strongly believe that the patterns mentioned by the reviewer are plausible and we attribute them to natural variability.

*We replace the anomaly hovmoeller from the paper with Fig. 1. Fig. 2 is added to the Annex. SSM/I information on brightness temperature calibration is added to Chapter 2.1.2 "Passive microwave data". Information about the concatenation of SSM/I is added to Chapter 3.2 "Preprocessing". The patterns mentioned by the reviewer (1992, 2003, 2012) are discussed in Chapter 4.2 "Spatio-temporal consistency". We mention that we cannot exclude an intercalibration issue between F08 and F11 in Chapter 4.2 "Spatio-temporal consistency" and in Chapter 5 "Conclusion".*

[Figure]

Figure 1: Hovmöller diagrams showing anomalies of the monthly means per latitude for VODCA CXKu and VODCA L. Anomalies have been computed as deviations from the climatology of the periods 1990 - 2020 (VODCA CXKu) and 2010 - 2021 (VODCA L).

[Figure]

Figure 2: Yearly global time-series of VODCA CXKu and fAPAR for the bulk signal (upper graphic) and for anomalies (lower graphic).

[Figure]

Figure 3: Hovmöller diagram showing anomalies of the monthly means per latitude for MODIS fAPAR. Anomalies have been computed as deviations from the long-term climatology (2000 - 2020).

**References**

Berg, W.: GPM SSMI on F08 Common Calibrated Brightness Temperatures L1C 1.5 hours 13 km V07, Greenbelt, MD, USA, Goddard Earth Sciences Data and Information Services Center (GES DISC), 2021.

Berg, W., Bilanow, S., Chen, R., Datta, S., Draper, D., Ebrahimi, H., Farrar, S., Jones, W. L., Kroodsma, R., McKague, D., et al.: Intercalibration of the GPM microwave radiometer constellation, Journal of Atmospheric and Oceanic Technology, 33, 2639–2654, 2016.

Berg, W., Kroodsma, R., Kummerow, C. D., and McKague, D. S.: Fundamental climate data records of microwave brightness temperatures, Remote Sensing, 10, 1306, 2018.

Brown, M. E., Pinzón, J. E., Didan, K., Morisette, J. T., and Tucker, C. J.: Evaluation of the consistency of long-term NDVI time series derived from AVHRR, SPOT-vegetation, SeaWiFS, MODIS, and Landsat ETM+ sensors, IEEE Transactions on geoscience and remote sensing, 44, 1787–1793, 2006.

Dorigo, W., Mösinger, L., van der Schalie, R., Zotta, R.-M., Scanlon, T. M., and De Jeu, R.: Long-term monitoring of vegetation state through passive microwave satellites [in" State of the Climate in 2020"], Bulletin of the American Meteorological Society, 102, S110–S112, 2021.

Moesinger, L., Dorigo, W., de Jeu, R., van der Schalie, R., Scanlon, T., Teubner, I., and Forkel, M.: The global long-term microwave vegetation optical depth climate archive (VODCA), Earth System Science Data, 12, 177–196, 2020.

Myneni, R. and Park, T.: MODIS/Terra+ Aqua Leaf Area Index/FPAR 4-Day L4 Global 500 m SIN Grid V061, The Land Processes Distributed Active Archive Center (LP DAAC): Sioux Falls, SD, USA, 2021.

Tian, F., Fensholt, R., Verbesselt, J., Grogan, K., Horion, S., and Wang, Y.: Evaluating temporal consistency of long-term global NDVI datasets for trend analysis, Remote Sensing of Environment, 163, 326–340, 2015.